# Unlocking bacterial potential to reduce farmland N$_2$O emissions

Elisabeth G. Hiis[1], Silas H. W. Vick[1], Lars Molstad[1], Kristine Røsdal[1], Kjell Rune Jonassen[2], Wilfried Winiwarter[3,4] & Lars R. Bakken[1✉]

Farmed soils contribute substantially to global warming by emitting N$_2$O (ref. 1), and mitigation has proved difficult[2]. Several microbial nitrogen transformations produce N$_2$O, but the only biological sink for N$_2$O is the enzyme NosZ, catalysing the reduction of N$_2$O to N$_2$ (ref. 3). Although strengthening the NosZ activity in soils would reduce N$_2$O emissions, such bioengineering of the soil microbiota is considered challenging[4,5]. However, we have developed a technology to achieve this, using organic waste as a substrate and vector for N$_2$O-respiring bacteria selected for their capacity to thrive in soil[6–8]. Here we have analysed the biokinetics of N$_2$O reduction by our most promising N$_2$O-respiring bacterium, *Cloacibacterium* sp. CB-01, its survival in soil and its effect on N$_2$O emissions in field experiments. Fertilization with waste from biogas production, in which CB-01 had grown aerobically to about $6 \times 10^9$ cells per millilitre, reduced N$_2$O emissions by 50–95%, depending on soil type. The strong and long-lasting effect of CB-01 is ascribed to its tenacity in soil, rather than its biokinetic parameters, which were inferior to those of other strains of N$_2$O-respiring bacteria. Scaling our data up to the European level, we find that national anthropogenic N$_2$O emissions could be reduced by 5–20%, and more if including other organic wastes. This opens an avenue for cost-effective reduction of N$_2$O emissions for which other mitigation options are lacking at present.

Until the mid-twentieth century, crop production was severely limited by nitrogen, requiring farmers to recycle this element in a reactive form within their agroecosystems. This constraint is reflected in the agricultural treatise by Marcus Porcius Cato (234–143 bc) *De Agri Cultura*, which recommends to "save carefully goat, sheep, cattle, and all other dung"[9]. The invention of the Haber–Bosch process in 1908 eliminated the nitrogen constraint by producing ammonium from atmospheric nitrogen. The Haber–Bosch process was a breakthrough, saving the world from starvation[10], but has also become a problem because it allowed farmers to use nitrogen in excess, with marginal economic penalties for losing nitrogen to the environment. As a result, most agroecosystems have become nitrogen-enriched and leaky, releasing ammonia to the atmosphere and nitrate to the groundwater and surface water, at scales that induce eutrophication and threaten the quality and resilience of both terrestrial and aquatic ecosystems worldwide[2,11–13]. The global scale of the problem becomes apparent when considering that the flux of reactive nitrogen into the biosphere has practically doubled since the industrial revolution, primarily owing to nitrogen produced through the Haber–Bosch process[14].

Nitrogen fertilization causes emissions of the greenhouse gas N$_2$O, both from agricultural soils themselves (direct emissions) and from the natural environments owing to the input of reactive nitrogen lost from the farms (indirect emissions). These farming-induced emissions account for substantial shares of the escalating concentration of N$_2$O in the atmosphere since the industrial revolution[1,15,16]. A comprehensive analysis of global N$_2$O emissions for 2007–2016[17] estimated that total direct and indirect emissions were 2.3–5.2 and 0.6–2.1 Tg N$_2$O-N yr$^{-1}$, respectively, in total accounting for >50% of the total anthropogenic N$_2$O emissions (4.1–10.3 Tg N$_2$O-N yr$^{-1}$).

## Mitigation

Reducing the anthropogenic impacts on nitrogen cycling and N$_2$O emissions has become a major environmental challenge for the twenty-first century owing to the severity of these issues. An obvious place to start is to improve the nitrogen-use efficiency of agroecosystems by reducing their losses of ammonia and nitrate[12]. This can be achieved by policy instruments to induce shifts in existing farming technologies and implementation of emerging ones[13,18–20].

Although improving nitrogen-use efficiency can reduce emissions, deliberately manipulating the soil microorganisms holds even greater potential for achieving substantial reductions. N$_2$O emitted from soils is produced by denitrifying bacteria, denitrifying fungi, ammonia-oxidizing archaea, ammonia-oxidizing bacteria[5] and abiotic chemical reactions[21]. Whereas ammonia-oxidizing archaea, ammonia-oxidizing bacteria and denitrifying fungi are net sources of N$_2$O because they lack the enzyme N$_2$O reductase, denitrifying bacteria can be either sinks, sources or both: N$_2$O is a free intermediate in their stepwise reduction of nitrate to molecular nitrogen, NO$_3^-$ to NO$_2^-$ to NO to N$_2$O to N$_2$, catalysed by enzymes encoded by the genes *nar* and

[1]Faculty of Chemistry, Biotechnology and Food Science, Norwegian University of Life Sciences, Ås, Norway. [2]Veas WWTP, Slemmestad, Norway. [3]International Institute for Applied Systems Analysis, Laxenburg, Austria. [4]Institute of Environmental Engineering, University of Zielona Góra, Zielona Góra, Poland. ✉e-mail: lars.bakken@nmbu.no

## Box 1

# NRB and NNRB as bacterial sinks for $N_2O$

Bacteria with a complete denitrification pathway sustain their anaerobic respiration by stepwise reduction of $NO_3^-$ to $N_2$, catalysed by four reductases, producing $NO_2^-$, NO and $N_2O$ as free intermediates:

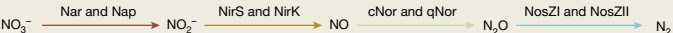

Many bacteria have a truncated pathway, lacking one to three of the reductase genes, with consequences for their role as sources or sinks for $N_2O$.

### Terminology

NRB: $N_2O$-respiring bacteria. NRB equipped with *nirS* and *nirK* and *cNor* and *qNor* are either sinks or sources for $N_2O$, depending on the regulatory network controlling their anaerobic respiration.
NNRB: non-denitrifying NRB. NNRB are NRB lacking the genes for denitrification sensu stricto (that is, *nirS* and *nirK*).

### $N_2O$ reductase types

There are two known versions of this copper enzyme: NosZI and NosZII.
Electrons are transferred to NosZII through a pathway other than NosZI, apparently generating more proton-motive force per electron[38,39].
NosZII seems to have a higher affinity for $N_2O$ (refs. 38,39).

*nap*; *nirS* and *nirK*; *cNor* and *qNor*; and *nosZ*, respectively[3]. The organisms use this pathway to sustain their respiratory metabolism under hypoxic and anoxic conditions. Denitrifying bacteria are extremely diverse regarding their catabolic potential, their regulation of denitrification[22,23] and their denitrification gene sets: a substantial share of denitrifying bacteria in soils have truncated denitrification pathways, lacking one to three of the four genes coding for the complete pathway[23,24]. This has been taken to suggest that denitrification is essentially 'modular' (that is, that each step of the pathway is catalysed by a separate group of organisms, rather than by organisms carrying out all of the steps of the pathway)[25]. The truth is probably a bit of both[4,26]. Of note, an organism with a truncated denitrification pathway lacking *nirS* and *nirK* is not a denitrifying bacterium sensu stricto.

Being the only sink for $N_2O$ in soils, the enzyme $N_2O$ reductase (NosZ) has been the target for recent attempts to mitigate $N_2O$ emissions from soils. An intervention that strengthens this sink will lower the $N_2O/N_2$ product ratio of denitrification and hence reduce the propensity of the soil to emit $N_2O$ into the atmosphere[5,27]. This can be achieved by liming to increase the soil pH: the synthesis of functional NosZ is enhanced by pushing the soil pH towards the upper end of the normal pH range of farmed soils (pH 5–7)[28]. As a result, liming acidified soils will reduce their $N_2O$ emissions by 10–20%, albeit with a next-to-neutral climate effect owing to the $CO_2$ emission induced by lime application[29,30].

## $N_2O$-respiring bacteria

Increasing the abundance of $N_2O$-respiring bacteria (NRB; Box 1) could decrease the emission of $N_2O$ (ref. 31). NRB with a complete denitrification pathway can be net sinks of $N_2O$ if their denitrification regulatory networks secure earlier and/or stronger expression of NosZ than of the other denitrification enzymes[6,32], or if their electron flow is channelled preferentially to NosZ (ref. 33). Their effect as $N_2O$ sinks is plausibly

conditional, however, as regulation of their anaerobic respiratory pathway can be influenced by environmental conditions. By contrast, bacteria that are equipped with *nosZ*, but lack *nirS* and *nirK*, are more likely to be effective sinks for $N_2O$ (ref. 34). In the following, we will call them non-denitrifying NRB (NNRB) because they are unable to denitrify, sensu stricto (Box 1). NNRB are sinks for $N_2O$ in hypoxia and anoxia, unless equipped with enzymes catalysing nitrate ammonification (that is, reduction of $NO_3^-$ to $NH_4^+$ via $NO_2^-$). Such NNRB organisms catalysing nitrate ammonification have been found to produce significant amounts of $N_2O$ if provided with high nitrate concentrations[35]; or when using $Fe^{3+}$ as electron acceptor, thus inducing abiotic $N_2O$ formation by chemical reaction of $Fe^{2+}$ with $NO_2^-$ (ref. 21).

We know too little about the ecology and physiology of NNRB to selectively enhance their growth in situ[4], but their potential as agents to reduce $N_2O$ emissions from soils is indisputable, as demonstrated by laboratory incubations of soils amended with NNRB grown ex situ[36]. Recently, it was suggested[6] that such soil amendment can be carried out inexpensively on a large scale, by using waste from biogas reactors (digestates), destined for soils as organic fertilizers, both as a substrate and vector for NRB or NNRB. By anoxic enrichment culturing with $N_2O$ as the sole electron acceptor, these authors successfully enriched and isolated NRB with a strong preference for $N_2O$, which could grow aerobically to high cell densities in digestates, and showed that amending soils with NRB-enriched digestates lowered the $N_2O/N_2$ product ratio of denitrification. The isolates obtained were not ideal, however, because they had genes for the entire denitrification pathway, and their catabolic capacities were streamlined for growth in digestate, not soil. In a follow-up study[7], the authors designed a dual substrate enrichment strategy, switching between sterilized digestate and soil as substrates, to deliberately select for NRB and NNRB with a broader catabolic capacity and physiochemical tolerance. The enrichments became dominated by strains classified as *Cloacibacterium* (based on 16S rRNA gene amplicon sequencing), and the isolated strain *Cloacibacterium* sp. CB-01 was deemed promising: it carries the genes for reduction of NO and $N_2O$ but lacks the genes for reduction of $NO_3^-$ and $NO_2^-$, thus qualifying as an NNRB (Box 1). A subsequent meta-omics analysis of the enrichments and the genome of CB-01 suggested that surface attachment and utilization of complex polysaccharides contributed to its fitness in soil[8].

Here we have evaluated the ability of CB-01 to reduce $N_2O$ emission from soil, when vectored by digestate. We examined several regulatory and enzyme kinetic traits to assess its inherent strength as an $N_2O$ sink. We then tested its capacity in 'real life' by conducting field experiments in which soils were fertilized with digestate in which CB-01 had been grown to a high cell density. Last, we assessed the potential of this technology for reducing $N_2O$ emissions across the European Union.

## The respiratory phenotype

The genome of CB-01 contains *nosZII* but lacks any genes coding for dissimilatory reduction of $NO_3^-$ and $NO_2^-$, predicting a phenotype able to respire $N_2O$ (but neither $NO_3^-$ nor $NO_2^-$), which was confirmed experimentally. In response to oxygen depletion, CB-01 reduced $N_2O$ to $N_2$, but was unable to produce $N_2O$ from $NO_2^-$ (ref. 7). The fact that it has *cNor*, coding for NO reductase, means that it could produce $N_2O$ from NO, but the NO kinetics indicates minor NO reductase activity[7]. This qualifies CB-01 as an NNRB (Box 1), and the laboratory incubation of soils fertilized with digestates containing CB-01 produced marginal amounts of $N_2O$ (ref. 7).

The capacity of a strain to reduce $N_2O$ emissions is commonly judged by a set of biokinetic parameters[31], and we investigated these for CB-01, for comparison with other strains. In all experiments (unless otherwise stated), CB-01 was grown as batch cultures in GranuCult nutrient broth (Merck) containing meat peptone and meat extract, at pH 7.3 and 23 °C.

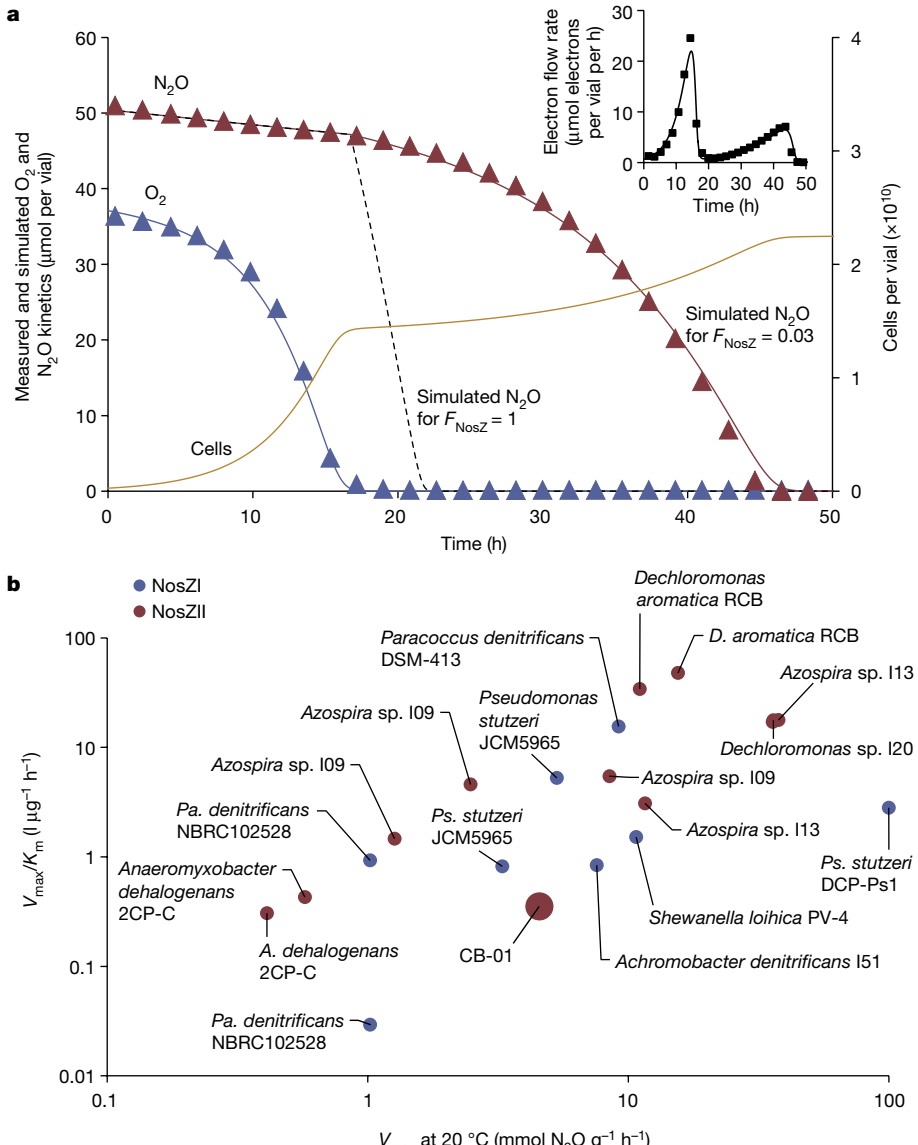

**Fig. 1 | The biokinetics of N₂O reduction, for CB-01 versus other strains.** As judged by kinetics of N₂O respiration in pure culture, CB-01 scores strikingly low compared to other N₂O-respiring organisms as a sink for N₂O: the kinetics of N₂O respiration in response to O₂ depletion indicate bet-hedging (that is, that only a fraction ($F_{NosZ}$) of the cells express NosZ and start growing by N₂O respiration after O₂ depletion). **a**, The phenomenon for a single vial. Measured O₂ and N₂O (triangles), and simulated values (solid lines), using a simplified version of the bet-hedging model of ref. 51, with $F_{NosZ}$ = 0.03. Of note, the decline of N₂O concentrations before about 18 h is due to sampling loss. The yellow line shows the simulated cell density, and the dashed black line shows simulated N₂O for $F_{NosZ}$ = 1. The inset shows measured and simulated total electron flow in the vial. Two replicate vials showed very similar kinetics, and their $F_{NosZ}$, estimated by model fitting, were 0.032 and 0.039. **b**, A condensed comparison of CB-01 with other N₂O-respiring organisms regarding its capacity to scavenge N₂O. Here we have plotted $V_{max}/K_m$ against $V_{max}$ (mmol N₂O per gram of cell dry weight per hour) for CB-01 and a range of other organisms with NosZI and NosZII, as measured by others (see Extended Data Table 1 for details and citations). The comparison shows that CB-01 is close to the average with respect to $V_{max}$, but its $V_{max}/K_m$ ratio is very low owing to the low apparent affinity for N₂O ($K_m$ = 12.9 μM N₂O).

## Growth yield

Based on the bioenergetics and charge separation for aerobic and anaerobic respiration of canonical denitrifying organisms, having NosZI (Box 1), the growth yield in terms of grams of cell dry weight per mole of electrons ($Y_{e-N_2O}$) is about 60% of that for aerobic growth ($Y_{e-O_2}$)[37]. For CB-01, which has NosZII, $Y_{e-N_2O}$ was 85% of $Y_{e-O_2}$ (Extended Data Fig. 1a,b), which lends support to the claim that electron flow to NosZII conserves more energy (by charge separation) than that to NosZI (refs. 38,39).

## Cell-specific respiration and growth rates

Measured aerobic and anaerobic respiration rates during unrestricted growth were used to estimate maximum growth rates, $\mu_{max}$, by nonlinear regression (Extended Data Fig. 1c,d), and the maximum rate of electron flow per cell to O₂ and N₂O was calculated on the basis of the measured growth yields ($V_{max} = \mu_{max}/Y$). The estimates are $\mu_{max O_2}$ = 0.29 h⁻¹ (s.d. = 0.006), $\mu_{max N_2O}$ = 0.11 h⁻¹ (s.d. = 0.001), $V_{maxO_2}$ = 0.72 fmol O₂ per cell per hour, $V_{maxN_2O}$ = 0.66 fmol N₂O per cell per hour. In terms of electron flow rates per cell, we get $V_{maxe-O_2}$ = 2.9 fmol of electrons to O₂ per cell per hour, $V_{maxe-N_2O}$ = 1.3 fmol of electrons to N₂O per cell per hour. This shows that CB-01 slows down its respiratory metabolism by about 50% when switching from aerobic to anaerobic respiration.

## Oxygen repression of N₂O respiration

N₂O respiration under oxic conditions has been reported for several organisms[31]. Such aerobic N₂O respiration would be desirable for an

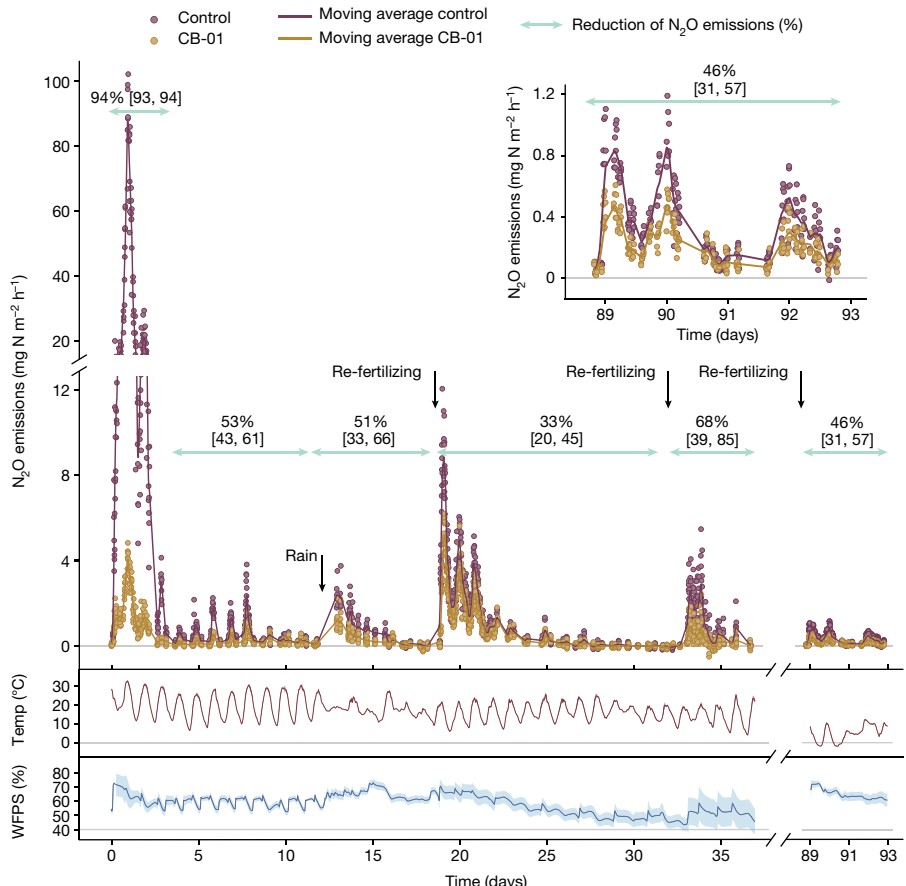

**Fig. 2 | CB-01 effects on N₂O emission from a clay loam soil of pH 6.7.** N₂O flux from buckets with soil throughout 90 days after fertilization (14 July 2021) with digestate (11 l m⁻²) in which the NNRB strain *Cloacibacterium* sp. CB-01 had been grown to about 6 × 10⁹ cells per millilitre, quantified by qPCR with primers specific for CB-01. Control buckets were fertilized with the same digestate in which CB-01 had been killed by heat (70 °C for 2 h). The buckets were sown with ryegrass (*Lolium perenne*), and the soil moisture content was sustained by daily water additions during the first 10 days. Buckets were re-fertilized with a lower dose of autoclaved and pH-adjusted digestate without CB-01 (4.6 l m⁻²) after 19, 33 and 89 days. The top panel shows N₂O flux measured by the dynamic chamber method[52] with 3 min enclosure time, operated by a field robot (Supplementary Fig. 1). The insert is a rescaled plot for day 89–93. The emissions are shown as single dots for each enclosure, and with a floating average for each treatment (solid lines, *n* = 8 replicate buckets for each treatment, calculated by a Gaussian kernel smoother). The lower panels show the average soil temperature (at 0–5.5 cm depth) and water-filled pore space (WFPS) from *n* = 4 loggers (s.d. of the mean is shown as lighter coloured ribbons). The fluxes show clear diurnal fluctuations, driven by temperature, and transient peaks in response to a rain event (day 12) and in response to re-fertilization (marked by arrows). The percentage reduction of N₂O emissions (cumulated flux) by CB-01 was calculated for selected periods, shown by the green arrows with 95% confidential intervals (Methods). The additional control buckets receiving water instead of digestates emitted negligible amounts of N₂O (result not shown).

organism to effectively scavenge N₂O in soil, but we found no evidence for this in CB-01: aerobically raised cells monitored as they depleted oxygen did not initiate N₂O respiration before the oxygen concentration reached below 1–2 µM, whereas cells previously exposed to anoxia (hence with intact NosZ enzymes) initiated N₂O respiration at 4–6 µM O₂ (Extended Data Fig. 2).

### Affinity for O₂ and N₂O

It is commonly assumed that an organism's ability to effectively mitigate N₂O emissions depends on its affinity for N₂O. We determined the apparent half-saturation constant for O₂ and N₂O reduction in CB-01 by nonlinear regression of rates per cell versus concentrations of the two gases in the liquid, and found $K_{mO_2} = 0.9$ µM O₂ (s.e. = 0.27) and $K_{mN_2O} = 12.9$ µM N₂O (s.e. = 1.2; Extended Data Fig. 3). The relatively low $K_{mO_2}$ was expected as the genome of CB-01 contains genes coding for cbb3-type high-affinity cytochrome *c* oxidases[8].

### Comparing the N₂O sink strength

To compare CB-01 with other organisms as a sink for N₂O in soil, we have summarized the biokinetic parameters for various N₂O-respiring

organisms by plotting their 'catalytic efficiency' ($V_{max}/K_m$) against their $V_{max}$ on a cell dry weight basis (Fig. 1b). This suggests that CB-01 is far from being the best among N₂O-respiring organisms: it is on par with the average of others with respect to $V_{max}$, which is a measure of the N₂O sink strength at high N₂O concentrations (»$K_m = 12.9$ µM N₂O ≈ 389 ppmv in the gas phase at 15 °C), but it scores poorly at low N₂O concentrations ($V_{max}/K_m$ for CB-01 is only 3% of the average for the others). The apparent bet-hedging (Fig. 1a), explored in more detail in several experiments (Extended Data Fig. 4) would clearly add to its inferiority as an N₂O sink. However, the bet-hedging was clearly depending on the growth medium: when growing in digestate, all cells switch to anaerobic respiration in response to oxygen depletion (Extended Data Fig. 5d–g).

### Effects of CB-01 on N₂O emissions

CB-01 was found to grow exponentially by aerobic respiration in auto-claved digestate, reaching a cell density of about 10⁹ cells per millilitre after 20 h. At this point, about 1% of the organic C in the digestate had been consumed, and the growth rate declined gradually, plausibly

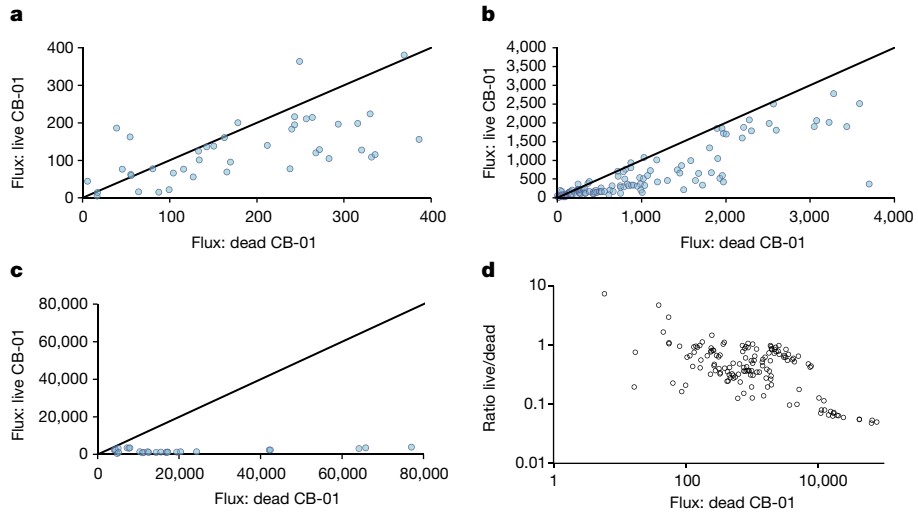

**Fig. 3 | Inspecting the contingent effect of CB-01.** Emissions from soil fertilized with digestate containing live *Cloacibacterium* sp. CB-01 plotted against the emissions from soil fertilized with digestate containing dead CB-01 cells (same data as in Fig. 2). **a**–**c**, The results for the low-emission (<400 µg N$_2$O-N m$^{-2}$ h$^{-1}$; part **a**), intermediate-emission (<4,000 µg N$_2$O-N m$^{-2}$ h$^{-1}$; part **b**) and high-emission (>4,000 µg N$_2$O-N m$^{-2}$ h$^{-1}$; part **c**) ranges. **d**, A log-scaled plot of the ratio between emissions from soil with live and dead CB-01 plotted against the emission from soil fertilized with digestate containing dead CB-01.

owing to depletion of the most easily available substrate components reaching a final density of about $6 \times 10^9$ cells per millilitre after 2 days, as judged by oxygen consumption (Extended Data Fig. 5a–c), and growth yield based on quantitative PCR (qPCR) quantification of CB-01 cells (Extended Data Fig. 1b).

We conducted three outdoor experiments in which the soils were fertilized with digestates in which CB-01 had been grown to about $6 \times 10^9$ cells per millilitre. Control treatments were fertilized with the same digestate, in which the CB-01 cells had been killed by heat (70 °C), thus securing practically identical N and C availability in the soils with and without metabolically active CB-01 cells. This type of control treatment is crucial for correctly assessing the effect of CB-01 metabolism, as the incorporation of any organic material will induce transient peaks of N$_2$O emissions. Experimental details are provided in the Methods.

The first field experiment demonstrated that the initial peak of N$_2$O flux induced by the fertilization with digestate was practically eliminated by CB-01 (Fig. 2), and that CB-01 continued to have a strong effect throughout; a second peak in N$_2$O emission induced by precipitation (day 12) was reduced by 51%; and the later emission peaks induced by re-fertilization with digestate without CB-01 (indicated by arrows) were reduced by 31, 67 and 46%.

Given the number of CB-01 cells added with the digestate ($6.6 \times 10^{13}$ cells per square metre of soil surface), and the $V_{\max} = 0.6$ fmol N$_2$O per cell per hour (Extended Data Fig. 1), the potential N$_2$O consumption rate, if all the added CB-01 cells were respiring N$_2$O at maximum rate, is 1.1 g N$_2$O-N m$^{-2}$ h$^{-1}$. The peak N$_2$O flux 1–2 days after fertilization was reduced by about 85 mg N$_2$O-N m$^{-2}$ h$^{-1}$, which is about 8% of the estimated potential. For the subsequent peaks of N$_2$O flux, the apparent N$_2$O respiration by CB-01 (that is, the reduction of the flux) was ≤4 mg N$_2$O-N m$^{-2}$ h$^{-1}$, which is ≤0.36% of the initial potential. This decline in apparent N$_2$O respiration by CB-01 was plausibly a result of two factors: a gradually declining rate of N$_2$O provision by the indigenous microbiome, and a gradually declining number of CB-01 cells.

One would expect that the effect of CB-01 as an N$_2$O sink would be marginal in periods with low emissions: low emissions are due to low water-filled pore space (that is, drained soil), low respiration rate (limited by available organic C substrates) or both, resulting in marginal hypoxic and anoxic volumes within the soil matrix[40]. Under such conditions, the primary source of N$_2$O emission could be nitrification[41], and CB-01 as an N$_2$O sink would be confined to the remaining hypoxic microsites. Inspections of the relationship between the effect of CB-01 and the N$_2$O emissions in the control soil (that is, with dead CB-01) lend some support to this: although CB-01 reduced the emissions even

for periods with modest emissions, the effect was clearly strongest in periods with high emissions (Fig. 3).

We reasoned that the capacity of CB-01 to reduce N$_2$O emissions could be influenced by soil type. Soil pH is plausibly crucial because the synthesis of functional N$_2$O reductase is increasingly impeded by declining pH within the range 4–7, both in CB-01 (ref. 7) and most other NRB[5]. Soil organic carbon content (SOC) could also have an impact. This is because the abundance of CB-01 relative to the abundance of indigenous N$_2$O-producing bacteria would be inversely related to SOC, as the abundance of indigenous bacteria in soil is directly related to SOC[42]. To explore this, we replicated the bucket experiment (Fig. 2), but with four different soils spanning a range of pH levels and including a soil with very high organic carbon content (Fig. 4 and Extended Data Fig. 6).

The emissions were low compared to those in the first experiment, plausibly owing to lower temperatures (September versus July), but CB-01 significantly reduced the emissions from all four soils. The strong effect in the acidic sandy silt soil (pH 4.15) was unexpected, as CB-01 proved unable to reduce N$_2$O at such low pH (ref. 7). However, the incorporation of digestate in this soil increased the pH(CaCl$_2$) of the sandy silt soil by more than one pH unit (Extended Data Fig. 6), reflecting its weak buffer capacity. Most probably, the CB-01 embedded in the digestate experienced an even higher local pH (pH of the digestate was 7.3). The results for the three clay loam soils show a stronger effect of soil pH: CB-01 had a clearly stronger effect in the neutral-pH clay loam (pH 6.7) than in the two more acidic clay loams (low-pH clay loam of pH 4.5; organic-rich clay loam of pH 5.26).

Finally, we scaled up to a field plot experiment, fertilizing 0.5-m$^2$ plots with digestate with live and dead CB-01, mixed into the upper 10-cm layer of the soil as in the bucket experiments. The experiment was conducted on field plots that had been limed with 2.3 kg m$^{-2}$ of dolomite in 2014, with an average pH(CaCl$_2$) = 6.13 (s.d. = 0.10). The high emissions during the first 4 days (Fig. 5) show diurnal variations, peaking when the soil temperatures reach their maximum, and a substantial effect of CB-01. Subsequent emissions, measured at low frequency throughout 280 days, were much lower and the effect of CB-01 was not statistically significant, albeit with a wide confidence interval. The very low soil temperature could be the reason for the meagre effect.

## Survival in soil

Soil microbiome engineering by inoculation is an emerging field, promising new possibilities in enhancing agricultural efficiency and sustainability[43]. It is challenging, however, because inoculants are invariably found to die out rapidly, plausibly due to a multitude

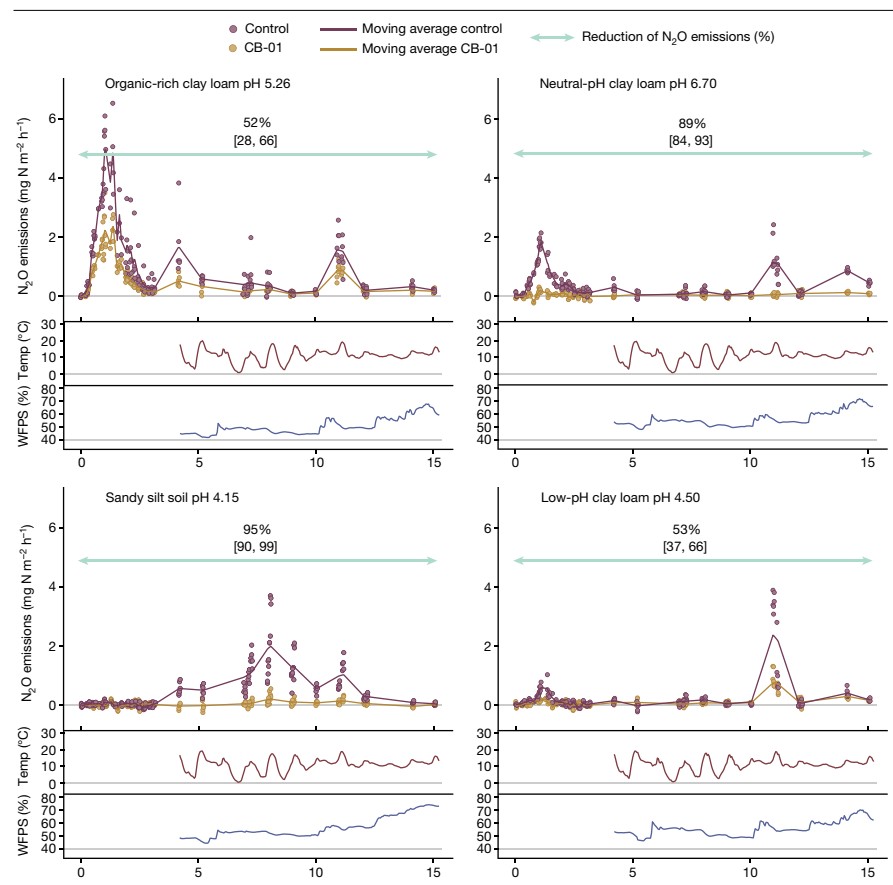

**Fig. 4 | Reduction of N₂O emissions in different soils in field buckets.** The measured emission after application of digestates with and without CB-01 to four different soils (17 September 2021). The organic carbon contents of the soils were 15.8% (organic-rich clay loam of pH 5.26), 3.21% (neutral-pH clay loam of pH 6.70), 0.75% (sandy silt soil of pH 4.15) and 3.23% (low-pH clay loam of pH 4.50) of dry weight. The pH(CaCl₂) before fertilization with digestate is given in the panels. The emissions are shown as single dots for each enclosure, and with a floating average for each treatment (solid lines, $n = 6$ replicate buckets for each treatment) as in Fig. 2. The percentage reduction of N₂O emissions (cumulated flux) by CB-01 is shown by the green arrows with 95% confidential intervals (Methods).

of abiotic and biotic barriers impeding establishment[44]. CB-01 was obtained through a dual substrate enrichment technique aimed at isolating organisms capable of withstanding the abiotic challenges of soil[7]. However, this selection process did not account for the biotic barriers that organisms may encounter in soils, such as competition for resources, antagonism and predation, as highlighted previously[45].

To assess the ability of CB-01 to survive in soil, we used qPCR with specific primers to measure the abundance of CB-01 genomes in soil (Methods) throughout the long-term field bucket experiment (Fig. 2), and throughout a laboratory incubation of soil amended with digestate with CB-01 (Methods); the results are shown in Fig. 6. During the laboratory incubation, there was a fast first-order reduction in abundance during days 3–7, and a much slower first-order reduction thereafter. By contrast, the abundance was sustained at a high level throughout 90 days in the field buckets, albeit gradually declining. The sustained CB-01 population in the bucket experiment explains why the effect on the N₂O emission was sustained (Fig. 2).

The discrepancy between the field and the laboratory experiments demands a scrutiny. In the field bucket experiment, digestate (not inoculated with CB-01) was applied three times during the course of the experiment, with soil sampling for quantification of CB-01 abundance conducted 2 days after each application. As digestate is a suitable substrate for CB-01, growth of CB-01 in response to each dose could contribute to the sustained population.

Another factor could be protozoal grazing, which was plausibly more intense in the laboratory incubations than in the field experiment, owing to the higher soil moisture content at the time of CB-01 incorporation. In the laboratory experiment, the digestate with CB-01 was dripped onto soil that was already very wet (0.53 ml per gram of soil dry weight) and retained this high soil moisture throughout. In the field bucket experiment, CB-01–digestate was harrowed into

relatively dry soil (0.34 ml per gram of soil dry weight), and the soil remained modestly moist throughout (Fig. 2). There is ample evidence that low soil moisture protects a bacterial inoculum against protozoal grazing, ascribed to increasing tortuosity, and localization of bacteria in small pores that are inaccessible to the protozoa[46]. Although we recognize that this is a speculative explanation, it warrants further experimental investigation owing to the potential practical implications.

A legitimate concern would be that the heavy inoculation with CB-01 could affect the indigenous microbiota[47]. We investigated this by analysis of 16S rRNA gene amplicons, excluding the operational taxonomic unit that circumscribed CB-01 (Methods), and found that the digestate itself had a transient impact (with or without live CB-01), but we were unable to discern any consistent difference between the treatments with live versus dead CB-01, which both converged towards the composition of pristine soil (Extended Data Fig. 7).

Laws and regulations for the use of inoculants vary from country to country, but all are likely to forbid the use of NNRB if they carry genes for antibiotic resistance or pathogenicity. We were unable to identify such genes in CB-01 (Methods).

## Extrapolating to national emissions

To assess the potential emission reductions by NNRB compared with other available techniques such as optimized N fertilization and nitrification inhibitors, we estimated emissions for Europe 2030 with the greenhouse gas and air pollution interactions and synergies (GAINS) model[48,49] (Methods).

Consistent with using a uniform emission factor in GAINS (from the Intergovernmental Panel on Climate Change (IPCC)[50]) of 1% of N applied to be emitted as N₂O, a uniform factor for emission reductions was also assumed. From the experiments, we conclude that 60% of

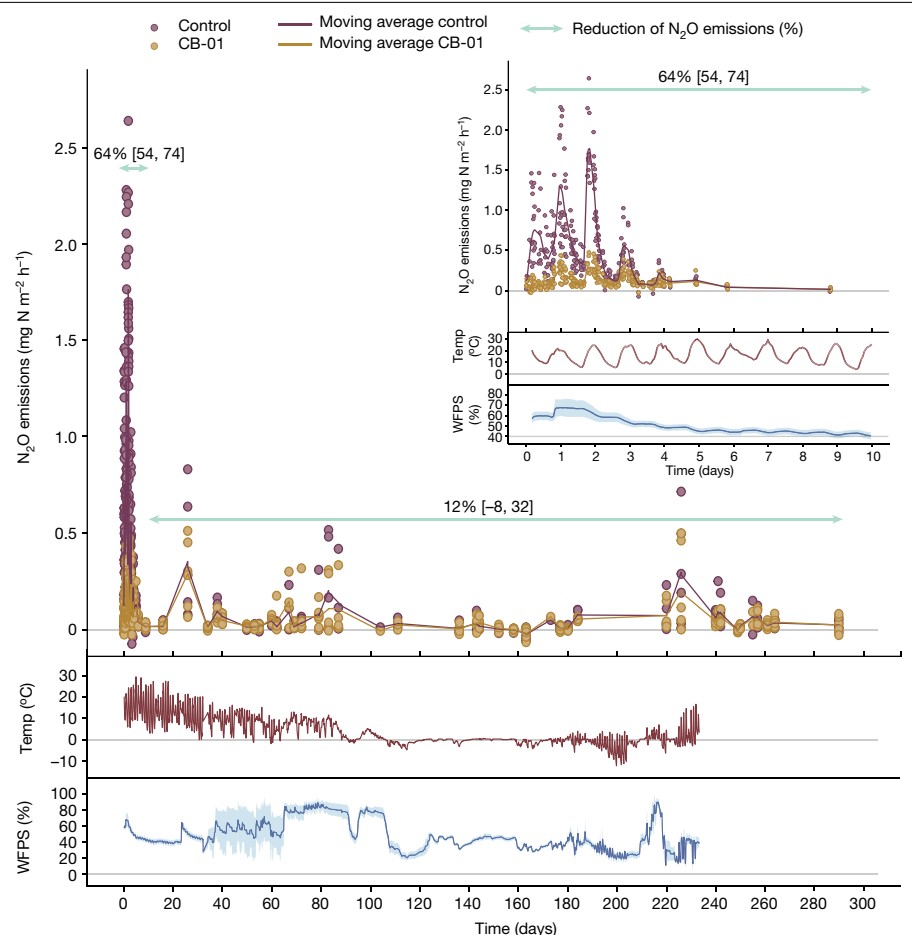

**Fig. 5 | Reduction of N₂O emissions in field plots.** The 0.5-m² field plots with clay loam of pH 6.13 were fertilized by mixing digestate into the upper 10 cm (20 August 2022), with live and dead CB-01 as in previous experiments ($n$ = 6 replicate plots for each treatment). The top panel shows emissions throughout 290 days, and the insert shows emissions during the first 10 days. The percentage reduction of N₂O emissions (cumulated flux) by CB-01 for the periods 0–10 and 10–290 days is shown by the green arrows with 95% confidential intervals (Methods). The lower panels show soil temperature and WFPS for $n$ = 4 loggers, with ribbons representing the s.d. of the mean.

emission reductions due to NNRB may be considered a conservative estimate. In Extended Data Table 3, emission reductions are shown by European country for a 2030 scenario if emissions from the application of liquid manure alone are reduced by 60%. All other anthropogenic emissions have been left unchanged. Under these assumptions, the total anthropogenic N₂O emissions from Europe decrease by 2.7% owing to NNRB being introduced and applied to all liquid manure systems. This figure is higher in countries that have a high share of liquid manure systems in their agriculture; hence, it increases to 4.0% for EU27 (27 EU member countries).

Ongoing work explores the possibility to extend the technology by growing NNRB in all types of organic waste used to fertilize soils, and by combining the application of mineral N fertilizers with incorporation of NNRB-amended organic wastes. This requires new strains, technologies and investments, but with a great potential, reducing EU27 agricultural emissions by a third (31%; Extended Data Table 3).

It needs to be pointed out that an emission reduction of 60% as derived here for NNRB is much larger than emission reductions typically reported for N₂O abatement measures. GAINS, for example, assumes nitrification inhibitors to be able to reduce emissions by as much as 38%, and high-tech mechanical fertilizer-saving technologies ('variable rate application') to be able to save only 24% of the emissions[48].

## Future development

This study presents a proof of concept demonstrating a feasible utilization of NNRB to curb N₂O emissions from farmland. By using organic waste as substrates and vectors, massive soil inoculation is achieved, which can secure reduced N₂O emissions throughout an entire growth season, despite a gradually declining NNRB abundance.

To ensure the robustness and versatility of this biotechnology, we will need an ensemble of new NNRB strains, capable of thriving in waste materials beyond digestates. New NNRB strains will probably vary

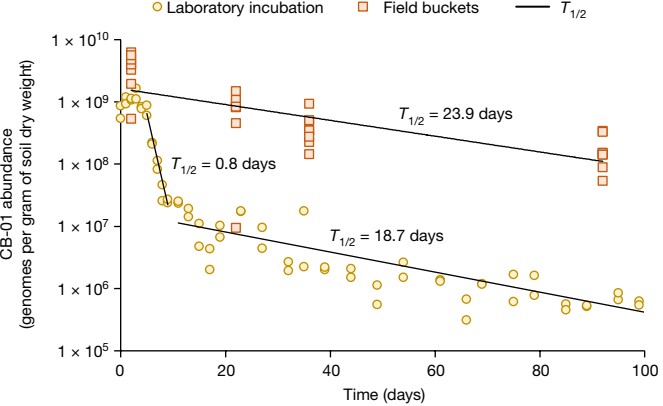

**Fig. 6 | Survival of CB-01 in soil.** The abundance of CB-01 was assessed by qPCR (Methods). The panel shows the genome abundance in the long-term field bucket experiment (Fig. 2) and in the laboratory incubation experiment (Methods). In the field bucket experiment, additional digestate (without CB-01) was incorporated 2 days before each soil sampling for qPCR. A single dot represents an individual soil sample ($n$ = 8), and the line is the fitted exponential function $N_t = N_0 e^{-d \times t}$, in which $N_t$ is the abundance at time $t$, and $d$ is the apparent first-order death rate (estimated half-life $T_{1/2} = \ln(2)/d$). For the laboratory incubation, three phases can be recognized: an initial apparent growth during the first 2–3 days, followed by a rapid first-order decline during the subsequent 4–5 days, and a slow first-order decline thereafter. Of note, the measured CB-01 genome abundance in the field plots after 280 days indicated similar average first-order death rates (0.02 per day, $T_{1/2}$ = 34 days; Extended Data Table 2).

regarding their ability to tolerate abiotic and biotic stress factors present in the soil. The dual substrate enrichment technique[7] selects for strains tolerant of abiotic, but not biotic, stress. Consequently, innovative techniques are necessary for selecting strains that tolerate the biotic stress.

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

## Methods

### Robotized batch cultivations for respiratory phenotype

NNRB have attracted much interest recently as net sinks for $N_2O$ in soils, potentially curbing $N_2O$ emissions[4,31]. NNRB strains vary grossly in their apparent capacity to act as $N_2O$ sinks, assessed by determining their biokinetic parameters: NNRB strains are commonly assumed to be strong $N_2O$ sinks if they have strong affinity (low apparent $K_m$) for $N_2O$ and a high maximal rate of $N_2O$ reduction ($V_{max}$), or simply a high catalytic efficiency (that is, a high $V_{max}/K_m$)[38]. Another desirable, albeit speculative, feature would be to reduce $N_2O$ under oxic or at least hypoxic conditions[53].

To assess *Cloacibacterium* sp. CB-01 along these criteria, we conducted in-depth investigations of its respiratory phenotype by batch culturing in the robotized incubation system designed and described previously[54,55], with the OpenLAB CDS 2.3 software for GC data acquisition (Agilent). The system hosts up to 30 parallel stirred batch cultures (normally 50 ml) in 120-ml gas-tight serum vials (crimp-sealed with butyl rubber septa) with a He atmosphere (with or without $N_2O$ and $O_2$), which are sampled frequently for measuring the concentrations of $O_2$, $N_2$, $N_2O$, NO and $CO_2$ in the headspace. Robust routines are established for calculating the rates of production and consumption of all the gases (taking sampling loss and leakage into account), and for calculating gas concentrations in the liquid as a function of measured gas concentrations in the headspace and the rate of transport between liquid and headspace. These routines are included in a spreadsheet that is publicly available, including a set of instruction videos[56]. The system has been used in numerous investigations of the respiratory phenotypes of denitrifying bacteria[6,7,33,57–62].

To enable refined analyses of the respiratory phenotype of CB-01, we initially determined the cell dry weight (femtograms per cell), and the growth yields for aerobic ($Y_{O_2}$, cells per mole of $O_2$) and anaerobic ($Y_{N_2O}$, cells per mole of $N_2O$) respiration by measuring the cell yields in batches provided with various amounts of $O_2$ and $N_2O$. This enabled inspection of the cell-specific respiration rates (fmoles per cell per hour) throughout subsequent batch incubations, based on measured rates (moles of $O_2$ and $N_2O$ per vial per hour) for each time interval between two gas samplings, and the estimated cell number in the vial for the same time interval ($=N_{ini} + Y_{O_2} \times \text{cumO}_2 + Y_{N_2O} \times \text{cumN}_2O$, in which $N_{ini}$ is the initial number of cells at time 0, and $\text{cumO}_2$ and $\text{cumN}_2O$ are the cumulated consumption of the two gases). The cell-specific rates calculated this way allowed an analysis of the affinity for $O_2$ and $N_2O$ by plotting cell-specific rates of $O_2$ and $N_2O$ against the concentrations of the two gases in the liquid as the cultures depleted the gases, and fitting the Michaelis–Menton function to these data (least squares). Batch cultures provided with both $N_2O$ and $O_2$ in the headspace were monitored as they depleted $O_2$ and switched to respiring $N_2O$, thus determining the critical concentration of $O_2$ (in the liquid) at which the cells started to respire $N_2O$. The kinetics of electron flow throughout such transitions from aerobic to anaerobic respiration were used to assess the fraction of cells expressing $N_2O$ reductase in response to $O_2$ depletion, using a simplified version of the model developed previously[60].

All phenotype experiments were conducted at 23 °C. The medium used was GranuCult nutrient broth (product number 1.05443, Merck): 8 g l$^{-1}$, containing meat peptone and meat extract, pH-adjusted to 7.3 with NaOH. Additional experiments were conducted with autoclaved digestate (aerated and pH-adjusted to 7.3, as described below).

### Culturing CB-01 in digestate for field experiments

For each field experiment, fresh digestate was collected from a wastewater treatment plant close to Oslo (VEAS), described in ref. 6. Averaged values of the quality parameters for the period of digestate collection were: dry matter content = 3.97 wt% (s.d. = 0.16),

ignition loss of dry matter = 55.6% (s.d. = 2), pH = 7.72 (s.d. = 0.07) and $NH_3 + NH_4^+ = 1.71$ g N l$^{-1}$ (s.d. = 0.12).

Before cultivation of CB-01, the digestate was heat-treated, aerated and pH-adjusted. For the field bucket experiments, the digestate was autoclaved (121 °C for 20 min), and then sparged with air (while stirred) for 48 h to secure chemical oxidation of $Fe^{2+}$ to $Fe^{3+}$, and then autoclaved again. Oxidation of $Fe^{2+}$ by air sparging was considered necessary to avoid abiotic oxygen consumption, as the digestate had high concentrations of $Fe^{2+}$ originating from the $Fe^{3+}$ used as precipitation chemicals in the primary wastewater treatment, and reduced to $Fe^{2+}$ in the anaerobic digesters[6]. The sparging caused the pH to increase to 9.4 owing to the removal of $CO_2$, requiring a final pH adjustment to 7.3 (with HCl). The same procedure was used for the field plot experiment, except that autoclaving was replaced by heat treatment: 70 °C for 4 h.

CB-01 was then grown aerobically in the pretreated digestates, inoculated to an initial cell density of about $5 \times 10^7$ cells per millilitre, which were stirred and sparged with sterile air (filtered) at 23 °C. To monitor the growth of CB-01, we transferred subsamples of each batch (after inoculation) to 120-ml vials (50 ml per vial) with Teflon-coated magnetic stirring bars, which were placed in the incubation robot system for monitoring the $O_2$ consumption (Extended Data Fig. 5a–c).

### Field experiments

Emissions of $N_2O$ in all outdoor experiments were monitored by the 'dynamic chamber' technique[52,63], operated by an autonomous field flux robot described previously[64], and shown in detail in Supplementary Fig. 1.

**Field bucket experiments.** Soils for the bucket experiments were collected from agricultural fields in southern Norway, spanning a range of soil characteristics. The acid sandy silt soil (S) was taken from an agricultural field in Solør, Norway, dominated by fluvial sandy silt soils. The clay loam soils L, I and N were from different plots within a liming experiment near the Norwegian University of Life Sciences (59° 39′ 48.2″ N 10° 45′ 44.8″ E), limed in 2014 (ref. 41): the low-pH clay loam (L) received no lime, the intermediate-pH clay loam (I) was limed with 2.3 kg m$^{-2}$ of dolomite, and the neutral-pH clay loam (N) was limed with 3 kg m$^{-2}$ of finely ground calcite. Soil O was a clay loam soil from the same area as L, I and N (hence, with similar mineral components), but with a much higher content of organic C because it had been a wetland before cultivation. The soil characteristics are listed in Extended Data Fig. 6.

The soils used in the bucket experiments (S, L, N and O) were sieved (10 mm) in moist conditions and mixed thoroughly before filling into the buckets. The conically shaped buckets (height = 21.5 cm, top diameter = 23.5 cm, bottom diameter = 21.5 cm) had a total volume of 8.6 l. An approximately 1-cm layer of gravel (4–8 mm diameter) was placed at the bottom, covered with a nylon fibre cloth to prevent eluviation of the soil by drainage. For soils S, L and N, 8 kg soil dry weight was filled into each bucket, packed by thumping the bucket on the ground until the soil had reached a bulk density of 1 kg l$^{-1}$. For the organic-rich clay loam soil, each bucket was filled with only 5.92 kg soil dry weight, reaching a bulk density of 0.74 kg l$^{-1}$ after being packed to 8 l. The soil surface area of the buckets was 0.043 m$^2$.

To secure equal initial amounts of $NO_3$ m$^{-2}$ for all soils, we mixed an amount of $KNO_3$ to each soil to reach a level of 12 g N m$^{-2}$ soil surface = 516 mg $NO_3$-N per bucket (soil surface area = 0.043 m$^2$). Digestate (480-ml per bucket = 11 l m$^{-2}$ soil surface area) was mixed into the top ≈10 cm of the soil by 'harrowing', using a small hand-held rake. We used autoclaved digestates in which CB-01 had been grown to about $6 \times 10^9$ cells per millilitre, and as the control treatment we heat-treated this digestate (70 °C, 2 h), which effectively killed the CB-01 cells (tested by measuring respiration, results not shown). As an additional control treatment, buckets received water alone. The density of CB-01 cells per soil surface area immediately after application was $6.6 \times 10^{13}$ cells m$^{-2}$.

The cell density in the upper 10 cm of the soil was about $6 \times 10^8$ cells per gram of soil dry weight for the soils S, L and N (bulk density = 1 kg l$^{-1}$), and about $8 \times 10^8$ g$^{-1}$ for soil O.

The buckets were placed on 1-m$^2$ Plexiglass plates (1.5 mm), to avoid gas exchange with the soil below. The soil moisture (volumetric water content, m$^3$ m$^{-3}$) and temperature (°C) in the upper 5.5 cm of the soil were monitored by four Teros 11 sensors, connected to an EM50 logger (Meter Group). Emissions were measured by field flux robot, lowering the chambers over the buckets (Supplementary Fig. 1g).

In the first bucket experiment, using only soil N (Extended Data Fig. 6), starting on 14 July 2021, ryegrass (*L. perenne*) was sown the day after the incorporation of the digestate, and the emissions were monitored for 90 days. Within this time span, we added 200 ml autoclaved and pH-adjusted digestate (4.6 l m$^{-2}$) without CB-01 three times (after 19, 33 and 89 days), to induce transient bursts of N$_2$O emission. By the end of each burst of N$_2$O emission induced by applying digestates, the upper 10 cm of the soil was sampled with an auger (diameter 1 cm) and stored in the freezer (−4 °C) until DNA extraction and subsequent molecular work. The auger was washed and sterilized with 70% ethanol between each sampling.

In a follow-up bucket experiment, all soils were included and monitored for 10 days, with no re-fertilization. Soil sampling was carried out after the first peak of N$_2$O emissions, as described for the 90-day bucket experiment.

The digestate application's influence on soil pH was tested in the laboratory by mixing soil with the same type and amount of digestate as applied to the 0–10-cm soil layers of the field buckets (0.11 ml per gram of soil) ±50% to show the potential pH in pockets with higher or lower than average concentration of digestate. Water was added (if needed) together with digestate to reach the same water-filled pore space (%) as in the field bucket experiment. The most prominent increase in soil pH was seen in the sandy silt soil (Extended Data Fig. 6), reflecting its low buffer capacity due to low content of clay and organic material (Extended Data Fig. 6), both known to be crucial factors determining the buffer capacity of soil[65].

**Field plot experiment.** We established small (0.5 m$^2$) test plots within larger field plots (8 m × 3 m) of a soil liming experiment (limed in 2014) on clay loam soil[41,66] and re-limed with 174 g dolomite per square metre in 2019. We used the plots with soil I (Extended Data Fig. 6) that were previously limed with dolomite to pH(CaCl$_2$) = 6.13 (s.d. = 0.10), and within each of the six replicate plots, we established two 0.7 m × 0.7 m test plots side by side (distance = 30 cm), fertilized with autoclaved digestate in which CB-01 had been grown to a cell density of about $6 \times 10^9$ cells per millilitre. We applied 4.5 l digestate per plot (= 9 l m$^{-2}$), which was mixed into the upper ≈10 cm of the soil by a hand-held cultivator. The initial density of CB-01 was $5.4 \times 10^{13}$ cells per square metre. If distributed throughout the soil layer that was sampled for analyses (0–10 cm depth = 125 kg soil dry weight per square metre, assuming a bulk density of 1.25 kg l$^{-1}$), the initial cell density in the soil would be $4.3 \times 10^8$ cells per gram of soil. Soil samples for determining CB-01 abundance were taken from each plot (three replicate samples) before incorporation of digestate with CB-01, 9 days later, and after 10 months. The soil samples were stored in the freezer (−20 °C) until DNA extraction and following quantification by PCR.

The 0.5-m$^2$ test plots were situated along the boardwalk for the autonomous field flux robot, which was used to monitor the N$_2$O emissions (Supplementary Fig. 1f).

## Calculations of emissions and statistical analyses

From the slope of the N$_2$O regression lines (Supplementary Fig. 1e), the flux of N$_2$O is calculated by the equation

$$q_{N_2O} = \frac{10^{-6}\,ahp}{RT}$$

in which $q_{N_2O}$ is the flux of N$_2$O (mol m$^{-2}$ s$^{-1}$), $a$ is the slope of the regression line (ppm s$^{-1}$), $h$ is the height (that is, the volume divided by the ground surface area) of the chamber (m), $p$ is the pressure (Pa), $R$ is the universal gas constant (J mol$^{-1}$ K$^{-1}$) and $T$ is the temperature (K).

For graphic presentation of the emissions, we used the Gaussian kernel smoother[67] to plot floating averages for each treatment (solid curves) together with individual measurements (as dots; Figs. 2, 4 and 5).

Cumulated N$_2$O emissions over a period of time are approximated by using the trapezoidal rule on the estimated fluxes ($\int q_{N_2O}(t)dt \approx \sum (q_{N_2O}(t_i) + q_{N_2O}(t_{i+1}))(t_{i+1} - t_i)/2$). This was carried out for each individual bucket and field plot.

The field plot experiment yielded paired data—six pairs ($X_i$, $Y_i$), $i = 1 \ldots 6$, in which $X_i$ are cumulated emissions from plots treated with NNRB, and $Y_i$ are cumulated emissions for control plots. This gives six ratios $R_i = X_i/Y_i$. Confidence intervals for the mean of the ratios, 1/6 $\Sigma R_i$, for two time periods were made with a Student's $t$ distribution (assuming that the ratios were normally distributed). These confidence intervals were similar to confidence intervals found by the Fieller method for ratios of paired data and also by simple nonparametric bootstrapping[68].

As the field bucket experiments did not yield paired data, flux reduction statistics are calculated as ratios of means, rather than means of ratios, of cumulated fluxes. Confidence intervals of these ratios were made by the Fieller method for unpaired data[69] and by simple nonparametric bootstrapping (the results were similar). The 95% coverage of the Fieller confidence intervals was tested by numerical simulations and a bootstrap-calibration of the confidence level was made, with negligible effects on the confidence intervals.

The plots in Figs. 2, 4 and 5 were prepared using the packages Tidyverse (v2.0.0)[70], Pracma (v2.4.2)[71], ggbreak (v0.1.2)[72], patchwork (v1.1.3)[73] and scico (v1.5.0)[74], in the R Studio software (v4.3.2)[75]. Colours used in the figures are, in general, from the scientific colour maps as described in ref. 76. The Fieller and bootstrap confidence intervals were calculated using Python (v3.11.5)[77] with Scipy (v1.11.2)[78] and Pandas (v2.1.1)[79], and Julia (v1.9.3)[80].

## Tracing CB-01 in digestate and soil

To quantify CB-01 cells in digestate and soil, we used qPCR with primers specific to members of the genus *Cloacibacterium* developed previously[81]. The primers 5′-TATTGTTTCTTCGGAAATGA-3′ (Cloac-001f) and 5′-ATGGCAGTTCTATCGTTAAGC-3′ (Cloac-001r) target a region of the 16S rRNA gene.

DNA was extracted with the DNeasy PowerSoil Pro Kit (Qiagen) according to the manufacturer's protocol, except for the first step: bead beating of the cells was carried out at 4.5 m s$^{-1}$ for 45 s in a FastPrep-24 (MP Biomedicals), instead of a vortex. To measure the concentration of DNA in the extract, we used a broad-range or high-sensitivity Qubit dsDNA Assay Kit (Thermo Fisher Scientific), depending on the expected concentration. The number of CB-01 16S rRNA gene copies in extracted DNA was quantified using a CFX96 Touch Real-Time PCR Detection System (Bio-Rad), running for 15 min at 95 °C followed by 40 cycles of denaturation (30 s at 95 °C), annealing (30 s at 55 °C) and elongation (45 s at 72 °C). The final concentration of the master mix contained 0.2 µM of each primer (Cloac-001f and Cloac-001r), and 1× HOT FIREPol EvaGreen qPCR Supermix (Solis BioDyne).

For calibration, we used DNA-extracted suspensions of washed cells containing $10^3$, $10^4$, $10^5$, $10^6$, $10^7$ and $10^8$ cells per millilitre, resulting in $2.4 \times 10^1$–$2.4 \times 10^6$ 16S templates per PCR tube (taking dilution into account, and the fact that each genome of CB-01 contains three 16S rRNA genes). Results from the qPCR were analysed using the CFX Maestro 1.1 software (v4.1.2433.1219 from Bio-Rad). To enable the use of the Cq values to estimate copy numbers, we used the generalized reduced gradient solver in Excel to fit the model (equation (1)) to the data:

$$N = \frac{N_T}{(2 \times e)^{Cq}} \qquad (1)$$

in which $N$ is the initial number of 16S rRNA gene templates in the PCR tube, $N_T$ is the number of amplicons per tube needed for signal detection (above background), $e$ is the efficiency of the PCR amplification and Cq is the number of cycles needed for detection of a signal. The fitted parameters were $N_T = 7.68 \times 10^{10}$ copies per tube and $e = 0.85$ (85% efficiency).

An independent dataset was provided by running qPCR with the same primers on extracted DNA from suspensions of unwashed CB-01 cells (in nutrient broth) with densities $10^4$, $10^5$, $10^6$, $10^7$ and $10^8$ cells per millilitre. The $\log_{10}$ values of cell densities estimated by the Cq values were on average 104% of the expected value, with a standard deviation of 6%.

When using qPCR to estimate the CB-01 abundance in soil and digestate, inhibition of the polymerase can result in too high Cq numbers, hence resulting in underestimation of the gene abundance[82]. To investigate this, we spiked the different soils and the digestate with $10^9$ CB-01 cells per gram of soil dry weight and per millilitre of digestate, respectively, extracted DNA from 0.2 g soil and 0.2 ml digestate, and eluted to a 50-μl DNA solution for each material, which was then diluted in tenfold steps from 0 (undiluted) down to $1/10^7$. The results show a reasonable fit between model (predicted) and measured Cq values for all materials if diluting the extracted DNA to ≤1/10, except for the intermediate-pH clay loam (pH(CaCl$_2$) = 6.13), which required dilution to ≤1/100 to eliminate inhibition (for further details, see Supplementary Fig. 2).

The result was used to approximate the lower limit for detection of CB-01 in soils and digestate: a cautious upper limit for Cq values to be trusted is 40 (that is, 34 templates per PCR tube; equation (1)). The polymerases were evidently inhibited by using undiluted DNA in the reaction (Supplementary Fig. 2); hence, a 1/10 dilution of the extracted DNA is needed for all soils except soil I, for which 1/100 dilution is required. This means that the PCR tube can maximally be loaded with DNA from 0.8 mg soil (0.08 mg for soil I) and 0.8 μl digestate. This implies a limit of detection around $4.3 \times 10^4$ templates per gram of soil ($4.3 \times 10^5$ for soil I owing to dilution to 1/100) and per millilitre of digestate, or $1.4 \times 10^4$ CB-01 genomes per gram of soil and per millilitre of digestate (as the genome contains three copies of the 16S rRNA gene).

The real limit of detection for a CB-01 inoculum in soil and digestate could be higher than this, if indigenous genes are amplified with the primers. This was tested by running PCR on soil and digestates that had not been spiked with CB-01, along with analysing spiked samples in various experiments. The results are summarized in Supplementary Fig. 2. As there were several tubes with a negative result (Cq > 40), average values cannot be calculated. A cautious judgement would be that the 'background' PCR signal of the soil is Cq = 39–38, which is equivalent to 67–107 templates per PCR tube, or 21–36 CB-01 genomes per tube. For all soils except I, we used the Cq values for the PCR tubes loaded with 1/10 dilutions, which were thus loaded with DNA from 0.8 mg soil. For these, the background PCR signal is equivalent to $2.6$–$4.8 \times 10^4$ CB-01 genomes per gram, and 10 times higher for soil I (owing to 1/100 dilution of the DNA from this soil). For digestate, the average Cq was 31.98 (Fig. 2), which means that the untreated digestate contains $3.2 \times 10^6$ CB-01 16S templates per millilitre, or $1.1 \times 10^6$ CB-01 genomes per millilitre.

## Survival of CB-01 in soil

**Laboratory experiment.** A soil incubation experiment was designed to assess the survival of CB-01 in soil, vectored by digestate, under constant temperature and moisture conditions, and without any subsequent incorporation of digestate (thus contrasting with the field bucket experiment, Fig. 2). CB-01 was first grown to about $6 \times 10^9$ cells per millilitre in autoclaved, aerated and pH-adjusted digestate (as for the field experiments). Neutral-pH clay loam soil (soil N, see

Extended Data Fig. 6) was portioned into a set of 50-ml Falcon tubes (9.4 g soil dry weight, moisture content = 0.5 ml g$^{-1}$ soil dry weight). To each tube, 4.2 ml sterile water and 0.85 ml digestate (with CB-01) were dripped onto the soil. The tubes were stored in a dark moist chamber at 15 °C, with loose lids to allow exchange of air. Control tubes received only sterile water. At intervals, two replicate tubes were frozen (−20 °C) for quantification of CB-01 16S rRNA gene abundance by qPCR as described above.

**Field plot experiment.** From each individual plot (Fig. 5) we took three replicate soil samples, 9 and 280 days after fertilization, for quantification of CB-01 abundance by qPCR.

## Extrapolating to national emission reductions
We use the emissions quantified with the GAINS model[48,49] for 2030 in Europe to estimate the possible reductions of the measure.

The experiments described in this paper demonstrate marked emission reductions on all soils tested, over extended periods. The strongest reductions have been seen for the initial N$_2$O peak immediately after fertilization, but NNRB has shown to remain active over a period of 90 days. Cumulated emissions over the whole period have been reduced by at least 41% (for clay loam soils), up to 95% reduction. We may disregard the case of the smallest reduction as the emissions from these soils are also rather small, but the organic loam soils (55% reductions) need to be considered. Consistent with the uniform emission factor used in GAINS (from IPCC[50]) of 1% of N applied to be emitted as N$_2$O for all conditions of crops, soil or type of fertilizer added, a uniform reduction factor of 60% of emission reductions due to NNRB, which we consider a conservative estimate, was also applied. In Extended Data Table 3, emission reductions are shown by European country for 2030 if emissions from application of liquid manure alone are reduced by 60%. This assumption is based on the understanding that liquid manure can easily be treated in biodigesters. The authors of ref. 83 assume, for the purpose of methane abatement, that anaerobic digestion becomes profitable only for large agricultural entities of at least 100 livestock units. According to GAINS numbers, this concerns 70% of all farms in Europe, which more probably reflect liquid rather than solid manure systems, so the above estimate remains valid for the main fraction of liquid manure available. Indirect emissions as well as other soil emissions due to grazing, mineral fertilizer additions or application of farmyard manure (solid manure systems) have been left unchanged. Note that the GAINS model (in agreement with IPCC[50]) does not account for potentially increased emissions due to dry periods or freeze–thaw cycles (the latter considered to potentially contribute as much as 17–28% to global soil emissions[84]) but it covers increased emissions from cropping histosols.

Under these assumptions, total N$_2$O emissions from Europe decrease by 2.7% owing to NNRB introduced. This figure is higher in countries that have a high share of liquid manure systems in their agriculture; hence, for EU27 (27 EU member countries) the corresponding figure is 4.0%, if NNRB were used for all manure nitrogen applied from liquid manure systems.

If it were possible to extend the NNRB technology, using solid manure and plant residues as substrates and vectors, we speculate emission reductions could be achieved for all mineral and natural fertilizer actively applied on fields. Ongoing work has shown that although *Cloacibacterium* sp. CB-01 grows to high cell densities in plant residues, new strains that grow in manure have been enriched and isolated (K. R. Jonassen and S. H. W. Vick, unpublished results). Although further development will be needed to implement this, it is relevant to estimate their impacts. Applying NNRB also to these other substrates at the same reduction efficiency could decrease European emissions as well as EU27 emissions by about a quarter (24% and 23%, respectively). For agricultural emissions alone, this means that roughly a third (31%) could be eliminated. For this calculation, we assume that indirect emissions from

agriculture (due to re-deposition of ammonia released from fertilizers, or due to nitrate leaching), manure-management-related emissions and emissions from histosols remain unaffected.

It needs to be pointed out that an emission reduction of 60% as derived here for NNRB is much larger than emission reductions typically reported for $N_2O$ abatement measures. For example, GAINS assumes nitrification inhibitors to be able to reduce emissions by as much as 38%, and high-tech mechanical fertilizer-saving technologies ('variable rate application') to be able to save only 24% of the emissions[48]. Of note, the percentage reduction of $N_2O$ emission by the NNRB technology is plausibly unaffected by 'variable rate application' and nitrification inhibitor, as the target for NNRB is to reduce the $N_2O/N_2$ product ratio of denitrification, whereas the two others target the concentration of $NO_3^-$ and nitrification, respectively.

## Effect of CB-01 on the soil microbiome

Microbial community composition was examined by amplicon sequencing of the 16S rRNA gene V3–V4 region. Purified DNA from soil samples was sent to Novogene Europe for amplification, library preparation and sequencing to generate 250-base-pair paired-end reads using the Illumina Novoseq platform. Reads, after primer removal, were processed using GHAP (v2.4)[85], an in-house amplicon clustering and classification pipeline built around Usearch (v11.0.66)[86], the RDP classifier (v2.13)[87] and locally written tools for generating operational taxonomic units (OTU) tables. Reads were processed using default quality control and trimming parameters. Clustering was carried out at both 97% and 100% similarity to generate OTUs and zero-radius OTUs (zOTUs), respectively. The 16S rRNA gene sequence of *Cloacibacterium* sp. CB-01 (GCA_907163125) was then matched against the OTU and zOTU representative sequences using the Usearch usearch_global command at 97% similarity and 99% similarity, respectively, to determine which OTU and zOTUs circumscribe the *Cloacibacterium* sp. CB-01 inoculant. From visual inspection it appeared that two zOTUs (zotu45 and zotu611) may circumscribe *Cloacibacterium* sp. CB-01 owing to shared abundance profiles and taxonomic classifications. To confirm that these two zOTUs both matched to *Cloacibacterium* sp. CB-01, the two representative sequences were BLAST-searched[88] against the *Cloacibacterium* sp. CB-01 genome, and it was observed that both zOTU sequences matched closely to two separate regions of the genome, presumably harbouring multiple slightly divergent copies of the 16S rRNA gene. To confirm this, the two 16S rRNA genes from the *Cloacibacterium* sp. CB-01 genome were matched back against the zOTU representative sequences using the usearch_global command at 99% similarity, at which they matched to both zotu45 and zotu611, separately. Owing to this, zotu45 and zotu611 were combined for downstream analyses.

To assess the impact of the various treatments on the soil microbial communities, α- and β-diversity measures were calculated for microbial communities from all samples using the OTU tables generated above. OTU tables were first modified by removing the OTU circumscribing *Cloacibacterium* sp. CB-01 (OTU_27) before rarifying the tables to 72,846 reads per sample using the Usearch otutab_rare command. Shannon's[89] and Simpson's[90] diversity indices were calculated using the Usearch -alpha_div command and β-diversity measures were calculated using the Usearch -beta_div command. Jaccard's dissimilarity measures[91] were then used to generate multidimensional scaling plots using the Scikitlearn MDS module[92].

The β-diversity as shown by Jaccard's dissimilarity measures indicated that early during the soil incubation period there is greater between-sample variation both within treatments and between soils treated with live CB-01 and those treated with water or dead CB-01, indicating an effect of CB-01 on the soil microbial communities (Extended Data Fig. 7a). This effect, however, disappears by the final time point, at which samples from live-CB-01-, dead-CB-01- and water-treated soils cluster together, suggesting that the effect of live CB-01 on native soil

microbial communities is transient and microbial soil communities are not affected in the longer term by the addition of live CB-01. It should be noted that the effect over time throughout the experiment is also a much larger source of microbial community variation than the addition of live CB-01 cells, presumably owing to disturbances to the soil from digging, sieving and packing of pots. Similarly, no systematic effects are observed on the α-diversity of soil microbial communities throughout the experiment indicating that the CB-01 treatment does not reduce the complexity or evenness of soil microbial communities when added to soils with digestate organic matter as can be seen in the Shannon and Simpson diversity measures of samples taken throughout the experiment (Extended Data Fig. 7b,c).

## Search for antibiotics resistance genes and pathogenicity in CB-01

Microorganisms produce secondary metabolites crucial for diverse microorganism–microorganism interactions, enhancing survivability and competitive fitness through antagonistic effects on competitors under limited growth conditions. This array of metabolites, including antibiotics, toxins, pigments, growth hormones and anti-tumour agents, can also contribute to virulence and human pathogenicity. Such traits, if encoded in the inoculant's genome, would restrict the use of such organisms as inoculants in agricultural soil. Likewise, the use of an inoculant would be restricted if its genome contains antibiotic resistance genes.

We checked CB-01 for such traits, scrutinizing its assembled draft genome[7] in Pathogenfinder (v1.1)[93] and ResFinderFG (v2.0)[94], using standard settings. This revealed no evidence of human pathogenicity or antimicrobial resistance genes.

## Reporting summary

Further information on research design is available in the Nature Portfolio Reporting Summary linked to this article.

## Data availability

Data that support the findings reported in this study are available at Figshare (https://doi.org/10.6084/m9.figshare.25130507)[95]. The assembled draft genome of CB-01 was downloaded from the European Nucleotide Archive (accession number GCA_907163125). 16S rRNA sequence data were deposited in the National Center for Biotechnology Information Sequence Read Archive database under accession number PRJNA878624.

## Code availability

Code for calculating confidence intervals of ratios of time-integrated fluxes are available at https://github.com/larsmolstad/cloacipaper_stats.

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

59.	experimental determination of NO reductase kinetics in vivo in *Paracoccus denitrificans*. *Environ. Microbiol.* **18**, 2964–2978 (2016).

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

**Acknowledgements** We thank T. Fredriksen for assistance and supervision in the field, and the Centre for Plant Research in Controlled Climate (SKP) at the Norwegian University of Life Sciences for facilitating the field experiments. This work was supported by the projects NENIM, Research Council of Norway No. 286888, and NOX2N, Research Council of Norway No. 331811.

**Author contributions** L.R.B., E.G.H., L.M., S.H.W.V., K.R. and K.R.J. designed experiments and analysed data. E.G.H., L.R.B., S.H.W.V., L.M., K.R and K.R.J. designed and conducted phenotyping experiments. E.G.H., L.R.B., S.H.W.V. and L.M. designed and conducted the field experiments. E.G.H., K.R.J. and K.R. traced CB-01 in soils. W.W. extrapolated national emission reductions. All authors contributed to writing.

**Competing interests** The authors declare no competing interests.

**Additional information**
**Correspondence and requests for materials** should be addressed to Lars R. Bakken.

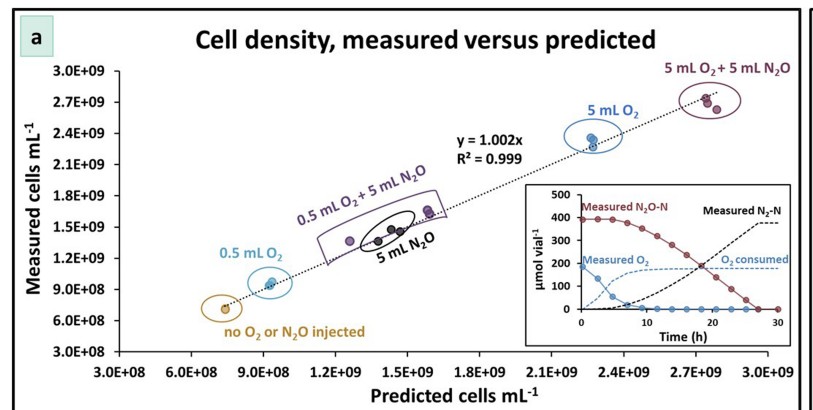

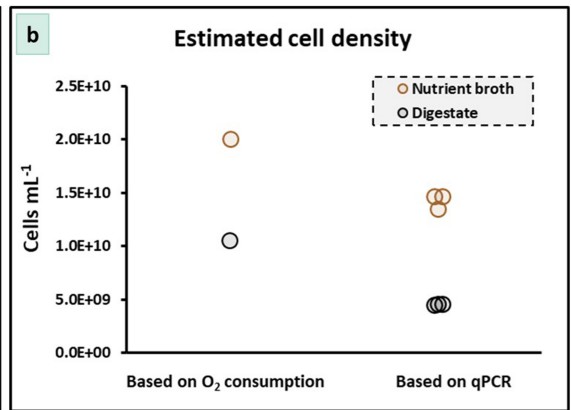

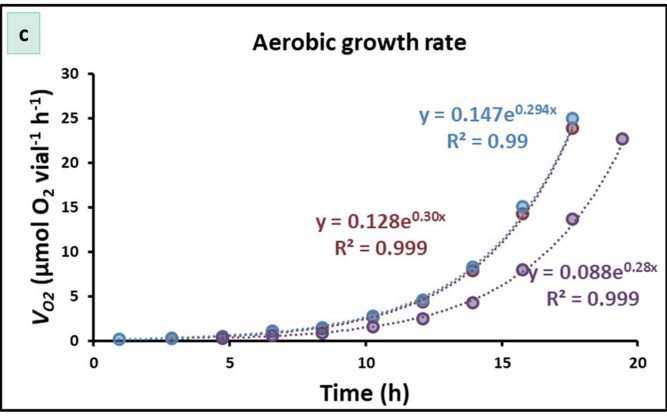

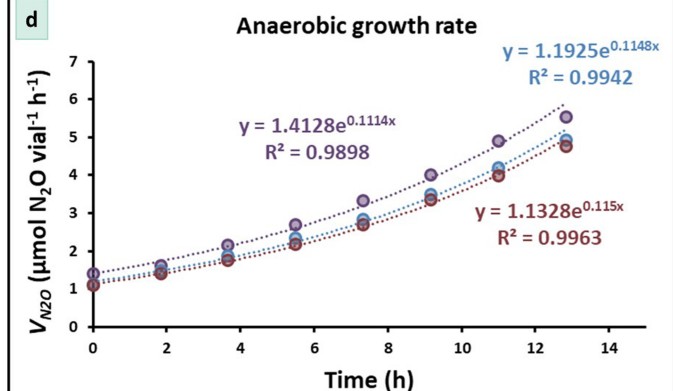

**Extended Data Fig. 1 | Growth yield and growth rates by aerobic and anaerobic respiration. Panel a:** The growth yield of CB-01 by aerobic and anaerobic respiration assessed by batch cultivation in 50 mL nutrient broth (meat-peptone and meat extract) in 120 mL vials (crimp-sealed with butyl-rubber septa) with He-atmosphere, provided with $N_2O$ and $O_2$. Vials were placed in the thermostatic water-bath (23 °C) of the robotized incubation system[54,55]. After temperature equilibration and subsequent release of overpressure due to $N_2O$- and $O_2$-injection, the vials were inoculated ($3.5 \times 10^{10}$ cells vial$^{-1}$). Based on measured $O_2$, $N_2O$ and $N_2$ in the headspace, the cumulated reduction of $O_2$ and $N_2O$ was estimated. The inserted panel shows an example of the gas kinetics in a single vial. When $O_2$ and $N_2O$ had been depleted, the cell density was measured by $OD_{600}$. The relationship between cell density and $OD_{600}$ was determined in a separate experiment comparing $OD_{600}$ with microscopic counts. A linear relationship was found for $OD_{600} \leq 0.5$ (cell density = $3.34 \times 10^9$ mL$^{-1}$ OD$^{-1}$). The cell dry weight, determined by weighing (cells washed three times in distilled water by dispersion and centrifugation, then dried at 105 °C), was 108 fg cell$^{-1}$ ± s.e. = 7.5 (n = 9). The measured yield per mol of $N_2O$ and $O_2$ was found by using the Generalized Reduced Gradient Solver in Excel (Microsoft Office 365, v2309) for the entire dataset. The panel shows the result for individual vials, as a plot of the predicted cell density (based on the yields given below) against measured cell density. The estimated yields were $Y_{N2O} = 1.7 \times 10^{14}$ cells mol$^{-1}$ $N_2O$ and $Y_{O2} = 4 \times 10^{14}$ cells mol$^{-1}$ $O_2$. The yields per mol electrons are $Y_{e-N2O} = 0.85 \times 10^{14}$ mol$^{-1}$ e$^-$ to $N_2O$ and $Y_{e-O2} = 1.0 \times 10^{14}$ cells mol$^{-1}$ e$^-$ to $O_2$. The yields in terms of dry weight g are **$Y_{e-N2O} = 9.2 \pm 0.6$ g mol$^{-1}$ e$^-$ to $N_2O$ and $Y_{e-O2} = 11 \pm 0.8$ g mol$^{-1}$ e$^-$ to $O_2$**. In comparison, Bergaust et al.[58] found *Paracoccus denitrificans* to have $Y_{e-O2} = 3.75 \times 10^{13}$ cells mol$^{-1}$ e$^-$ = 11.2 g cell dry weight mol$^{-1}$ e$^-$ to $O_2$ (cell dry weight = 298 fg), which is practically identical to

$Y_{e-O2}$ for CB-01. $Y_{e-N2O}$ was 85% of $Y_{e-O2}$ for CB-01, which is high compared to that measured for *P. denitrificans* (53%), and compared to the expectations (~ 60%) based on the charge separation per electron for aerobic and anaerobic respiration for NosZ clade I[37]. However, there is mounting evidence that the electron pathway to NosZ Clade II generates more charge separations than the pathway to NosZ Clade I, which is thermodynamically possible[31,38,96]. **Panel b:** The growth yield is plausibly declining as a culture reach stationary phase by depleting the C-sources. To assess the growth yields under these conditions, we quantified the cell densities by real-time quantitative PCR (qPCR), and compared this with the estimated cell densities based on the oxygen consumption and $Y_{O2} = 4 \times 10^{14}$ cells mol$^{-1}$ $O_2$. The panel shows this comparison for aerobic growth to high cell densities in nutrient broth and digestate, confirming lower $Y_{O2}$ when the cultures approach stationary phase, more so in digestate than in nutrient broth. We find that $Y_{O2}$ for growth in digestate is $2 \times 10^{14}$ cells mol$^{-1}$ $O_2$, which has been used to calculate cell densities in the digestate for fertilization experiments. The cell densities are based on $O_2$ consumption from a single vial, and technical replicates from that same vial were used for qPCR analysis (all points plotted). **Panel c and d:** The aerobic and anaerobic growth rates, culturing as explained for panel a. Panel c: $O_2$-consumption rates, 6 vol% $O_2$ in the headspace, inoculated with ~7 × 10$^8$ cells mL$^{-1}$, and estimated growth rate ($\mu$, h$^{-1}$) for each vial (average = 0.29 h$^{-1}$, s.d. = 0.006). Panel d: rates of $N_2O$ reduction, anoxic vials with 1.1 vol% $N_2O$ in headspace, inoculated with ~3.4 × 10$^8$ cells mL$^{-1}$, and estimated $\mu$ (average = 0.11 h$^{-1}$, s.d. = 0.001). Given these growth rates, and the growth yields (panel a) we calculate $V_{max} = \mu/Y$: $V_{maxN2O} = 0.65$ fmol $N_2O$ cell$^{-1}$ h$^{-1}$, $V_{maxO2} = 0.73$ fmol $O_2$ cell$^{-1}$ h$^{-1}$. The maximal electron transport rates are $V_{e max O2} = 2.9$ fmol e$^-$ to $O_2$ cell$^{-1}$ h$^{-1}$, $V_{e max e N2O} = 1.3$ fmol e$^-$ to $N_2O$ cell$^{-1}$ h$^{-1}$.

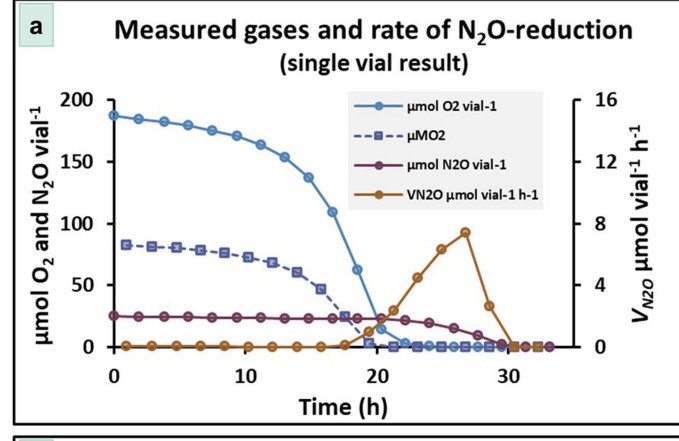

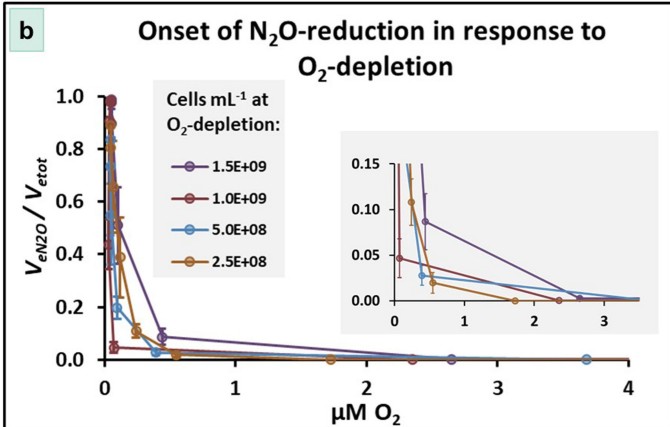

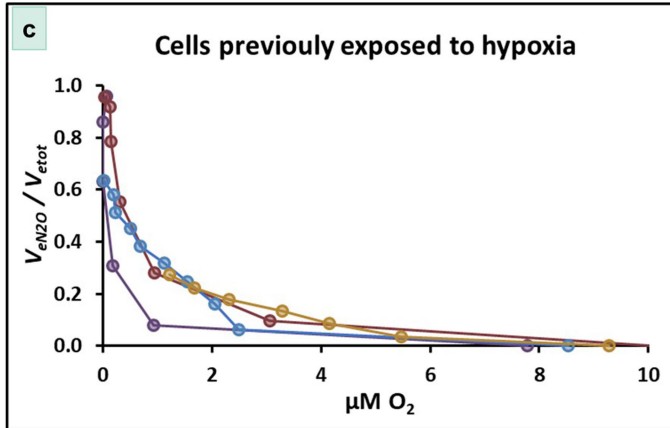

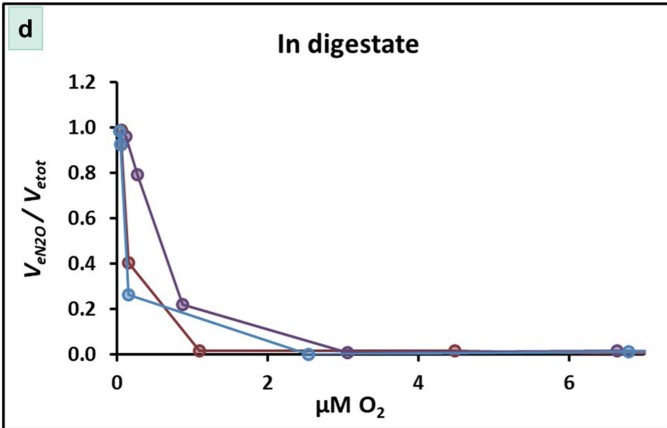

**Extended Data Fig. 2 | Onset of N₂O-reduction during O₂ depletion.**
Denitrifying bacteria vary as to how early they initiate anaerobic respiration during O₂ depletion. To explore this for *Cloacibacterium* sp. CB-01, we ran several experiments, both in nutrient broth (panel a-c) and in digestate (panel d). **Panel a and b** show the results for experiments where the inoculum was raised through >10 generations under strict oxic conditions, thus diluting out any N₂O reductase that might be present in the cells. Panel a shows the gas measurements, the O₂ concentration in the liquid as calculated from the O₂

transport rate (see Molstad et al.[54]), and the rate of N₂O-reduction ($V_{N2O}$). Panel b shows the ratio $V_{eN2O}/V_{etot}$, i.e. the fraction of total electron flow that goes to N₂O (for each time increment), plotted against the O₂ concentration in the liquid. Error bars show s.d. of the mean for n = 3 replicate vials. **Panel c** shows $V_{eN2O}/V_{etot}$ (plotted against [O₂]) for an experiment where the inoculum had been grown in hypoxia, thus with NosZ expressed already. **Panel d** shows the result for growth in digestate, inoculated with cells raised aerobically.

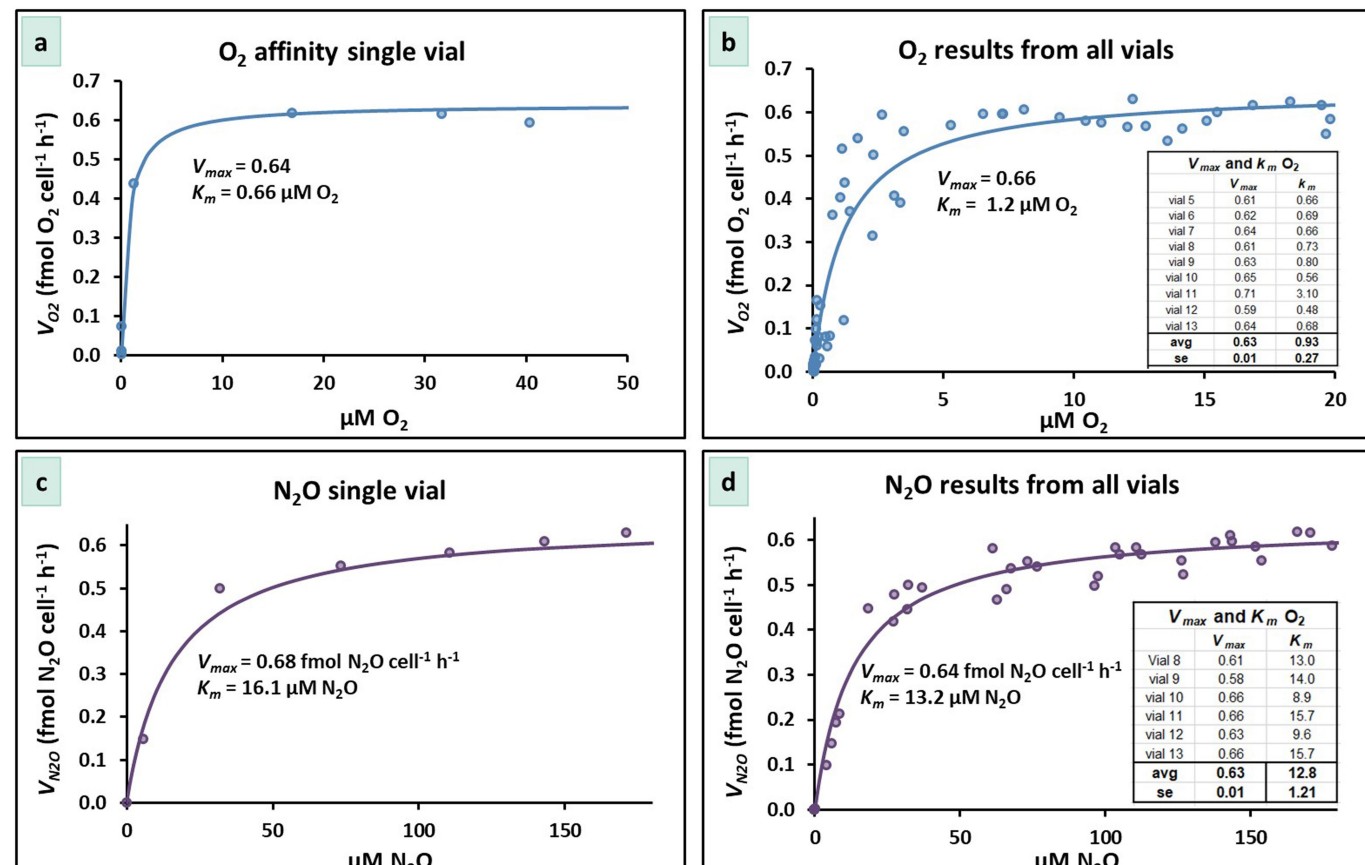

**Extended Data Fig. 3 | Apparent affinity for $O_2$ and $N_2O$.** To assess the affinity for $O_2$ and $N_2O$, we measured the rates of $O_2$- and $N_2O$ reduction as the batch cultures depleted the two electron acceptors. The rates as measured (mol vial$^{-1}$ h$^{-1}$) were converted to rates per actively respiring cell ($V_{O2}$ and $V_{N2O}$, fmol cell$^{-1}$ h$^{-1}$) based on the numbers of active cells in the vial at each time point (= the midpoint between two samplings). The number of active cells were the numbers of cells in the inoculum + new-grown cells as calculated from the cumulated consumption of $O_2$ and $N_2O$ (and the growth yield per mol, Extended Data Fig. 1), as done previously for determining the affinity for NO[59]. For $V_{N2O}$, the number of actively $N_2O$-respiring cells was only a fraction of the total

(as shown in Extended Data Fig. 4). The maximum rates $V_{max}$ and the apparent $K_m$ values were found by fitting the Michaelis Menten model V = $V_{max}$ × S / ($K_m$ + S) to the data (S is the concentration of $O_2$ and $N_2O$ in the liquid) by least square, using the Generalized Reduced Gradient Solver in Excel. This was done for each individual vial, and for the collective datasets. **Panels a and b** show the results for $O_2$, for a single vial (a) and for the entire dataset (b). Embedded in the panel is the estimated $K_m$ for each individual vial. **Panels c and d** show the results for $N_2O$, for a single vial (c) and for the entire dataset (d). Embedded in the panel is the estimated $K_m$ for each individual vial. These results show a relatively strong affinity for $O_2$ ($K_m$ ~ 1 µM $O_2$), and a rather weak affinity for $N_2O$ ($K_m$ ~ 13 µM $N_2O$).

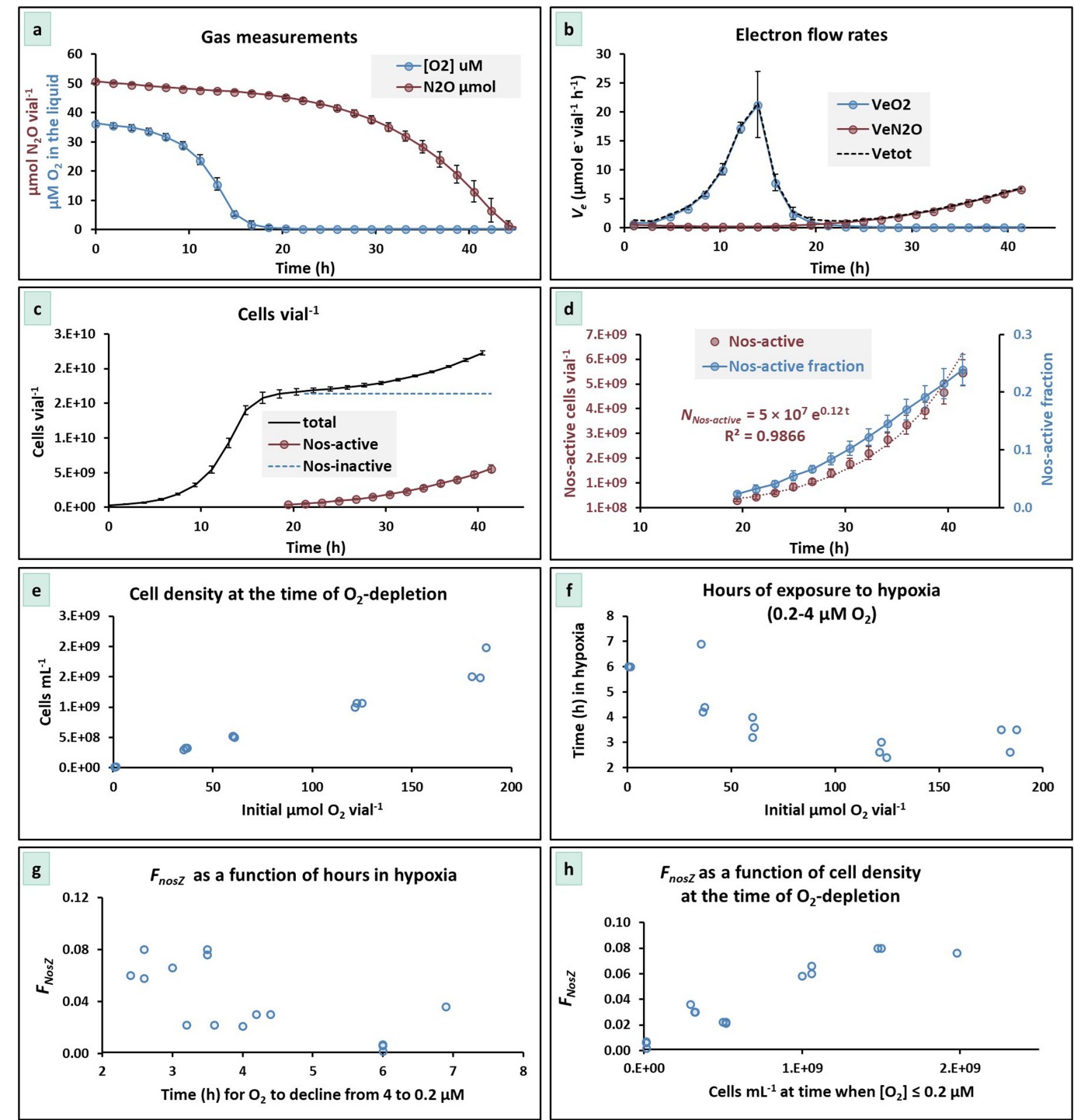

**Extended Data Fig. 4** | See next page for caption.

**Extended Data Fig. 4 | Bet-hedging during transition from aerobic to anaerobic respiration in CB-01.** To investigate the characteristic denitrification regulatory phenotype of CB-01, 120 mL vials with 50 mL nutrient broth and $O_2 + N_2O$ in He-atmosphere were inoculated with $2.7 \times 10^8$ cells per vial, which had been raised under strict oxic conditions. The vials were monitored for gas kinetics as the cultures grew by oxygen initially, and then switched to respiring $N_2O$ in response to $O_2$-depletion. The experiment included four treatments, all with 1 mL $N_2O$ (50 µmol $N_2O$ vial$^{-1}$), but five different amounts of $O_2$ (0, 0.8, 1.4, 3 and 4.6 mL $O_2$) and 3 replicate vials for each $O_2$ level. The panel shows the result for the vials with 0.8 mL initial $O_2$. Error bars represent s.d. of the mean for n = 3 replicate vials, in all panels (4a-d). **Panel a** shows the measured amounts of $O_2$ and $N_2O$ per vial. **Panel b** shows the electron flow rates to $O_2$ and $N_2O$ (and total electron flow rate as a dashed line) as calculated from the measured $O_2$ and $N_2O$. Two phenomena stand out: 1) as oxygen was depleted, the electron flow rate declined to very low values and the subsequent electron flow rate to $N_2O$ increased exponentially, and 2) the apparent growth rate is 0.12 h$^{-1}$, which is slightly higher than the anaerobic growth rate of CB-01 determined previously. This is the typical pattern for a bet-hedging denitrifying organism, i.e. an organism which expresses denitrification enzymes only in a fraction of the cells[32,51]. Assuming this, we investigated the possible fraction of cells that express NosZ and engaged in anaerobic respiration and growth: **Panel c** shows the estimated total number of cells (based on the cumulated $O_2$- and $N_2O$-consumption and the yields per mol $O_2$ and $N_2O$, Extended Data Fig. 1), and the number of $N_2O$-respiring cells (Nos-active) calculated from the measured $N_2O$-reduction rate ($V_{N2O}$, mol $N_2O$ vial$^{-1}$h$^{-1}$) and the assumption that $V_{maxN2O} = 0.65$ fmol $N_2O$ cell$^{-1}$h$^{-1}$ (as determined previously): NosZ-active cells vial$^{-1}$ = $V_{N2O}/V_{maxN2O}$. The blue dashed line is the estimated number of cells without Nos (Nos inactive), assumed to be cells entrapped in anoxia without NosZ, hence unable to synthesize NosZ. **Panel d**

shows the number of Nos-active cells as numbers per vial, and as fraction of the total number of cells in the vial. This fraction increases with time due to growth by $N_2O$-respiration, and the fraction at the time of $O_2$ depletion is a crude estimate of the fraction of cells which were able to express NosZ before $O_2$ is completely exhausted. We coin this fraction $F_{nosZ}$, analogous to $F_{den}$ for *Paracoccus denitrificans*, which is the fraction of cells that express NirS before $O_2$ is depleted[51], thus avoiding entrapment in anoxia[61]. In *P. denitrificans*, $F_{den}$ was proportional to the time length of the hypoxic phase preceding complete anoxia, ascribed to a stochastic initiation of transcription of *nirS* once the cells experience hypoxia. To investigate if the apparent bet-hedging in CB-01 shows the same pattern, we estimated $F_{nosZ}$ for 15 batch cultures, all provided with 1 mL $N_2O$ but different amounts of $O_2$ (0, 0.8, 1.4, 3 and 4.6 mL $O_2$, n = 3 replicate vials for each $O_2$-level). The cell density at the time of $O_2$-depletion increased with increasing initial $O_2$ (**panel e**), whereas the time length of exposure to hypoxia (arbitrarily defined as 0.2–4 µM $O_2$) declined (**panel f**). A simplified version of the bet-hedging model[60], assuming instantaneous expression of NosZ in a fraction of the cells ($F_{NosZ}$) as $O_2$ reached below 0.5 µM, was fitted to observed gas kinetics ($O_2$ and $N_2O$) for each vial to estimate $F_{NosZ}$ (all other parameters were as determined previously (Extended Data Figs. 1–2). Contrary to our expectations, the estimated $F_{NosZ}$ decreased with increasing time length of exposure to hypoxia (**panel g**) and increased with cell density at the time of $O_2$ depletion (**panel h**). Interestingly, practically all cells appeared to become entrapped in anoxia ($F_{NosZ} = 0.002$-$0.006$) in the vials without any $O_2$ injected. The results warrant further investigations to provide direct evidence for the cell differentiation (bet-hedging), and the mechanism causing $F_{NosZ}$ to increase with cell density. A tantalizing hypothesis is that quorum sensing induction of NosZ expression is involved. Interestingly, all cells expressed NosZ in response to oxygen depletion when growing in digestate (Extended Data Fig. 5d–g).

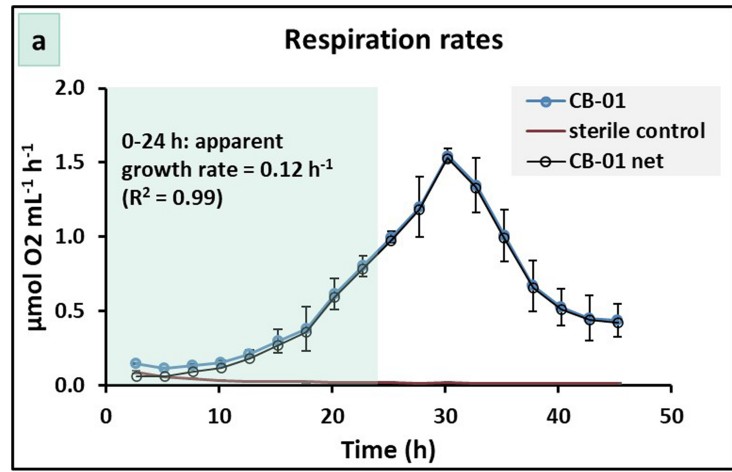

**a  Respiration rates**

0-24 h: apparent growth rate = 0.12 h⁻¹ (R² = 0.99)

Legend: CB-01, sterile control, CB-01 net

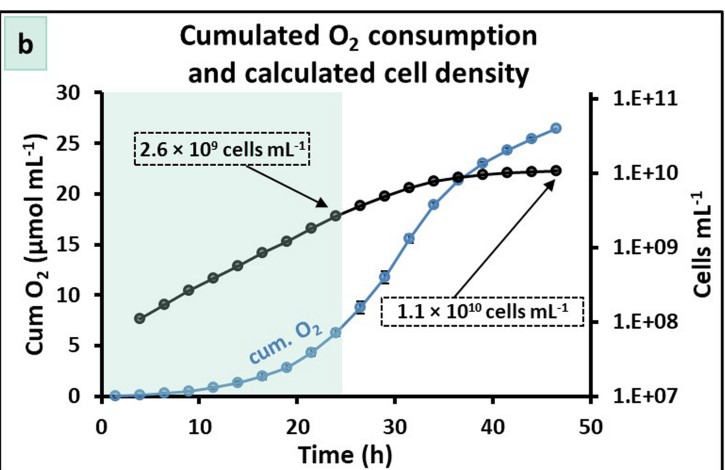

**b  Cumulated O₂ consumption and calculated cell density**

$2.6 \times 10^9$ cells mL⁻¹

cum. O₂

$1.1 \times 10^{10}$ cells mL⁻¹

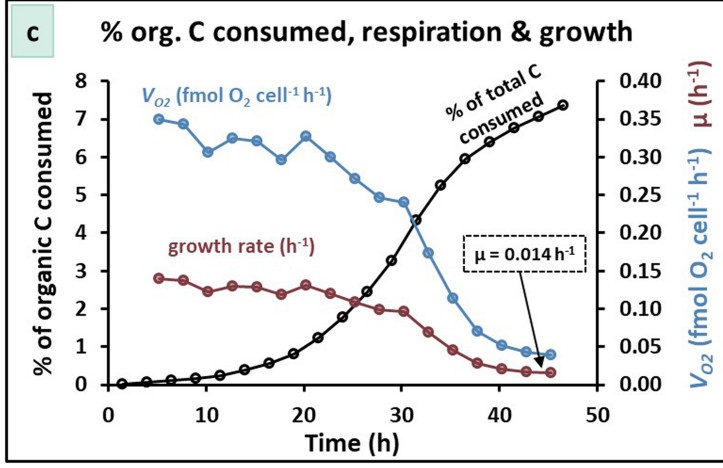

**c  % org. C consumed, respiration & growth**

$V_{O_2}$ (fmol O₂ cell⁻¹ h⁻¹)

% of total C consumed

growth rate (h⁻¹)

μ = 0.014 h⁻¹

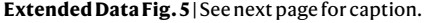

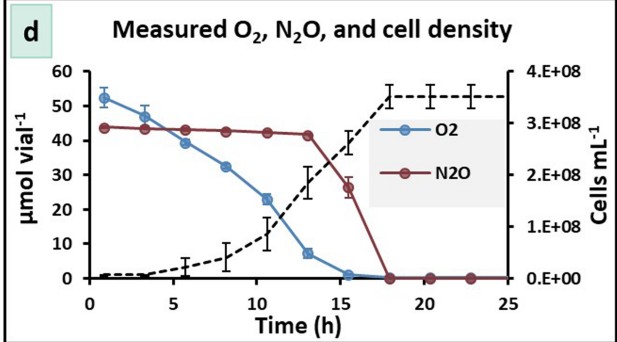

**d  Measured O₂, N₂O, and cell density**

Legend: O2, N2O

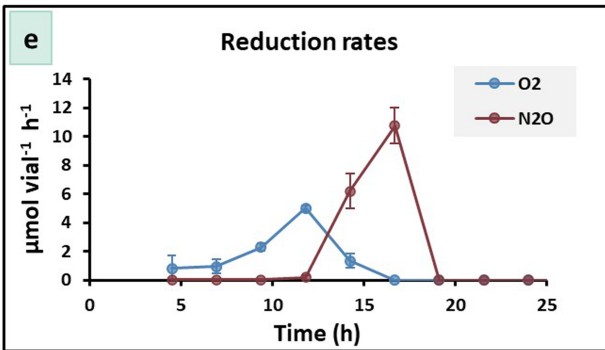

**e  Reduction rates**

Legend: O2, N2O

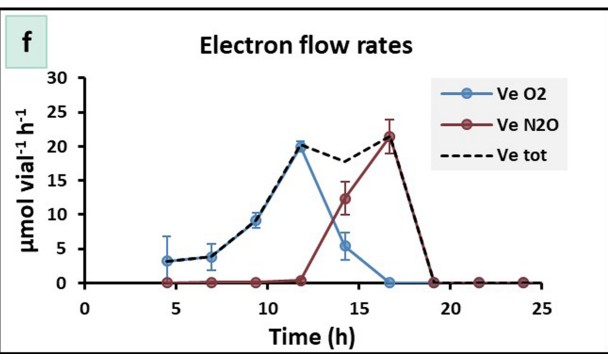

**f  Electron flow rates**

Legend: Ve O2, Ve N2O, Ve tot

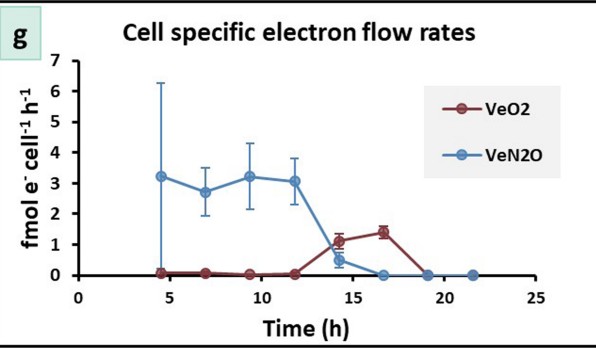

**g  Cell specific electron flow rates**

Legend: VeO2, VeN2O

**Extended Data Fig. 5** | See next page for caption.

**Extended Data Fig. 5 | Aerobic respiration, growth and and transition anaerobic respiration of CB-01 in digestate.** Panels a-c show the kinetics of $O_2$-consumption during cultivation of CB-01 in digestate for the field experiments, measured in 50 mL subsamples placed in the incubation robot. The error bars (seen in all panels except c), represent the s.d. of the mean for n = 3 replicate vials. **Panel a** shows the rates of $O_2$ consumption in vials with CB-01 and in the sterile controls, and the net consumption by CB-01 (CB-01 minus sterile control). This increased exponentially during the first 24 h, with apparent growth rate 0.12 $h^{-1}$, which is much slower than in nutrient broth (0.29 $h^{-1}$, Extended Data Fig. 1). **Panel b** shows the cumulated $O_2$ consumption by CB-01, and the estimated cell density assuming $Y_{O2} = 4.06 \times 10^{14}$ cells $mol^{-1}$ $O_2$ (Extended Data Fig. 1), reaching $1.1 \times 10^{10}$ $mL^{-1}$. The cell density quantified by qPCR for a similar experiment only 47% of the density based on $O_2$ (Extended Data Fig. 1), suggesting that $Y_{O2}$ for growth to high cell densities in digestate is ~50% of $Y_{O2}$ for optimal growth in nutrient broth (Extended Data Fig. 1b). **Panel c** shows estimated cell specific $O_2$ consumption ($V_{O2}$, fmol $O_2$ $cell^{-1}$ $h^{-1}$), estimated growth rate, μ ($h^{-1}$) = $V_{O2} \times Y$, where Y = the measured growth yield by aerobic respiration (4.06 $\times 10^{14}$ cells $mol^{-1}$ $O_2$), and the estimated fraction of organic C in the digestate (10 mg C $mL^{-1}$) consumed by CB-01 (sum of $CO_2$ and assimilated C). This suggests that ~1% of the organic C in the digestate was easily available monomers, supporting rapid growth of CB-01 to a cell density of ~1 $\times 10^9$ cells $mL^{-1}$ after 20 h, while subsequent growth was gradually declining as the organism utilized increasingly recalcitrant substrates, plausibly with a lower growth yield ($Y_{O2}$). **Panels d-g** show results of experiments designed to investigate if CB-01 is bet-hedging when growing in digestate: Bet-hedging of CB-01, as observed when cultured in nutrient broth (Extended Data Fig. 4) would reduce its capacity to scavenge $N_2O$ when vectored by digestate to soil. In theory, however, the bet-hedging could depend on the growth medium, since the fraction of cells expressing NosZ ($F_{nosZ}$) increased with the cell density (Extended Data Fig. 4), plausibly due to accumulation of compounds stimulating the expression of NosZ. On this background, we conducted experiments similar to that shown in Extended Data Fig. 4, but using digestate instead of nutrient broth as a growth medium. The panels show the results of one treatment, in which 120 mL serum vials with 50 mL autoclaved and aerated digestate (and stirring magnets), He-atmosphere + 1.4 mL $O_2$ and 1 mL $N_2O$, were inoculated with 2.5 $\times 10^8$ CB-01-cells ($5 \times 10^6$ cells $mL^{-1}$), raised under strict oxic conditions, and monitored for gas kinetics while incubated at 23 °C. **Panel d** shows the measured $O_2$ and $N_2O$, together with the cell density as calculated from the initial cell density and growth as calculated from the measured $O_2$ and $N_2O$-reduction, using the yield per mol $O_2$ and $N_2O$ determined previously ($Y_{O2} = 4 \times 10^{14}$ cells $mol^{-1}$ $O_2$, $Y_{N2O} = 1.7 \times 10^{14}$ cells $mol^{-1}$ $N_2O$, Extended Data Fig. 1). **Panel e** shows the $O_2$ and $N_2O$ consumption rates (μmol $vial^{-1}$ $h^{-1}$) for each time increment, and **panel f** shows the rates of electron flow to $O_2$ (aerobic respiration) and $N_2O$ (anaerobic respiration), and the total electron flow as a dashed line. In contrast to the results with nutrient broth (Extended Data Fig. 4) there is hardly any depression of the electron flow in response to oxygen depletion, suggesting that $F_{nosZ}$ ~ 1, i.e. all cells switch to respiring $N_2O$ (express nosZ). To inspect the validity of this further, the electron flow rates per cell (fmol $e^-$ $cell^{-1}$ $h^{-1}$) to $O_2$ ($V_{eO2}$) and $N_2O$ ($V_{eN2O}$) were calculated, and shown in **panel g**. As expected, $V_{eO2}$ remained stable around 3 fmol $e^-$ $cell^{-1}$ $h^{-1}$ until oxygen became limiting, which is close to the maximum rate determined previously ($V_{maxO2} = 0.72$ fmol $O_2$ $cell^{-1}$ $h^{-1}$ = 2.88 fmol $e^-$ $cell^{-1}$ $h^{-1}$), and $V_{eN2O}$ reached ~1.4 fmol $e^-$ $cell^{-1}$ $h^{-1}$ immediately after $O_2$-depletion, which is close to the maximum rate determined in nutrient broth cultures ($V_{maxN2O} = 0.6$ fmol $N_2O$ $cell^{-1}$ $h^{-1}$ = 1.2 fmol $e^-$ $cell^{-1}$ $h^{-1}$). All panels show the average of three replicate vials, with standard deviation as vertical lines. The experiment included treatments with 0.8, 3 and 4.6 mL $O_2$ (3 replicates of each), hence with widely different cell densities at the time of $O_2$-depletion, and they all showed $F_{nosZ}$ to be close to 1 (results not shown). In summary, CB-01 express NosZ in all cells (hence no bet-hedging) in response to $O_2$-depletion when grown in digestate, unaffected by the cell density at the time of $O_2$-depletion.

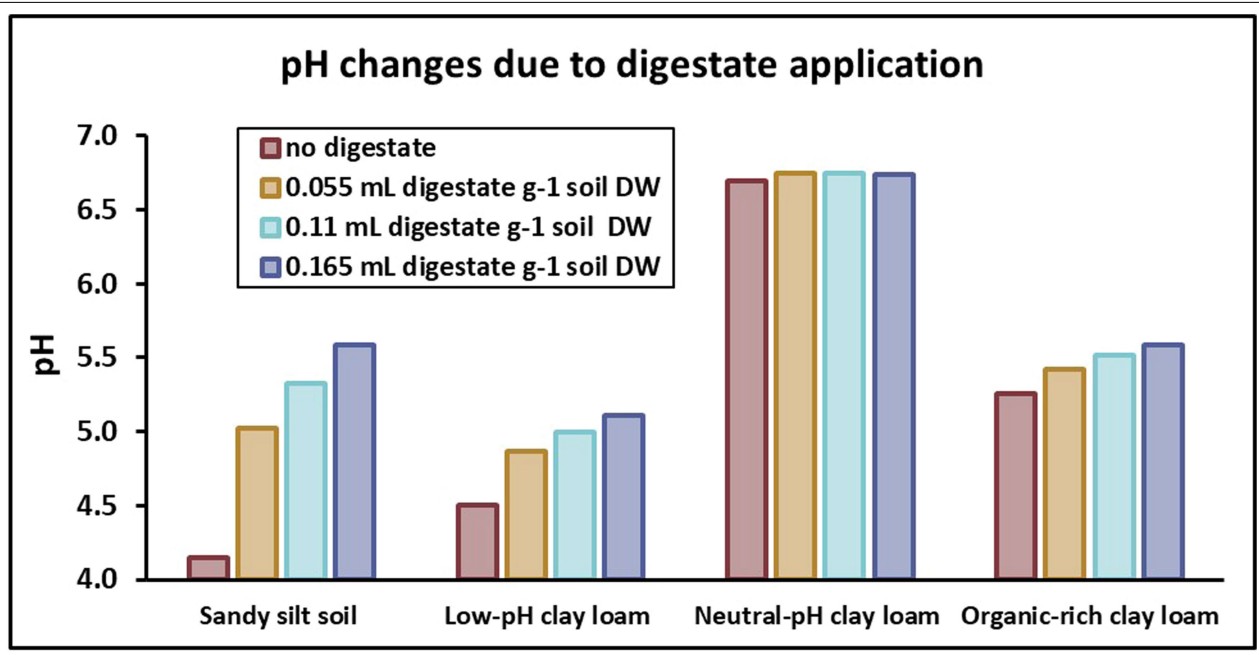

| Soil type | pH (CaCl$_2$)* | Tot C (%)** | Tot N (%)** | [NO$_3$] mg N kg$^{-1}$ ** |
|---|---|---|---|---|
| **S:** Sandy silt soil | 4.15 | 0.75 | 0.07 | 30.2 |
| **L:** Low-pH clay loam | 4.50 | 3.23 | 0.24 | 55.6 |
| **I:** Intermediate-pH clay loam | 6.13 | 3.23 | 0.24 | - |
| **N:** Neutral-pH clay loam | 6.70 | 3.21 | 0.25 | 21.1 |
| **O:** Organic-rich clay loam | 5.26 | 15.8 | 0.78 | 11.2 |

**Extended Data Fig. 6 | Soil characteristics and changes to soil pH due to application of digestate.** Some key characteristics of the different soil types are listed in the table above. The acid sandy silt soil (**S**) was taken from an agricultural field in Solør, Norway, dominated by fluvial sandy silt soils. The clay loam soils **L**, **I** and **N** were from different plots within a liming experiment near the Norwegian University of Life Sciences (59°39′48.2″N 10°45′44.8″E), limed in 2014[41]. **O** was a clay loam soil from the same area (hence with similar mineral components), but with a much higher content of organic C because it had been a wetland prior to cultivation. Soils **S**, **L**, **N** and **O** were used in the field bucket experiments. Soil **I** is the soil of the plots used for the field plot experiment. The bar chart shows pH(CaCl$_2$) in the four soils as affected by applying 0.055, 0.11 and 0.165 mL digestate g$^{-1}$ soil dry weight, which is 50, 100 and 150 % of the amounts added to the soils in the field bucket experiment (0.11 mL digestate g$^{-1}$ soil dry weight). Water was added (if needed) to reach a water-filled pore space (%) equivalent to that in the buckets after digestate application. *pH(CaCl$_2$) was measured after dispersing 10 g soil in 25 mL of 0.01 M CaCl$_2$. **Tot C = total organic C, Tot N = total organic N, and [NO$_3$] = mg NO$_3$N kg$^{-1}$ soil dry weight.

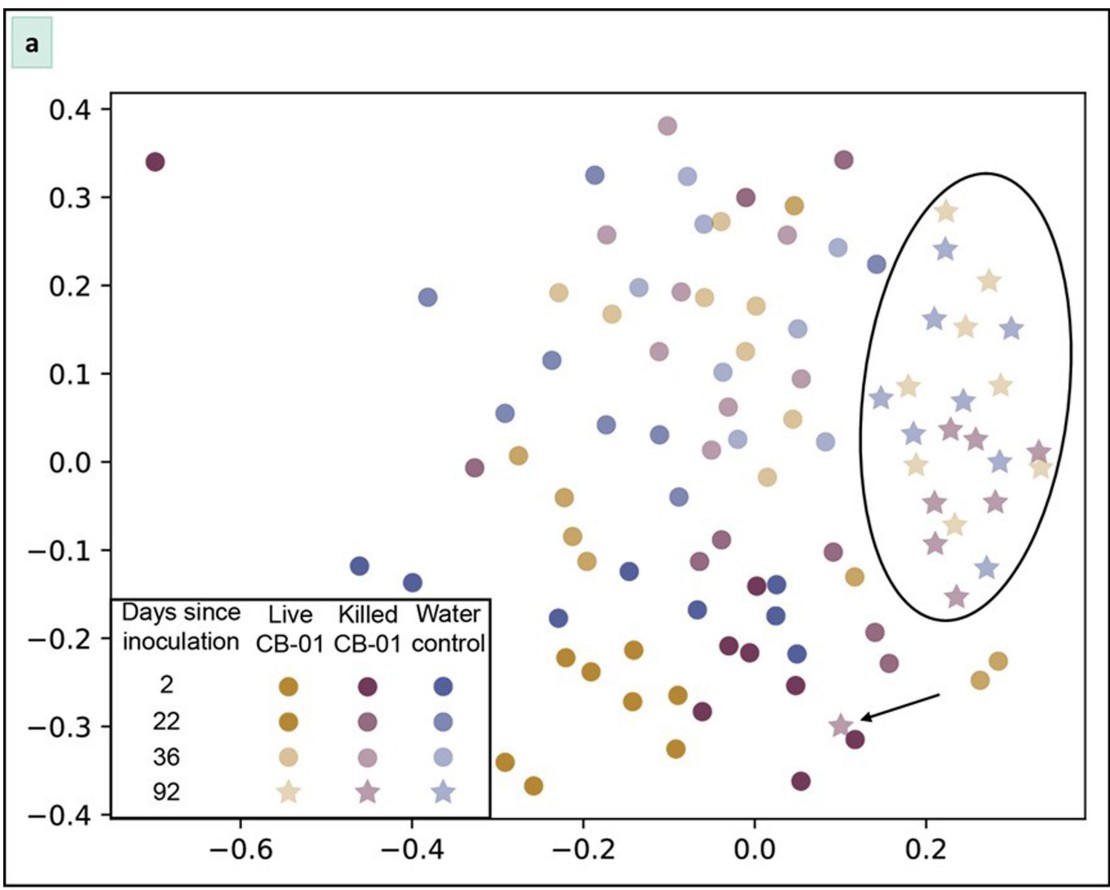

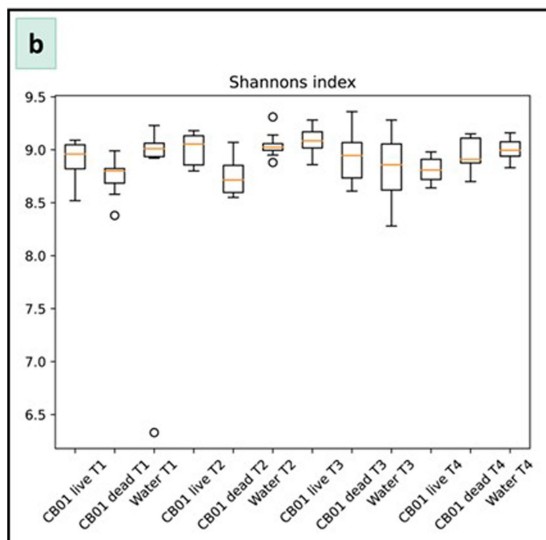

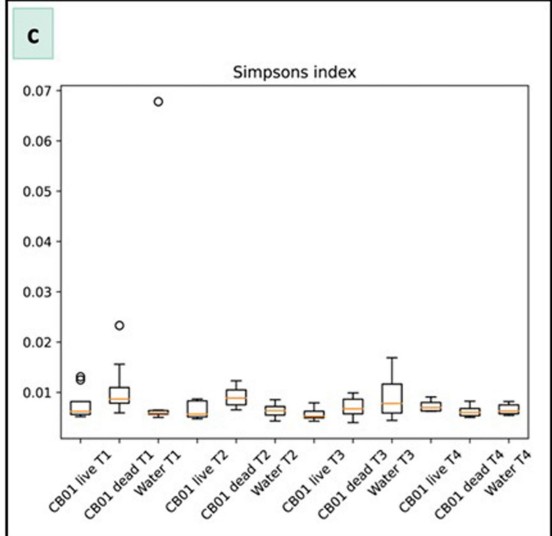

**Extended Data Fig. 7 | The influence of CB-01 inoculation on the soil microbial communities.** **Panel a** shows an MDS plot based on Jaccard's dissimilarity measures for microbial communities in the soil of the first field bucket experiment treated with live CB-01 in digestate, killed CB-01 in digestate, and water treatment only (Methods 8). The panels show the results for soils sampled 2, 22, 36 and 92 days after fertilization with digestate containing CB-01. Results for the last sampling (92 days) are encircled, and a single outlier is marked by arrow. **Panel b** shows the Shannon's diversity indices for microbial soil communities sampled throughout the field bucket

experiment treated with live CB-01 in digestate, killed CB-01 in digestate and water treatment only. **Panel c** shows the Simpson's diversity indices for microbial soil communities sampled throughout the field bucket experiment treated with live CB-01 in digestate, killed CB-01 in digestate and water treatment only. Boxes in the box plots shown in panels b and c indicate the interquartile range with a line within the box representing the median. The whiskers extending from the box extend to the furthest datapoint contained within 1.5 times the interquartile range from the boxes and circles past these whiskers denote outlier values (n = 8).

## Extended Data Table 1 | Comparison of biokinetic parameters for CB-01 and other $N_2O$-respiring bacteria

| Organism | NosZ Clade | temp °C | $V_{max}$ fmol $N_2O$ cell$^{-1}$ h$^{-1}$ | $V_{max}$ mol $N_2O$ g$^{-1}$ DW h$^{-1}$ | $K_m$ µM $N_2O$ | $V_{max20}$ mmol $N_2O$ g$^{-1}$ h$^{-1}$ at 20 °C | $V_{max20}/K_m$ L µg$^{-1}$ h$^{-1}$ at 20 °C | Reference |
|---|---|---|---|---|---|---|---|---|
| Pseudomonas stutzeri DCP-Ps1 | I | 30 | | 0.250 | 35.5 | 99.8 | 2.8 | [2] |
| Shewanella loihica PV-4 | I | 30 | | 0.027 | 7.07 | 10.7 | 1.5 | [2] |
| Paracoccus denitrificans DSM-413 | I | 20 | | 0.027 | 0.59 | 26.8 | 45.4 | [3] |
| Pseudomonas stutzeri JCM5965 | I | 30 | 1.64 | 0.008 | 4.01 | 3.3 | 0.82 | [4] |
| Paracoccus denitrificans NBRC102528 | I | 30 | 0.51 | 0.003 | 34.8 | 1.0 | 0.029 | [4] |
| Paracoccus denitrificans NBRC102528 | I | 30 | 0.51 | 0.003 | 1.1 | 1.0 | 0.93 | [5] |
| Pseudomonas stutzeri JCM5965 | I | 30 | 2.66 | 0.013 | 1.01 | 5.3 | 5.26 | [5] |
| Alicycliphilus denitrificans I51 | I | 30 | 3.78 | 0.019 | 8.98 | 7.6 | 0.84 | [6] |
| Dechloromonas aromatica RCB | II | 30 | | 0.028 | 0.324 | 11.1 | 34.1 | [2] |
| Anaeromyxobacter dehalogenans 2CP-C | II | 30 | | 0.001 | 1.34 | 0.41 | 0.31 | [2] |
| Cloacibacterium sp. CB-01 | II | 23 | | 0.006 | 12.9 | 4.56 | 0.35 | this study |
| Azospira sp. I09 | II | 30 | 0.634 | 0.003 | 0.868 | 1.27 | 1.46 | [4] |
| Azospira sp. I13 | II | 30 | 5.8 | 0.029 | 3.76 | 11.6 | 3.09 | [4] |
| Azospira sp. I09 | II | 30 | 1.24 | 0.006 | 0.54 | 2.48 | 4.59 | [5] |
| Azospira sp. I13 | II | 30 | 18.84 | 0.094 | 2.12 | 37.7 | 17.77 | [5] |
| Dechloromonas sp. I20 | II | 30 | 18 | 0.090 | 2.04 | 36.0 | 17.65 | [6] |
| Azospira sp. I09 | II | 30 | 4.23 | 0.021 | 1.55 | 8.5 | 5.46 | [6] |
| Azospira sp. I13 | II | 30 | 17.9 | 0.090 | 2.1 | 35.8 | 17.0 | [6] |
| Dechloromonas aromatica RCB | II | 30 | 7.748 | 0.039 | 0.324 | 15.5 | 47.8 | [6] |
| Anaeromyxobacter dehalogenans 2CP-C | II | 30 | 0.287 | 0.001 | 1.34 | 0.57 | 0.43 | [6] |

The maximum rates of $N_2O$-respiration ($V_{max}$), and the affinity for $N_2O$, normally expressed as the half saturation concentration ($K_m$), have been measured in various $N_2O$-respiring strains to assess and compare their capacity to scavenge $N_2O$ in soil. A plausible way to rank strains according to their capacity to reduce $N_2O$ at low $N_2O$-concentrations is to calculate their $V_{max}/K_m$ ratios, since this approximates the slope of the rate against $N_2O$-concentration ([$N_2O$]) at very low concentrations ([$N_2O$] « $K_m$). In the table below, we have listed $V_{max}$ and $K_m$ determined for a range of $N_2O$-respiring strains. Some report $V_{max}$ as fmol $N_2O$ cell$^{-1}$ h$^{-1}$, and others report it as mol g$^{-1}$ cell dry weight h$^{-1}$. To enable a comparison, we have converted all $V_{max}$ values to mol $N_2O$ g$^{-1}$ cell dry weight h$^{-1}$, assuming a cell volume = ~0.6 µm$^3$ and 200 fg dry weight per cell[97], and converted all values to $V_{max}$ at 20 °C assuming the rates increase exponentially with temperature by a factor of 2.5 per 10 °C increase in temperature ($Q_{10}$=2.5). The calculated $V_{max20}/K_m$ values are plotted against $V_{maxN20}$ in Fig. 1b in the main paper. refs. 38,60,97–100.

**Extended Data Table 2 | Estimated abundance of CB-01 genomes in the field plot experiment**

| plot pair | Plots with live CB-01 | | | | Plots with heat-killed CB-01 | | | |
| --- | --- | --- | --- | --- | --- | --- | --- | --- |
| | Time = 9 days | | Time = 280 days | | Time = 9 days | | Time = 280 days | |
| | avg | se | avg | se | avg | se | avg | se |
| a | 698 007 | 194 735 | 2 331 | 1 891 | 7 064 | 3 446 | 70 | 33 |
| b | Na* | - | 3 510 | 1 321 | 98 195 | 90 261 | 203 | 17 |
| c | 1 531 | 533 | 652 | 469 | 424 | 196 | 160 | 79 |
| d | 471 431 | 189 461 | 1 229 | 650 | 58 446 | 29 613 | 176 | 14 |
| e | 2 298 | 1 619 | 814 | 531 | 43 173 | 27 419 | 81 | 44 |
| f | 608 617 | 599 881 | 677 | 447 | 599 | 281 | 191 | 82 |
| average | 356 377 | 149 141 | 1 536 | 471 | 34 650 | 16 092 | 147 | 23 |

The numbers are the average genome abundance based on qPCR analysis of 3 replicate samples for each indivdual plot (unit: $10^3$ genomes $g^{-1}$ soil dry-weight), taken 9 and 280 days after fertilization with CB-01 containing digestate (live or heat killed). Initial numbers of CB-01 (at time 0) was ~6 × $10^8$ cells $g^{-1}$ soil dry-weight, as calculated from the measured $O_2$-consumption during cultivation in the digestate (Extended Data Fig. 5a–c), corrected for the low growth yield in digestate (Extended Data Fig. 1b), and the amount of digestate per g soil dry-weight). The genome abundance in the soil samples taken 9 days after fertilization are remarkably variable, suggesting a fast initial decline in two of the plots with live CB-01 (plot c and e), at rates comparable to the rapid initial decline in the laboratory incubation experiment (Fig. 6 in the main paper), tentatively ascribed to protozoal grazing. The variability suggests a patchy distribution of protozoa in the field. Of note, the genome abundance after 280 days in the plots with live CB-01 are in some agreement with the first order decline rates estimated for the field bucket experiment: The decline in genome abundance from 3.56 × $10^8$ at time = 9 days to 1.5 × $10^6$ at time 280 days implies an average first order decay rate of 0.02 $d^{-1}$ (=average half life of 34 days). *samples lost

**Extended Data Table 3 | Potential emission reductions as a consequence of implementing NNRB measures**

| | Total N$_2$O emissions<br><br>kt N$_2$O y$^{-1}$ | Reduction by NNRB kt N$_2$O y$^{-1}$ | | % Reduction of total emissions | | % reduction of agricultural emissions |
| --- | --- | --- | --- | --- | --- | --- |
| | | NNRB in liquid manure only | NNRB with all N-fertilizers | NNRB in liquid manure only | NNRB with all N-fertilizers | NNRB with all N-fertilizers |
| Albania | 4.3 | 0.10 | 0.85 | 2.3% | 20% | 24% |
| Austria | 13.4 | 0.80 | 2.99 | 5.9% | 22% | 34% |
| Belarus | 44.7 | 0.20 | 8.29 | 0.4% | 19% | 23% |
| Belgium | 23.0 | 0.96 | 5.11 | 4.2% | 22% | 33% |
| Bosnia-Herzegovina | 3.9 | 0.12 | 0.75 | 3.0% | 19% | 25% |
| Bulgaria | 13.6 | 0.09 | 4.44 | 0.6% | 33% | 40% |
| Croatia | 6.0 | 0.12 | 1.67 | 2.0% | 28% | 34% |
| Cyprus | 0.9 | 0.05 | 0.15 | 5.9% | 16% | 21% |
| Czech Republic | 19.2 | 0.18 | 5.11 | 0.9% | 27% | 37% |
| Denmark | 18.8 | 1.83 | 5.24 | 9.8% | 28% | 33% |
| Estonia | 3.2 | 0.06 | 0.73 | 1.8% | 23% | 31% |
| Finland | 17.6 | 0.45 | 2.93 | 2.5% | 17% | 25% |
| France | 143.3 | 5.16 | 36.76 | 3.6% | 26% | 31% |
| Germany | 132.5 | 7.46 | 30.09 | 5.6% | 23% | 31% |
| Greece | 14.0 | 0.18 | 2.60 | 1.3% | 19% | 25% |
| Hungary | 17.4 | 0.18 | 5.85 | 1.1% | 34% | 41% |
| Iceland | 1.0 | 0.04 | 0.24 | 4.3% | 24% | 28% |
| Ireland | 30.6 | 1.40 | 5.73 | 4.6% | 19% | 20% |
| Italy | 60.0 | 3.08 | 13.28 | 5.1% | 22% | 33% |
| Kosovo | 1.2 | 0.02 | 0.18 | 1.5% | 14% | 28% |
| Latvia | 4.9 | 0.08 | 0.86 | 1.6% | 17% | 20% |
| Lithuania | 13.2 | 0.30 | 2.58 | 2.2% | 20% | 22% |
| Luxembourg | 1.2 | 0.05 | 0.18 | 4.3% | 15% | 26% |
| Malta | 0.2 | 0.01 | 0.04 | 2.3% | 17% | 32% |
| Moldavia | 2.6 | 0.03 | 0.58 | 1.0% | 22% | 28% |
| Montenegro | 0.5 | 0.01 | 0.06 | 1.5% | 11% | 19% |
| Netherlands | 36.0 | 2.69 | 5.28 | 7.5% | 15% | 23% |
| Northern Macedonia | 1.9 | 0.04 | 0.29 | 2.0% | 15% | 25% |
| Norway | 11.1 | 0.41 | 2.17 | 3.7% | 20% | 30% |
| Poland | 81.3 | 2.32 | 20.65 | 2.8% | 25% | 33% |
| Portugal | 11.4 | 0.32 | 1.91 | 2.8% | 17% | 23% |
| Romania | 28.8 | 0.48 | 8.00 | 1.7% | 28% | 35% |
| Russia | 291.4 | 2.22 | 79.37 | 0.8% | 27% | 33% |
| Serbia | 10.7 | 0.31 | 2.78 | 2.9% | 26% | 37% |
| Slovakia | 7.3 | 0.08 | 2.09 | 1.1% | 29% | 40% |
| Slovenia | 2.4 | 0.16 | 0.46 | 6.8% | 19% | 31% |
| Spain | 62.1 | 2.47 | 14.75 | 4.0% | 24% | 32% |
| Sweden | 18.7 | 0.49 | 3.47 | 2.6% | 19% | 29% |
| Switzerland | 9.3 | 0.57 | 2.01 | 6.1% | 22% | 31% |
| Turkey | 109.4 | 0.81 | 23.45 | 0.7% | 21% | 28% |
| Ukraine | 73.4 | 0.34 | 18.61 | 0.5% | 25% | 34% |
| United Kingdom | 89.4 | 1.69 | 19.39 | 1.9% | 22% | 29% |
| **All Europe (incl. non-European parts of Russia and Turkey)** | 1435.9 | 38.51 | 341.97 | 2.7% | 24% | 31% |
| **sum EU27** | 781.0 | 31.46 | 182.97 | 4.0% | 23% | 31% |

(anthropogenic emissions in kt N$_2$O per year projected for 2030).

# Reporting Summary

## Statistics

For all statistical analyses, confirm that the following items are present in the figure legend, table legend, main text, or Methods section.

| n/a | Confirmed | |
|-----|-----------|---|
| ☐ | ☒ | The exact sample size (*n*) for each experimental group/condition, given as a discrete number and unit of measurement |
| ☐ | ☒ | A statement on whether measurements were taken from distinct samples or whether the same sample was measured repeatedly |
| ☒ | ☐ | The statistical test(s) used AND whether they are one- or two-sided<br>*Only common tests should be described solely by name; describe more complex techniques in the Methods section.* |
| ☐ | ☒ | A description of all covariates tested |
| ☐ | ☒ | A description of any assumptions or corrections, such as tests of normality and adjustment for multiple comparisons |
| ☐ | ☒ | A full description of the statistical parameters including central tendency (e.g. means) or other basic estimates (e.g. regression coefficient) AND variation (e.g. standard deviation) or associated estimates of uncertainty (e.g. confidence intervals) |
| ☒ | ☐ | For null hypothesis testing, the test statistic (e.g. *F*, *t*, *r*) with confidence intervals, effect sizes, degrees of freedom and *P* value noted<br>*Give P values as exact values whenever suitable.* |
| ☒ | ☐ | For Bayesian analysis, information on the choice of priors and Markov chain Monte Carlo settings |
| ☒ | ☐ | For hierarchical and complex designs, identification of the appropriate level for tests and full reporting of outcomes |
| ☒ | ☐ | Estimates of effect sizes (e.g. Cohen's *d*, Pearson's *r*), indicating how they were calculated |

*Our web collection on statistics for biologists contains articles on many of the points above.*

## Software and code

Policy information about availability of computer code

| | |
|---|---|
| Data collection | OpenLAB (v2.3). |
| Data analysis | R Studio version 4.3.2 (2023-06-16 ucrt) with tidyverse v2.0.0, pracma v2.4.2, ggbreak v0.1.2, patchwork v1.1.3 and scico v1.5.0.<br>Microsoft Excel for Microsoft 365 MSO (v2309 Build 16.0.16827.20278) 64-bit.<br>Bio-Rad CFX Maestro 1.1 (v4.1.2433.1219).<br>Pathogenfinder (v1.1).<br>ResFinderFG (v2.0).<br>Greenfield Hybrid Analysis Pipeline (v2.4).<br>Usearch (v11.0.66).<br>RDP classifier (v2.13).<br>BLAST (v2.13.0).<br>Scikitlearn (v1.1.1) MDS module.<br>Python 3.11.5 with Scipy 1.11.2 and Pandas 2.1.1.<br>Julia (v1.9.3).<br>Custom code: https://github.com/larsmolstad/cloacipaper_stats |

For manuscripts utilizing custom algorithms or software that are central to the research but not yet described in published literature, software must be made available to editors and reviewers. We strongly encourage code deposition in a community repository (e.g. GitHub). See the Nature Portfolio guidelines for submitting code & software for further information.

## Data

Policy information about availability of data
All manuscripts must include a data availability statement. This statement should provide the following information, where applicable:
- Accession codes, unique identifiers, or web links for publicly available datasets
- A description of any restrictions on data availability
- For clinical datasets or third party data, please ensure that the statement adheres to our policy

Source data for all the figures in the manuscript and the Extended Data is available in the figshare repository (DOI: 10.6084/m9.figshare.25130507). The assembled draft genome of CB-01 was downloaded from the European Nucleotide Archive (accession GCA_907163125, https://www.ebi.ac.uk/ena/browser/view/GCA_907163125). 16S rRNA sequence data was deposited in the NCBI Sequence Read Archive database under accession PRJNA878624 (https://www.ncbi.nlm.nih.gov/bioproject/PRJNA878624).

## Research involving human participants, their data, or biological material

Policy information about studies with human participants or human data. See also policy information about sex, gender (identity/presentation), and sexual orientation and race, ethnicity and racism.

| | |
|---|---|
| Reporting on sex and gender | N/A |
| Reporting on race, ethnicity, or other socially relevant groupings | N/A |
| Population characteristics | N/A |
| Recruitment | N/A |
| Ethics oversight | N/A |

Note that full information on the approval of the study protocol must also be provided in the manuscript.

# Field-specific reporting

Please select the one below that is the best fit for your research. If you are not sure, read the appropriate sections before making your selection.

☒ Life sciences  ☐ Behavioural & social sciences  ☐ Ecological, evolutionary & environmental sciences

For a reference copy of the document with all sections, see nature.com/documents/nr-reporting-summary-flat.pdf

# Life sciences study design

All studies must disclose on these points even when the disclosure is negative.

| | |
|---|---|
| Sample size | For all laboratory incubations there were three replicate vials, typical for these types of experiments and in accordance with the literature. The number of replicate field plots (6) was the maximum possible, given the amount of digestate available. The number of replicate buckets likewise (6 in one experiment and 8 in the second). The replicate numbers used were sufficient, as seen by the confidence intervals presented. |
| Data exclusions | In the field experiment, 112 out of 1198 flux measurements were clearly anomalous due to instrument failure (random fluctuations, very low r square, strongly negative flux estimates) and excluded from the plots and the final analysis shown in the paper. However, the statistical analyses were also performed with all measurements included, with marginal consequence for the emissions (<1%), the confidential intervals (<5%) and no consequence regarding the statistical significance of treatment effects. The identification of anomalous measurements were done blinded (treatment was not identified by the person judging the data)<br>No measurements were excluded from the field bucket experiments. |
| Replication | Phenotyping experiments carried out in the laboratory were repeated at least twice, with triplicate vials each time. For field experiments, the effect of CB-01 on N2O emissions was shown repeatedly by testing it in three independent experiments: one long-term bucket experiment with several re-fertilization events, one bucket experiment with different soil types and one long-term field plot experiment. |
| Randomization | Field plots were pairwise plots, with and without Cloacibacterium sp. CB-01, randomly placed within the field site. |
| Blinding | Blinding was used when identifying anomalous measurements in the field experiments. |

# Behavioural & social sciences study design

All studies must disclose on these points even when the disclosure is negative.

| | |
|---|---|
| Study description | *Briefly describe the study type including whether data are quantitative, qualitative, or mixed-methods (e.g. qualitative cross-sectional, quantitative experimental, mixed-methods case study).* |
| Research sample | *State the research sample (e.g. Harvard university undergraduates, villagers in rural India) and provide relevant demographic information (e.g. age, sex) and indicate whether the sample is representative. Provide a rationale for the study sample chosen. For studies involving existing datasets, please describe the dataset and source.* |
| Sampling strategy | *Describe the sampling procedure (e.g. random, snowball, stratified, convenience). Describe the statistical methods that were used to predetermine sample size OR if no sample-size calculation was performed, describe how sample sizes were chosen and provide a rationale for why these sample sizes are sufficient. For qualitative data, please indicate whether data saturation was considered, and what criteria were used to decide that no further sampling was needed.* |
| Data collection | *Provide details about the data collection procedure, including the instruments or devices used to record the data (e.g. pen and paper, computer, eye tracker, video or audio equipment) whether anyone was present besides the participant(s) and the researcher, and whether the researcher was blind to experimental condition and/or the study hypothesis during data collection.* |
| Timing | *Indicate the start and stop dates of data collection. If there is a gap between collection periods, state the dates for each sample cohort.* |
| Data exclusions | *If no data were excluded from the analyses, state so OR if data were excluded, provide the exact number of exclusions and the rationale behind them, indicating whether exclusion criteria were pre-established.* |
| Non-participation | *State how many participants dropped out/declined participation and the reason(s) given OR provide response rate OR state that no participants dropped out/declined participation.* |
| Randomization | *If participants were not allocated into experimental groups, state so OR describe how participants were allocated to groups, and if allocation was not random, describe how covariates were controlled.* |

# Ecological, evolutionary & environmental sciences study design

All studies must disclose on these points even when the disclosure is negative.

| | |
|---|---|
| Study description | *Briefly describe the study. For quantitative data include treatment factors and interactions, design structure (e.g. factorial, nested, hierarchical), nature and number of experimental units and replicates.* |
| Research sample | *Describe the research sample (e.g. a group of tagged Passer domesticus, all Stenocereus thurberi within Organ Pipe Cactus National Monument), and provide a rationale for the sample choice. When relevant, describe the organism taxa, source, sex, age range and any manipulations. State what population the sample is meant to represent when applicable. For studies involving existing datasets, describe the data and its source.* |
| Sampling strategy | *Note the sampling procedure. Describe the statistical methods that were used to predetermine sample size OR if no sample-size calculation was performed, describe how sample sizes were chosen and provide a rationale for why these sample sizes are sufficient.* |
| Data collection | *Describe the data collection procedure, including who recorded the data and how.* |
| Timing and spatial scale | *Indicate the start and stop dates of data collection, noting the frequency and periodicity of sampling and providing a rationale for these choices. If there is a gap between collection periods, state the dates for each sample cohort. Specify the spatial scale from which the data are taken* |
| Data exclusions | *If no data were excluded from the analyses, state so OR if data were excluded, describe the exclusions and the rationale behind them, indicating whether exclusion criteria were pre-established.* |
| Reproducibility | *Describe the measures taken to verify the reproducibility of experimental findings. For each experiment, note whether any attempts to repeat the experiment failed OR state that all attempts to repeat the experiment were successful.* |
| Randomization | *Describe how samples/organisms/participants were allocated into groups. If allocation was not random, describe how covariates were controlled. If this is not relevant to your study, explain why.* |
| Blinding | *Describe the extent of blinding used during data acquisition and analysis. If blinding was not possible, describe why OR explain why blinding was not relevant to your study.* |

Did the study involve field work? ☐ Yes ☐ No

## Field work, collection and transport

| | |
|---|---|
| Field conditions | *Describe the study conditions for field work, providing relevant parameters (e.g. temperature, rainfall).* |
| Location | *State the location of the sampling or experiment, providing relevant parameters (e.g. latitude and longitude, elevation, water depth).* |
| Access & import/export | *Describe the efforts you have made to access habitats and to collect and import/export your samples in a responsible manner and in compliance with local, national and international laws, noting any permits that were obtained (give the name of the issuing authority, the date of issue, and any identifying information).* |
| Disturbance | *Describe any disturbance caused by the study and how it was minimized.* |

# Reporting for specific materials, systems and methods

We require information from authors about some types of materials, experimental systems and methods used in many studies. Here, indicate whether each material, system or method listed is relevant to your study. If you are not sure if a list item applies to your research, read the appropriate section before selecting a response.

### Materials & experimental systems

| n/a | Involved in the study |
|---|---|
| ☒ | ☐ Antibodies |
| ☒ | ☐ Eukaryotic cell lines |
| ☒ | ☐ Palaeontology and archaeology |
| ☒ | ☐ Animals and other organisms |
| ☒ | ☐ Clinical data |
| ☒ | ☐ Dual use research of concern |
| ☒ | ☐ Plants |

### Methods

| n/a | Involved in the study |
|---|---|
| ☒ | ☐ ChIP-seq |
| ☒ | ☐ Flow cytometry |
| ☒ | ☐ MRI-based neuroimaging |

## Antibodies

| | |
|---|---|
| Antibodies used | *Describe all antibodies used in the study; as applicable, provide supplier name, catalog number, clone name, and lot number.* |
| Validation | *Describe the validation of each primary antibody for the species and application, noting any validation statements on the manufacturer's website, relevant citations, antibody profiles in online databases, or data provided in the manuscript.* |

## Eukaryotic cell lines

Policy information about cell lines and Sex and Gender in Research

| | |
|---|---|
| Cell line source(s) | *State the source of each cell line used and the sex of all primary cell lines and cells derived from human participants or vertebrate models.* |
| Authentication | *Describe the authentication procedures for each cell line used OR declare that none of the cell lines used were authenticated.* |
| Mycoplasma contamination | *Confirm that all cell lines tested negative for mycoplasma contamination OR describe the results of the testing for mycoplasma contamination OR declare that the cell lines were not tested for mycoplasma contamination.* |
| Commonly misidentified lines (See ICLAC register) | *Name any commonly misidentified cell lines used in the study and provide a rationale for their use.* |

## Palaeontology and Archaeology

| | |
|---|---|
| Specimen provenance | *Provide provenance information for specimens and describe permits that were obtained for the work (including the name of the issuing authority, the date of issue, and any identifying information). Permits should encompass collection and, where applicable, export.* |
| Specimen deposition | *Indicate where the specimens have been deposited to permit free access by other researchers.* |

| Dating methods | *If new dates are provided, describe how they were obtained (e.g. collection, storage, sample pretreatment and measurement), where they were obtained (i.e. lab name), the calibration program and the protocol for quality assurance OR state that no new dates are provided.* |

☐ Tick this box to confirm that the raw and calibrated dates are available in the paper or in Supplementary Information.

| Ethics oversight | *Identify the organization(s) that approved or provided guidance on the study protocol, OR state that no ethical approval or guidance was required and explain why not.* |

Note that full information on the approval of the study protocol must also be provided in the manuscript.

# Animals and other research organisms

Policy information about studies involving animals; ARRIVE guidelines recommended for reporting animal research, and Sex and Gender in Research

| Laboratory animals | *For laboratory animals, report species, strain and age OR state that the study did not involve laboratory animals.* |

| Wild animals | *Provide details on animals observed in or captured in the field; report species and age where possible. Describe how animals were caught and transported and what happened to captive animals after the study (if killed, explain why and describe method; if released, say where and when) OR state that the study did not involve wild animals.* |

| Reporting on sex | *Indicate if findings apply to only one sex; describe whether sex was considered in study design, methods used for assigning sex. Provide data disaggregated for sex where this information has been collected in the source data as appropriate; provide overall numbers in this Reporting Summary. Please state if this information has not been collected. Report sex-based analyses where performed, justify reasons for lack of sex-based analysis.* |

| Field-collected samples | *For laboratory work with field-collected samples, describe all relevant parameters such as housing, maintenance, temperature, photoperiod and end-of-experiment protocol OR state that the study did not involve samples collected from the field.* |

| Ethics oversight | *Identify the organization(s) that approved or provided guidance on the study protocol, OR state that no ethical approval or guidance was required and explain why not.* |

Note that full information on the approval of the study protocol must also be provided in the manuscript.

# Clinical data

Policy information about clinical studies
All manuscripts should comply with the ICMJE guidelines for publication of clinical research and a completed CONSORT checklist must be included with all submissions.

| Clinical trial registration | *Provide the trial registration number from ClinicalTrials.gov or an equivalent agency.* |

| Study protocol | *Note where the full trial protocol can be accessed OR if not available, explain why.* |

| Data collection | *Describe the settings and locales of data collection, noting the time periods of recruitment and data collection.* |

| Outcomes | *Describe how you pre-defined primary and secondary outcome measures and how you assessed these measures.* |

# Dual use research of concern

Policy information about dual use research of concern

## Hazards

Could the accidental, deliberate or reckless misuse of agents or technologies generated in the work, or the application of information presented in the manuscript, pose a threat to:

No | Yes
☒ ☐ Public health
☒ ☐ National security
☒ ☐ Crops and/or livestock
☒ ☐ Ecosystems
☒ ☐ Any other significant area

## Experiments of concern

Does the work involve any of these experiments of concern:

| No | Yes | |
|----|-----|---|
| ☒ | ☐ | Demonstrate how to render a vaccine ineffective |
| ☒ | ☐ | Confer resistance to therapeutically useful antibiotics or antiviral agents |
| ☒ | ☐ | Enhance the virulence of a pathogen or render a nonpathogen virulent |
| ☒ | ☐ | Increase transmissibility of a pathogen |
| ☒ | ☐ | Alter the host range of a pathogen |
| ☒ | ☐ | Enable evasion of diagnostic/detection modalities |
| ☒ | ☐ | Enable the weaponization of a biological agent or toxin |
| ☒ | ☐ | Any other potentially harmful combination of experiments and agents |

# Plants

**Seed stocks**

*Report on the source of all seed stocks or other plant material used. If applicable, state the seed stock centre and catalogue number. If plant specimens were collected from the field, describe the collection location, date and sampling procedures.*

**Novel plant genotypes**

*Describe the methods by which all novel plant genotypes were produced. This includes those generated by transgenic approaches, gene editing, chemical/radiation-based mutagenesis and hybridization. For transgenic lines, describe the transformation method, the number of independent lines analyzed and the generation upon which experiments were performed. For gene-edited lines, describe the editor used, the endogenous sequence targeted for editing, the targeting guide RNA sequence (if applicable) and how the editor was applied.*

**Authentication**

*Describe any authentication procedures for each seed stock used or novel genotype generated. Describe any experiments used to assess the effect of a mutation and, where applicable, how potential secondary effects (e.g. second site T-DNA insertions, mosiacism, off-target gene editing) were examined.*

# ChIP-seq

## Data deposition

☐ Confirm that both raw and final processed data have been deposited in a public database such as GEO.

☐ Confirm that you have deposited or provided access to graph files (e.g. BED files) for the called peaks.

**Data access links**
*May remain private before publication.*

*For "Initial submission" or "Revised version" documents, provide reviewer access links.  For your "Final submission" document, provide a link to the deposited data.*

**Files in database submission**

*Provide a list of all files available in the database submission.*

**Genome browser session**
(e.g. UCSC)

*Provide a link to an anonymized genome browser session for "Initial submission" and "Revised version" documents only, to enable peer review.  Write "no longer applicable" for "Final submission" documents.*

## Methodology

**Replicates**

*Describe the experimental replicates, specifying number, type and replicate agreement.*

**Sequencing depth**

*Describe the sequencing depth for each experiment, providing the total number of reads, uniquely mapped reads, length of reads and whether they were paired- or single-end.*

**Antibodies**

*Describe the antibodies used for the ChIP-seq experiments; as applicable, provide supplier name, catalog number, clone name, and lot number.*

**Peak calling parameters**

*Specify the command line program and parameters used for read mapping and peak calling, including the ChIP, control and index files used.*

**Data quality**

*Describe the methods used to ensure data quality in full detail, including how many peaks are at FDR 5% and above 5-fold enrichment.*

**Software**

*Describe the software used to collect and analyze the ChIP-seq data. For custom code that has been deposited into a community repository, provide accession details.*

# Flow Cytometry

## Plots

Confirm that:

☐ The axis labels state the marker and fluorochrome used (e.g. CD4-FITC).

☐ The axis scales are clearly visible. Include numbers along axes only for bottom left plot of group (a 'group' is an analysis of identical markers).

☐ All plots are contour plots with outliers or pseudocolor plots.

☐ A numerical value for number of cells or percentage (with statistics) is provided.

## Methodology

| | |
|---|---|
| Sample preparation | *Describe the sample preparation, detailing the biological source of the cells and any tissue processing steps used.* |
| Instrument | *Identify the instrument used for data collection, specifying make and model number.* |
| Software | *Describe the software used to collect and analyze the flow cytometry data. For custom code that has been deposited into a community repository, provide accession details.* |
| Cell population abundance | *Describe the abundance of the relevant cell populations within post-sort fractions, providing details on the purity of the samples and how it was determined.* |
| Gating strategy | *Describe the gating strategy used for all relevant experiments, specifying the preliminary FSC/SSC gates of the starting cell population, indicating where boundaries between "positive" and "negative" staining cell populations are defined.* |

☐ Tick this box to confirm that a figure exemplifying the gating strategy is provided in the Supplementary Information.

# Magnetic resonance imaging

## Experimental design

| | |
|---|---|
| Design type | *Indicate task or resting state; event-related or block design.* |
| Design specifications | *Specify the number of blocks, trials or experimental units per session and/or subject, and specify the length of each trial or block (if trials are blocked) and interval between trials.* |
| Behavioral performance measures | *State number and/or type of variables recorded (e.g. correct button press, response time) and what statistics were used to establish that the subjects were performing the task as expected (e.g. mean, range, and/or standard deviation across subjects).* |

## Acquisition

| | |
|---|---|
| Imaging type(s) | *Specify: functional, structural, diffusion, perfusion.* |
| Field strength | *Specify in Tesla* |
| Sequence & imaging parameters | *Specify the pulse sequence type (gradient echo, spin echo, etc.), imaging type (EPI, spiral, etc.), field of view, matrix size, slice thickness, orientation and TE/TR/flip angle.* |
| Area of acquisition | *State whether a whole brain scan was used OR define the area of acquisition, describing how the region was determined.* |

Diffusion MRI     ☐ Used          ☐ Not used

## Preprocessing

| | |
|---|---|
| Preprocessing software | *Provide detail on software version and revision number and on specific parameters (model/functions, brain extraction, segmentation, smoothing kernel size, etc.).* |
| Normalization | *If data were normalized/standardized, describe the approach(es): specify linear or non-linear and define image types used for transformation OR indicate that data were not normalized and explain rationale for lack of normalization.* |
| Normalization template | *Describe the template used for normalization/transformation, specifying subject space or group standardized space (e.g. original Talairach, MNI305, ICBM152) OR indicate that the data were not normalized.* |
| Noise and artifact removal | *Describe your procedure(s) for artifact and structured noise removal, specifying motion parameters, tissue signals and physiological signals (heart rate, respiration).* |

| Volume censoring | *Define your software and/or method and criteria for volume censoring, and state the extent of such censoring.* |

## Statistical modeling & inference

| Model type and settings | *Specify type (mass univariate, multivariate, RSA, predictive, etc.) and describe essential details of the model at the first and second levels (e.g. fixed, random or mixed effects; drift or auto-correlation).* |

| Effect(s) tested | *Define precise effect in terms of the task or stimulus conditions instead of psychological concepts and indicate whether ANOVA or factorial designs were used.* |

Specify type of analysis: ☐ Whole brain ☐ ROI-based ☐ Both

| Statistic type for inference | *Specify voxel-wise or cluster-wise and report all relevant parameters for cluster-wise methods.* |

(See Eklund et al. 2016)

| Correction | *Describe the type of correction and how it is obtained for multiple comparisons (e.g. FWE, FDR, permutation or Monte Carlo).* |

## Models & analysis

| n/a | Involved in the study |
| --- | --- |
| ☐ | ☐ Functional and/or effective connectivity |
| ☐ | ☐ Graph analysis |
| ☐ | ☐ Multivariate modeling or predictive analysis |

| Functional and/or effective connectivity | *Report the measures of dependence used and the model details (e.g. Pearson correlation, partial correlation, mutual information).* |

| Graph analysis | *Report the dependent variable and connectivity measure, specifying weighted graph or binarized graph, subject- or group-level, and the global and/or node summaries used (e.g. clustering coefficient, efficiency, etc.).* |

| Multivariate modeling and predictive analysis | *Specify independent variables, features extraction and dimension reduction, model, training and evaluation metrics.* |

nature portfolio | reporting summary

April 2023

