## [Peer Review File · Nature]

Manuscript Title: Unlocking bacterial potential to reduce farmland N₂O emissions

Reviewer Comments & Author Rebuttals

Reviewer Reports on the Initial Version:

Referees' comments:

Referee #1 (Remarks to the Author):

Summary of results

The manuscript reports on a set of interlinked laboratory and field experiments to assess the capacity for introducing a bacteria inoculum with a high capacity to reduce N₂O to N₂, thereby reducing emissions of a potent greenhouse gas from agricultural soils. Full growing season field experiment results are used to extrapolate the potential value for abating soil N₂O emissions at EU scale, by using this (or similar) species of bacteria together with a waste-derived organic amendment, as a management practice.

Originality and significance

Both originality and significance are very high. Of ALL anthropogenic greenhouse gas emissions, one can make an argument that N₂O emissions are THE most difficult to effectively mitigate. The technology outlined in the ms suggests a substantially greater abatement potential, compared to existing interventions, and thus has enormous relevance for environmental and land use policy and management. What is particularly impressive is the degree of detail and thoroughness of the research going from the cellular level, in terms of gene expression and specific metabolic potentials, to field-scale biogeochemical investigations of gas flux rates with and without the inoculum as a function key biogeochemical and biophysical process controls, at the ecosystem scale, for different soils.

Methodology, data and statistics

As stated above, I find the methodology to be very comprehensive and well-executed and have no suggestions for improvements in the statistical analysis or presentation of the data.

From what I can tell the main conclusions are very robust and well supported. The one major suggestion for improvement, aside from some narrative text revisions to improve clarity, is to provide additional explanation for the country- and EU-scale of estimated abatement potential (based on data in Table S5. As described in the supplemental, country-scale emissions (as reported in national inventories), are shown based on a simple emission factor model of 1% of applied N to soil. The baseline emissions (2nd column totals) are not broken out by subsource category so it is unclear if they represent only ag soil N₂O, or all agricultural N₂O emissions, or ALL N₂O emissions (including industrial, transportation, etc) for the country. Using the 1% emission factor model (GAINS) implies that the emissions are from agricultural soils only. If that is the case, then the assumption of a 60% abatement by use of non-denitrifying N₂O-respiring bacteria (NNRB) inoculum for all N amendments to soil should yield a much higher reduction than the overall 31% emission reduction that is

estimated.

In case it will be useful to the authors, I include some grammatical edits and/or suggested edits to improved clarity in the attached (anonymized) pdfs

Unlocking bacterial potential: A solution to farmland N₂O-emissions

Elisabeth G Hiis¹, Silas H W Vick¹, Lars Molstad¹, Kristine Røsdal¹, Kjell Rune Jonassen², Wilfried Winiwarter^{3,4}, Lars R Bakken^{1*}

- 1) Faculty of Chemistry, Biotechnology and Food Science, Norwegian University of Life Sciences, Norway
- 2) VEAS WWTP, Bjerkåsholmen 125, 3470 Slemmestad, Norway
- 3) International Institute for Applied Systems Analysis International Institute for Applied Systems Analysis, Laxenburg, Austria
- 4) Institute of Environmental Engineering, University of Zielona Góra, Zielona Góra, Poland

Summary

Farmed soils contribute substantially to global warming by emitting N₂O, and mitigation has proven difficult. Several microbial nitrogen transformations produce N₂O, while the only biological sink for N₂O is the enzyme NosZ, catalyzing the reduction of N₂O to N₂. While strengthening the NosZ-activity in soils ~~will~~ would reduce ~~the~~ N₂O emissions, such bioengineering of the soil microbiota is commonly deemed impossible. However, we have developed a technology to achieve this, using organic waste as substrate and vector for N₂O-respiring bacteria (NRB) selected for their capacity to thrive in soil. Here we have analyzed the biokinetics of N₂O reduction by our most promising NRB, *Cloacibacterium* sp. CB-01, its survival in soil, and its effect on N₂O emissions in field experiments ~~-~~ Fertilization with waste from biogas-production, in which CB-01 had grown aerobically to $\sim 6 \cdot 10^9$ cells mL⁻¹, reduced N₂O-emissions by 50-95 %, depending on soil type. The strong and long-lasting effect of CB-01 is ascribed to its tenacity in soil, rather than its biokinetic parameters, which were inferior to other NRB-strains. Scaling up to EU level, we find that national anthropogenic N₂O-emissions ~~can~~ could be reduced by 5-20 %, and more if including other organic wastes. This opens an avenue for cost-effective reduction of N₂O-emissions for which other mitigation options are currently lacking.

Introduction

The nitrogen footprints of farming

Until the mid-20th century, ~~plant~~ crop production was severely limited by nitrogen, requiring farmers to recycle this element in a reactive form within their agroecosystems. This constraint is reflected in the agricultural treatise by Marcus Porcius Cato (234-143 BC) *De Agri Cultura*, which recommends to: "... save carefully goat, sheep, cattle, and all other dung ..." (Hooper and Ash 1934). The invention of the Haber-Bosch process (HB) in 1908 eliminated the nitrogen constraint by producing ammonium from atmospheric nitrogen. HB was a blessing, saving the world from starvation (Erismann et al., 2008), but has become a curse because it allowed farmers to use nitrogen in excess, with marginal penalties for losing nitrogen to the external environment. As a result, most agroecosystems have become nitrogen-enriched and leaky, releasing ammonia to the atmosphere and nitrate to the drains ground- and surface waters, at scales that induce eutrophication and threaten the quality and resilience of both terrestrial and aquatic ecosystems worldwide (Rockström et al. 2009, Sutton et al. 2011, Seitzinger & Phillips 2017, Gu et al 2023).

Commented [P1]: Maybe tone down a bit

Commented [P2R1]: Also penalties is a policy social issue - don't need it - used in excess, which has caused off site environmental degradation

Nitrogen fertilisation also causes emissions of the ~~climate-greenhouse~~ gas N₂O, both from agricultural soils themselves (direct emissions) and from the natural environments due to the input of reactive nitrogen lost from the farms (indirect emissions). These farming-induced emissions account for substantial shares of the escalating concentration of N₂O in the atmosphere since the industrial revolution (Davidson 2009, Reay et al. 2012, Kanter & Brownlie 2019). A comprehensive analysis of global N₂O-emissions for 2007-2016 (Tian et al. 2020), estimated ~~ds~~ that ~~total the~~ direct and indirect emissions were 2.3-5.2, and 0.6-2.1 Tg N₂O-N y⁻¹, ~~respectively, in total~~ accounting for > 50 % of the total anthropogenic N₂O emissions (= 4.1-10.3).

Commented [P3]: The largest!

Mitigation

Reducing the anthropogenic impacts on nitrogen cycling and N₂O emissions has become a major environmental challenge for the 21st century due to the severity of these issues. An obvious place to start is to improve the nitrogen use efficiency (NUE) of agroecosystems by reducing their losses of ammonia and nitrate (Sutton et al. 2011). This can be achieved by policy instruments to induce shifts in existing- and implementation of emerging farming technologies (Vatn 2015, Davidson et al. 2014, Zhang et al. 2015, Gu et al. 2023).

Commented [P4]: Not obvious how this happens! Needs some explanation - needs define

Commented [P5R4]: Need 1 sentence on difference of denitrification and nitrification emissions

While improving NUE can reduce emissions, deliberately manipulating the soil microbes involved holds even greater potential for achieving substantial reductions. Most of the N₂O emitted from soils is produced by denitrifying bacteria (DB), and denitrifying fungi (DF), and a minor fraction is produced by ammonia-oxidizing archaea (AOA) and -bacteria (AOB) (Bakken and Frostegård 2020). While AOA&AOB and DF are net sources of N₂O because they lack the enzyme N₂O-reductase, DB can be both sinks and sources: N₂O is a free intermediate in their stepwise reduction of nitrate to molecular nitrogen, NO₃⁻ → NO₂⁻ → NO → N₂O → N₂, catalyzed by enzymes encoded by the genes *nar/nap*, *nirK/S*, *norB/Q* and *nosZ*, respectively (Zumft 1997). The organisms use this pathway to sustain their respiratory metabolism under hypoxic/anoxic conditions. DB are extremely diverse both regarding their catabolic potential, their regulation of denitrification (Bakken et al. 2012, Lycus et al. 2017), and their denitrification gene sets: a substantial share of DB in soils have truncated denitrification pathways, lacking 1-3 of the 4 genes coding for the complete pathway (Lycus et al. 2017, Pessi et al. 2020). This has been taken to suggest that denitrification is essentially "modular", i.e. that each step of the pathway is catalyzed by a separate group of organisms, rather than by organisms performing all the steps of the pathway (Roco et al. 2017). The truth is probably a bit of both (Gowda et al. 2021, Hallin et al. 2018).

Being the only sink for N₂O in soils, the enzyme N₂O-reductase (NosZ) has been the target for recent attempts to mitigate N₂O emissions from soils. An intervention that strengthens this sink will lower the N₂O/N₂ product ratio of denitrification and hence reduce the propensity of the soil to emit N₂O into the atmosphere (Bakken and Frostegård 2020, Klimasmith and Kent, 2022). This can be achieved by liming to increase the soil pH: the synthesis of functional *nosZ* is enhanced by pushing the soil pH towards the upper end of the normal pH-range of farmed soils (pH 5-7) (Liu et al. 2014). As a result, liming acidified soils will reduce their N₂O emissions by 10-20 %, albeit with a next-to neutral climate effect due to the CO₂ emission induced by lime application (Henault et al. 2019, Wang et al. 2021).

N₂O-respiring bacteria

Increasing the soils' population of N₂O-respiring bacteria (*NRB*, BOX 1) could decrease the emission of N₂O (Simon 2021). *NRB* with a complete denitrification pathway can be net sinks of N₂O if their denitrification regulatory networks secure earlier and/or stronger expression of *NosZ* than of the other denitrification enzymes (Lycus et al. 2018, Jonassen et al. 2022a), or if their electron flow is channeled preferentially to *NosZ* (Gao et al. 2021). Their effect as N₂O-sinks is plausibly conditional, however, since regulation of their anaerobic respiratory pathway can be influenced by environmental conditions. In contrast, bacteria equipped only with the gene for N₂O-reduction, will be unconditional net sinks for N₂O whenever deprived of O₂ (Shan et al. 2021). In the following, we will call them *Non-denitrifying N₂O-respiring Bacteria*, ***NNRB*** because they are unable to denitrify (BOX 1).

We know too little about the ecology and physiology of *NNRB* to selectively enhance their growth *in situ* (Hallin et al. 2018), but their potential as agents to reduce N₂O-emissions from soils is indisputable, as demonstrated by laboratory incubations of soils amended with *NNRB* grown *ex situ* (Domeignoz-Horta et al. 2016). Recently, Jonassen et al. (2022a) suggested that such soil amendment can be done inexpensively on large scale, by using waste from biogas reactors (digestates), destined for soils as organic fertilizers, both as substrate and vector for *NRB*/*NNRB*. They successfully enriched and isolated *NRB* with a strong preference for N₂O, which could grow aerobically to high cell densities in digestates, and showed that amending soils with *NRB*-enriched digestates lowered the N₂O/N₂ product ratio of denitrification. Their isolates were not ideal, however, because they had genes for the entire denitrification pathway, and their catabolic capacities were streamlined for growth in digestate, not soil. In a follow up study, Jonassen et al. (2022b) designed a dual substrate enrichment strategy, switching between sterilized digestate and soil as substrates, to deliberately select for *NRB*/*NNRB* with a broader catabolic capacity and physiochemical tolerance. The enrichments became dominated by strains classified as *Cloacibacterium* (based on 16S amplicon sequencing), and the isolated strain *Cloacibacterium* sp. CB-01 was deemed promising: It carries the genes for NO- and N₂O-reduction but lacks the genes for reduction of NO₃⁻ and NO₂⁻, thus qualifying as an *NNRB* (BOX 1). A subsequent meta-omics analysis

BOX 1. Bacterial sinks for N₂O: *NRB* and *NNRB*

Bacteria with a complete denitrification pathway sustain their anaerobic respiration by stepwise reduction of NO₃⁻ to N₂, catalyzed by four reductases, producing NO₂⁻, NO and N₂O as free intermediates:

Many bacteria have a truncated pathway, lacking 1-3 of the reductases, with consequences for their role as sources or sinks for N₂O.

Terminology:

NRB = N₂O-Respiring Bacteria = all bacteria with *NosZ*. *NRB* equipped with *Nir* and *Nor* are either sinks or sources for N₂O, depending on regulation.

NNRB = Non-denitrifying N₂O-respiring bacteria = *NRB* lacking the genes for denitrification *sensu stricto*, i.e. *NirK/S*. *NNRB* are net sinks for N₂O.

N₂O-reductase types:

There are two known versions of this copper enzyme: *NosZI* and *NosZII*.

Electrons are transferred to *NosZII* via another pathway than *NosZI*, apparently generating more proton-motive force per electron (Yoon et al. 2016, Hein et al. 2017).

NosZII appears to have a higher affinity for N₂O (Yoon et al. 2016).

of the enrichments and the genome of CB-01 suggested that surface attachment and utilization of complex polysaccharides contributed to its fitness in soil (Vick et al. 2023).

Here, we have evaluated CB-01's ability to reduce N₂O emission from soil, when vectored by digestate. We examined several regulatory and enzyme kinetic traits to assess its inherent strength as an N₂O sink. Then we tested its capacity in "real life" by conducting field experiments where soils were fertilized with digestate in which CB-01 had been grown to high cell density. Lastly, we assessed the potential of this technology for reducing N₂O emissions across the European Union.

Results and discussion

Evaluating the respiratory and regulatory phenotype of *Cloacibacterium* sp. CB-01

The genome of CB-01 contains *nosZII* but lacks any genes coding for dissimilatory reduction of NO₃⁻ and NO₂⁻, predicting a phenotype able to respire N₂O but neither NO₃⁻ or NO₂⁻, which was confirmed experimentally: in response to oxygen depletion, CB-01 reduced N₂O to N₂, but was unable to produce N₂O from NO₂⁻ (Jonassen et al. 2022b). The fact that it has *norBC*, coding for NO-reductase means that it could produce N₂O from NO, but the NO kinetics indicates minor NO-reductase activity (Jonassen et al. 2022b Suppl Item 8 B&C). This qualifies CB-01 as an NNRB (see BOX 1), and the laboratory incubation of soils fertilized with digestates containing CB-01 produced marginal amounts of N₂O (Jonassen et al. 2022b).

Commented [P6]: Fix the grammar and punctuation

The capacity of a strain to reduce N₂O-emissions is commonly judged by a set of biokinetic parameters (Simon 2021), and we decided to investigate these for CB-01, for comparison with other strains.

Growth yield: Based on the bioenergetics and charge separation for aerobic and anaerobic respiration of canonical denitrifying organisms₇ having *NosZI* (Box 1), the growth yield in terms of g cell dry weight per mol electrons (Y_{e-N_2O}) is ~60 % of that for aerobic growth (Y_{e-O_2}) (van Spanning et al. 2007). For CB-01, which has *NosZII*, Y_{e-N_2O} was 85 % of Y_{e-O_2} (Fig S1) which lends support to the claim that electron flow via *NosZII* conserves more energy (by charge separation) than via *NosZI* (Yoon et al. 2016, Hein et al. 2017).

Cell-specific respiration- and growth-rates: Measured aerobic and anaerobic respiration rates during unrestricted growth in nutrient broth at 23 °C were used to estimate maximum growth rates, μ_{max} , by nonlinear regression (Fig S2), and the maximum rate of electron flow per cell to O₂ and N₂O were calculated based on the measured growth yields ($V_{max} = \mu_{max}/Y$). The estimates are $\mu_{max-O_2} = 0.29 \text{ h}^{-1}$ (stdev = 0.006), $\mu_{max-N_2O} = 0.11 \text{ h}^{-1}$ (stdev = 0.001), $V_{maxO_2} = 0.72 \text{ fmol O}_2 \text{ cell}^{-1} \text{ h}^{-1}$, $V_{maxN_2O} = 0.66 \text{ fmol N}_2\text{O cell}^{-1} \text{ h}^{-1}$. In terms of electron flow rates per cell, we get $V_{e,maxO_2} = 2.9 \text{ fmol e}^- \text{ to O}_2 \text{ cell}^{-1} \text{ h}^{-1}$, $V_{e,maxN_2O} = 1.3 \text{ fmol e}^- \text{ to N}_2\text{O cell}^{-1} \text{ h}^{-1}$. This shows that CB-01 slows down its respiratory metabolism by ~50 % when switching from aerobic to anaerobic respiration.

Affinity for O₂ and N₂O. It is widely accepted that an organism's ability to effectively mitigate N₂O emissions is strongly influenced by its affinity to N₂O. We determined the apparent half saturation constant for O₂ and N₂O reduction in CB-01 by nonlinear regression of rates per cell versus concentrations of the two gases in the liquid, and found $k_{mO_2} = 0.9 \text{ }\mu\text{M O}_2$ (SE = 0.27) and $k_{mN_2O} = 12.9 \text{ }\mu\text{M N}_2\text{O}$ (SE = 1.2) (Fig S6). The relatively low k_{mO_2} was expected since the genome of CB-01 contains genes coding for *cbb3*-type high affinity cytochrome C oxidases (Vick et al. 2023).

Comparing the N₂O sink-strength. To compare CB-01 with other organisms as a sink for N₂O in soil, we have summarized the biokinetic parameters for various N₂O-respiring organisms by plotting their “catalytic efficiency” (V_{max}/K_m) against their V_{max} on a cell dry-weight basis (Fig 1B). This suggests that CB-01 is far from being the best among N₂O-respiring organisms: It is on par with the average of others with respect to V_{max} , which is a measure of the N₂O sink strength at high N₂O concentrations ($\gg k_m=12.9 \mu\text{M N}_2\text{O} \sim 280 \text{ ppmv}$ in the gas phase at 15 °C), while it scores poorly at low N₂O concentrations (V_{max}/K_m for CB-01 is only 3 % of the average for the others). The apparent bet-hedging (Fig 1A) adds to its inferiority, unless bet-hedging is an “artifact” of the culturing in stirred batches: the fraction of cells expressing NosZ increased with increasing cell density in these batches, and could conceivably reach 1 for cells if clustered in the soil.

Figure 1 The biokinetics of N₂O reduction, CB-01 versus other strains. As judged by kinetics of N₂O respiration in pure culture, CB-01 scores strikingly low compared to other N₂O-respiring organisms as a sink for N₂O: the kinetics of N₂O-respiration in response to O₂-depletion indicates bet-hedging, i.e. that only a fraction (F_{NosZ}) of the cells express NosZ and start growing by N₂O-respiration after O₂-depletion. Panel A illustrates the phenomenon for a single vial: Measured O₂ and N₂O (triangles), and simulated (continuous lines) values, using a simplified version of the bet-hedging model of Hassan et al. (2014), with $F_{NosZ} = 0.03$. The green line shows the simulated cell density, and the dashed black line shows simulated N₂O for $F_{NosZ} = 1$. The insert shows measured and simulated total electron flow in the vial. Panel B provides a condensed comparison of CB-01 with other N₂O-respiring organisms regarding its capacity to scavenge N₂O. Here we have plotted V_{max}/K_m against V_{max} (mmol N₂O g⁻¹ cell dryweight h⁻¹) for CB-01 and a range of other organisms with NosZI and NosZII, as measured by others (See Table S1 for details and citations). The comparison shows that CB-01 is close to the average with respect to V_{max} (37 % of the average of others), while its V_{max}/K_m - ratio is very low (~3 % of the average of others) due to the low apparent affinity for N₂O ($K_m = 13 \mu\text{M N}_2\text{O}$).

The effects of CB-01, vectored by digestate, on N₂O emissions

CB-01 was found to grow exponentially by aerobic respiration in autoclaved digestate, reaching a cell density of 10⁹ cells mL⁻¹ after 20 h. At this point ~1 % of the organic C in the digestate had been consumed, and growth rate declined gradually, plausibly due to depletion of the most easily available substrate components (Fig S7), reaching a final density of ~6*10⁹ cells mL⁻¹ after 2 days, as judged by oxygen consumption and qPCR quantification of CB-01 cells (Fig S7, Table S2).

To assess the capacity of such CB-01-enriched digestates to reduce the N₂O emission from soils under realistic agronomic situations, we conducted three outdoor experiments where N₂O emissions

were measured frequently, using a field flux robot equipped with a tunable diode laser, allowing N₂O emissions to be measured by 3 minutes enclosures. For details, see **Supplementary 2C**.

The first field experiment (field bucket experiment, described in Supplementary 2D) demonstrated (Fig 2) that the initial peak of N₂O-flux induced by the fertilization with digestate was practically eliminated by CB-01, and that CB-01 continued to have a strong effect throughout: a second peak in N₂O emission induced by precipitation (day 12) was reduced by 51 %, and the later emission peaks induced by re-fertilization with digestate without CB-01 (indicated by arrows) were reduced by 31, 67, and 46 %.

Commented [P7]: Seems improper use of comma here

Figure 2 CB-01 effects on N₂O emission from a clay loam soil pH 6.7. The panel shows N₂O-flux from buckets with soil throughout 90 days after fertilization (14.07.2021) with digestate (11 L m⁻²) in which the NNRB strain *Cloacibacterium* sp. CB-01 had been grown to ~6*10⁹ cells mL⁻¹, quantified by qPCR with primers specific for CB-01 (**Supplementary 2F**). Control buckets were fertilized with the same digestate in which CB-01 had been killed by heat (70 °C for 2 hours). The buckets were sown with ryegrass, the soil moisture content was sustained by daily water additions during the first 10 days. Buckets were re-fertilized with a lower dose of autoclaved and pH-adjusted digestate without CB-01 (4.6 L m⁻²) after 19, 33 and 89 days. The top panel shows N₂O-flux measured by dynamic chamber method (Cowan et al. 2014) with 3 minutes enclosure times, operated by a field robot (**Supplementary 2C**). The emissions are shown as single dots for each enclosure, and with a floating average for each treatment (continuous lines, n = 8 replicate buckets for each treatment), calculated by a Gaussian Kernel smoother). The lower panels show soil temperature (at 0-5.5 cm depth) and water filled pore-space (WFPS). The fluxes show clear diurnal fluctuations, driven by temperature, and transient peaks in response to a rain event (day 12) and in response to re-fertilization (marked by arrows). The % reduction of N₂O emissions (cumulated flux) by CB-01 were calculated for selected periods, shown in green bars with 95% confidential intervals. The additional control buckets receiving water instead of digestates emitted negligible amounts of N₂O (result not shown).

Commented [P8]: Scientific name of the grass

Given the number of CB-01 cells added with the digestate (6.6×10^{13} cells m^{-2} soil surface), and the $V_{max} = 0.6$ $fmol\ N_2O\ cell^{-1}\ h^{-1}$ (Fig S2), the potential N_2O -reduction rate, if all the added CB-01 cells were respiring N_2O at maximum rate, is $1.1\ g\ N_2O-N\ m^{-2}\ h^{-1}$. The peak N_2O -flux 1-2 days after fertilization was reduced by $\sim 85\ mg\ N_2O-N\ m^{-2}\ h^{-1}$, which is $\sim 8\%$ of the estimated potential. For the subsequent peaks of N_2O -flux, the apparent N_2O -respiration by CB-01 (i.e. the reduction of the flux) was $\leq 4\ mg\ N_2O-N\ m^{-2}\ h^{-1}$, which is $\leq 0.36\%$ of the initial potential. This decline in apparent N_2O -respiration by CB-01 was plausibly a result of two factors: a gradually declining rate of N_2O provision by the indigenous microbiome, and a gradually declining number of CB-01 cells.

Commented [P9]: Confusing, rephrase

One would expect that the effect of CB-01 as an N_2O sink would be marginal in periods with low emissions: Low emissions are due to low waterfilled pore-space (WFPS) (i.e. drained/dry soil), low respiration rate (limited by available organic C-substrates), or both, resulting in marginal hypoxic/anoxic volumes within the soil matrix (Ball, 2013). Under such conditions, the primary source of N_2O emission could be nitrification (Nadeem et al. 2020), while CB-01 as an N_2O -sink would be confined to the remaining hypoxic microsites. Inspections of the relationship between the effect of CB-01 and the N_2O emissions in the control soil (i.e with dead CB-01) lend some support to this: although CB-01 reduced the emissions even for periods with modest emissions, the effect was clearly strongest in periods with high emissions (Fig 3).

Figure 3 Inspecting the contingent effect of CB-01. Emissions from soil fertilized with digestate containing live *Cloacibacterium* sp. CB-01 plotted against the emissions from soil fertilized with digestate containing dead CB-01 cells (same data as in Figure 2). Panel A-C show the results for the low, intermediate and high emission ranges, respectively. Panel D is a log scaled plot of the ratio between emissions from soil with live and dead CB-01 plotted against the emission from soil fertilized with digestate containing dead CB-01.

We hypothesized that the capacity of CB-01 to reduce N_2O emissions could be influenced by soil type. Soil pH is plausibly crucial because the synthesis of functional N_2O -reductase is increasingly impeded by declining pH within the range 4-7, both in CB-01 (Jonassen et al. 2022b) and most other NRB (Bakken and Frostegård 2020). Soil organic carbon content (SOC) could also have an impact. This is because the abundance of CB-01 relative to the abundance of indigenous N_2O -producing bacteria would be inversely related to SOC, since the abundance of indigenous bacteria in soil is directly related to SOC (Taylor et al. 2002). To explore this, we replicated the bucket experiment (Fig 2), but with four different soils spanning a range of pH-levels and including a soil with very high organic carbon content.

Figure 4 Reduction of N₂O emissions in different soils in field buckets. The panels show the measured emission after application of digestates with and without CB-01 to four different soils (17.09.2021). The soils' content of organic C were 15.8 (soil O), 3.21 (soil N), 0.75 (soil S) and 3.23 (soil L) % of dry weight (Table S2). The pH(CaCl₂) prior to fertilization with digestate are given in the panels. The emissions are shown as single dots for each enclosure, and with a floating average for each treatment (continuous lines, n = 6 replicate buckets for each treatment) as in Fig 2. The % reduction of N₂O emissions (cumulated flux) by CB-01 were calculated for selected periods, shown in green bars with 95% confidential intervals.

The emissions were low compared to the first experiment, plausibly due to lower temperatures (September versus July), however CB-01 significantly reduced the emissions from all four soils. The strong effect in the acidic sandy silt soil (pH = 4.15) was a surprise, until we measured the pH of the soils after amendment with digestate: the incorporation of digestate in this soil increased the pH(CaCl₂) of the sandy silt soil by more than one pH unit (Table S3), reflecting its weak buffer capacity. Most probably, the CB-01 embedded in the digestate experienced even higher pH (pH of the digestate was 7.3). The results for the three clay loam soils show a stronger effect of soil-pH: CB-

01 had a clearly stronger effect in the neutral pH clay loam (pH = 6.7) than in the two more acidic clay loams (L: pH = 4.5, O: pH = 5.26).

Finally, we scaled up to a field plot experiment, fertilizing 0.5 m² plots with digestate with live and dead CB-01, carved into the upper 10 cm layer of the soil as in the bucket experiments. The experiment was conducted on field plots that had been limed with 2.3 kg m⁻¹ of dolomite in 2014, with an average pH(CaCl₂) = 6.13 (stdev = 0.10). The high emissions during the first 4 days (Fig 5A) show diurnal variations, peaking when the soil temperature reach their maximum, and a substantial effect of CB-01. Subsequent emissions, measured at low frequency throughout 280 days, were much lower, and the effect of CB-01 was not statistically significant, albeit with a wide confidential interval.

Commented [P10]: Mixed into? - meaning not clear

Figure 5 Reduction of N₂O emissions in field plots. The 0.5 m² field plots with clay loam pH = 6.13 (soil I, table S2) were fertilized by mixing digestate into the upper 10 cm (20.08.2022), with live and dead CB-01 as in previous experiments (n = 6 replicate plots for each treatment). The main panel shows emissions throughout 290 days, and the insert shows emissions during the first 10 days. The % reduction of N₂O emissions (cumulated flux) by CB-01 for the periods 0-10, and 10-290 days are shown in green bars with 95% confidential intervals.

Survival of CB-01 in soil, and its effect on the indigenous microbes

Soil microbiome engineering by inoculation/augmentation is an emerging field, promising new possibilities in enhancing agricultural efficiency and sustainability (Lawson et al. 2019). It is challenging, however, because inoculants are invariably found to die out rapidly, plausibly due to a multitude of abiotic and biotic barriers impeding establishment (Kaminsky et al. 2019). CB-01 was obtained through a dual substrate enrichment technique aimed at isolating organisms capable of withstanding the abiotic challenges of soil (Jonassen in 2022b). However, this selection process did not account for the biotic barriers organisms may encounter in soils, such as competition for resources, antagonism, and predation, as highlighted by Albright et al. (2022).

To assess the ability of CB-01 to survive in soil, we used qPCR with specific primers (**Supplementary 2G**) to measure the abundance of CB-01 genomes in soil throughout the long-term field bucket

experiment (Fig 2), and throughout a laboratory incubation of soil amended with digestate with CB-01 (Fig 6). During the laboratory incubation, there was a fast first-order reduction in abundance during day 3-7, and a much slower first order reduction thereafter. In contrast, the abundance was sustained at a high level throughout 90 days in the field buckets, albeit gradually declining. The sustained CB-01 population in the bucket experiment explains why the effect on the N₂O-emission was sustained (Fig 2).

But the glaring discrepancy between the field and the lab experiments demands a scrutiny. In the field bucket experiment, digestate (not inoculated with CB-01) were applied three times during the course of the experiment, with soil sampling for quantification of CB-01 abundance conducted two days after each incorporation. Since digestate is a suitable substrate for CB-01, growth of CB-01 in response to each dose could contribute to the sustained population.

Another factor of importance could be that protozoal grazing was plausibly more intense in the laboratory incubations than in the field experiment, due to the higher soil moisture content at the time of CB-01 incorporation: In the lab experiment, the digestate with CB-01 was dripped onto soil which was already very wet (0.53 mL g⁻¹ soil dry weight) and retained this high soil moisture throughout. In the field bucket experiment, CB-01-digestate was harrowed into relatively dry soil (0.34 mL g⁻¹ soil dry weight), and it remained modestly moist throughout (Fig 2). There is ample evidence that low soil moisture protects a bacterial inoculum against protozoal grazing, ascribed to increasing tortuosity, and localization of bacteria in small pores that are inaccessible to the protozoa (van Veen et al. 1997). While we recognize that this is a speculative explanation, it warrants further experimental investigation due to the potential practical implications.

Figure 6 Survival of CB-01 in soil. The abundance of CB-01 was assessed by qPCR as described in Supplementary 2G. The panel shows the genome abundance in the long-term field bucket experiment (Fig 2) and in the laboratory incubation experiment. For the field bucket experiment, where additional digestate was incorporated 2 days before each soil sampling for qPCR. A single dot represents an individual soil sample (n = 8), and the line is the fitted exponential function $N_t = N_0 * e^{-d*t}$ where N_t is the abundance at time = t , and d is the apparent first order death-rate (estimated half-life $T_{1/2} = \ln(2)/d$). For the laboratory incubation, three phases can be recognized: an initial apparent growth during the first 2-3 days, followed by a rapid first-order decline during the subsequent 4-5 days, and a slow first order decline thereafter. Of note, the estimated decline in CB-01 genome abundance in the field plot experiment, lasting 280 days indicated very similar average first order death rates (0.028 d⁻¹, half life =25 days, Supplementary 4).

Considering that the soil was heavily inoculated with CB-01 (> 10⁹ cells g⁻¹) and that the population was sustained > 10⁸ cells g⁻¹ throughout 90 days in the field buckets, there is a legitimate concern that this could affect the indigenous microbiota (Mallon et al. 2018). We investigated this by analysis of 16S rRNA gene amplicons, excluding the OTU that circumscribed CB-01 (Supplementary 3), and found that the digestate as such had a transient impact, but we were unable to discern any consistent difference between the treatments with live versus dead CB-01, which both converged towards the composition of pristine soil (Fig S12).

Extrapolating national emission reductions

To assess the potential emission reductions by NNRB as compared with other available techniques such as optimized N fertilization and nitrification inhibitors, we estimated emissions for Europe 2030 with the ~~gains~~-GAINS model (Winiwarter et al. 2018, Amann et al. 2011).

Consistent with using a uniform emission factor in GAINS (from IPCC, 2006) of 1% of N applied to be emitted as N₂O, we also assume a uniform factor for emission reductions. From the experiments we conclude that 60% of emission reductions due to NNRB may be considered a conservative estimate. In **Table S5**, emission reductions are shown by EU country for a 2030 scenario if emissions from the application of liquid manure only is reduced by 60%. All other anthropogenic emissions have been left unchanged. Under these assumptions, the total anthropogenic N₂O emissions from Europe decrease by 2.7% due to NRB being introduced and applied to all liquid manure systems. This figure is higher in countries that have a high share of liquid manure systems in their agriculture, hence it increases to 4.0% for EU27 (27 EU member countries).

Ongoing work explores the possibility to extend the technology by growing NNRB in all types of organic wastes used to fertilize soils, and by combining the application of mineral N-fertilizers with incorporation of NNRB-amended organic wastes. This requires new strains, technologies and investments, but with a great potential, reducing EU27 agricultural emissions by a third (31%).

It needs to be pointed out that an emission reduction of 60% as derived here for NNRB is much larger than emission reductions typically reported for N₂O abatement measures. E.g., GAINS assumes nitrification inhibitors to be able to reduce emissions by as much as 38%, and high-tech mechanical fertilizer-saving technologies (“variable rate application”) to be able to save 24% of the emissions only (Winiwarter et al. 2018).

Commented [P11]: Where is this calculated?

Commented [P12]: grammare

Future development

This study presents the first proof of concept, demonstrating a feasible utilization of non-denitrifying N₂O-respiring bacteria (NNRB) to curb N₂O emissions from farmland. By using organic waste as substrates and vectors, massive soil inoculation is achieved, which can secure reduced N₂O-emissions throughout an entire growth season, despite a gradually declining NNRB-abundance. To ensure the robustness and versatility of this biotechnology, we will need an ensemble of new NNRB strains, capable of thriving in waste materials beyond digestates. New NNRB strains will probably vary regarding their ability to tolerate abiotic and biotic stress factors present in the soil. The dual substrate enrichment technique (Jonassen et al. 2022b) selects for strains tolerant of abiotic, but not biotic stress. Consequently, innovative techniques are necessary for selecting strains that tolerate the biotic stress.

Methods

Determination of the respiratory phenotype was done by batch incubation in the robotized incubation system (**Supplementary 2A**). The same system was used to assess growth of CB-01 in digestate (**Supplementary 2B**), further elaborated by quantifying the abundance by qPCR with specific 16S primers (**Supplementary 2G**).

The effect of CB-01-enriched digestate on N₂O emission was measured in field bucket experiments (**Supplementary 2D**) and field plot experiments (**Supplementary 2E**), using a field robot

(Supplementary 2C). Flux calculations and statistics are described in Supplementary 2F. The extrapolation to national emission reductions is described in Supplementary 2I.

Survival of CB-01 in the soils in the field bucket experiment was assessed by qPCR with specific 16S primers (Supplementary 2G), which was also used to assess the survival in soil incubated in the laboratory (Supplementary 2H). The effect of CB-01 on the indigenous microbiota was assessed by analysis of 16S amplicons, general 16S primers (Supplementary 3).

Authors contributions

EGH, LRB and KRJ designed the experiments and analyzed data. LRB, EG, SHWW, LM, KR, KRJ designed and conducted phenotyping experiments, EGH and LM designed and conducted the field experiments, EGH, KRJ and KR traced *Cloacibacterium* in soils, WW extrapolated national emission reductions.

References

1. Albright MBN, Louca S, Winkler DE, Feeser KL, Haig SJ, Whiteson KL, Emerson JB, Dunbar J (2022) Solutions in microbiome engineering: prioritizing barriers to organism establishment. The ISME Journal 16:331–338; <https://doi.org/10.1038/s41396-021-01088-5>
2. Amann M, Bertok I, Borken-Kleefeld J, Cofala J, Heyes C, Höglund-Isaksson L, Klimont Z, Nguyen B, Posch M, Rafaj P, Sandler R, Schöpp W, Wagner F, Winiwarter W (2011) Cost-effective control of air quality and greenhouse gases in Europe: Modeling and policy applications. Env Modelling and Software 26:1489-1501. <https://doi.org/10.1016/j.envsoft.2011.07.012>
3. Bakken LR, Bergaust L, Liu B, Frostegård Å (2012) Regulation of denitrification at the cellular level: a clue to the understanding of N₂O emissions from soils. Phil. Trans. R. Soc. B 2012 367, 1226-1234
4. Bakken LR, Frostegård Å (2020) Emerging options for mitigating N₂O emissions from food production by manipulating the soil microbiota. Current Opinion in Environmental Sustainability 47:89-94.
5. Bergaust L, Bakken LR, Frostegård Å (2011) Denitrification regulatory phenotype, a new term for the characterization of denitrifying bacteria. Biochem. Soc. Trans. 39, 207–212; doi:10.1042/BST0390207
6. Ball B (2013) Soil structure and greenhouse gas emissions: a synthesis of 20 years of experimentation. European Journal of Soil Science. 64: 357–373
7. Cowan NJ, Famulari D, Levy P E, Anderson M, BellMJ, Rees RM, ReayDS and SkibaUM (2014) An improved method for measuring soil N₂O fluxes using a quantum cascade laser with a dynamic chamber: dynamic chamber method Eur. J. Soil Sci. 65 643–52
8. Davidson, E. A. (2009) The contribution of manure and fertilizer nitrogen to atmospheric nitrous oxide since 1860. Nat. Geosci. 2, 659–662.
9. Davidson EA, Galloway JN, Millar N, Leach AM (2014) N-related greenhouse gases in North America: innovations for a sustainable future. Current Opinion in Environmental Sustainability, 9–10:1–8
10. Domeignoz-Horta LA, Putz M, Spor A, Bru D, Breuil MC, Hallin S (2016) Non-denitrifying nitrous oxide-reducing bacteria - An effective N₂O sink in soil. Soil Biology & Biochemistry 103: 376e379
11. Erisman JW, Sutton MA, Galloway J, Klimont Z, Winiwarter W (2008) How a century of ammonia synthesis changed the world. Nature Geoscience 1:636-639

12. Gao Y, Mania D, Mousavi SA, Lycus P, Arntzen MØ, Woliy K, Lindström K, Shapleigh JP, Bakken LR, Frostegård Å (2021) Competition for electrons favours N₂O reduction in denitrifying Bradyrhizobium isolates. *Environmental Microbiology* 23: 2244–2259
13. Gowda K, Ping D, Mani M, Kuehn S (2021) Genomic structure predicts metabolite dynamics in microbial communities. *Cell* 185, 530–546.
14. Gu B, Zhang X, Lam SK, Yu Y, van Grinsven HJM, Zhang S, Wang X, Bodirsky BL, Wang S, Duan J, Ren C, Bouwman L, de Vries W, Xu J, Sutton MA, Chen D (2023) Cost-effective mitigation of nitrogen pollution from global croplands. *Nature* 613: 77-84.
<https://doi.org/10.1038/s41586-022-05481-8>
15. Hallin S, Philippot L, Löffler FE, Sanford RA, Jones C (2018) Genomics and Ecology of Novel N₂O-Reducing Microorganisms. *Trends in Microbiology* 26:43-55
16. Hassan J, Bergaust LL, Wheat D, Bakken LR (2014) Low Probability of Initiating nirS Transcription Explains Observed Gas Kinetics and Growth of Bacteria Switching from Aerobic Respiration to Denitrification. *PLoS Comput Biol* 10(11): e1003933.
[doi:10.1371/journal.pcbi.1003933](https://doi.org/10.1371/journal.pcbi.1003933)
17. Hein S, Witt S, Simon J (2017) Clade II nitrous oxide respiration of *Wolinella succinogenes* depends on the NosG, -C1, -C2, -H electron transport module, NosB and a Rieske/cytochrome bc complex. *Environ Microbiol* 19:4913–4925
18. Henault C, Bourennane H, Ayzac A, Ratie C, Saby NPA, Cohan JP, Eglin T, Gali CL (2019) Management of soil pH promotes nitrous oxide reduction and thus mitigates soil emissions of this greenhouse gas. *Scientific Reports* 9:20182 <https://doi.org/10.1038/s41598-019-56694-3>
19. Hooper WD, Ash HB (1934) *Cato and Varro, on agriculture (translation)*. Loeb Classical Library ISBN 9780674993136
20. IPCC (2006) 2006 IPCC Guidelines for National Greenhouse Gas Inventories. Institute for Global Environmental Strategies, Hayama, Japan
21. Jonassen KR, Hagen LH, Vick SHW, Arntzen MØ, Eijsink VGH, Frostegård Å, Lycus P, Molstad L, Pope PB, Bakken LR (2022a) Nitrous oxide respiring bacteria in biogas digestates for reduced agricultural emissions. *The ISME Journal*; <https://doi.org/10.1038/s41396-021-01101-x>
22. Jonassen KR, Ormåsén I, Duffner C, Hvidsten TR, Bakken LR, HW Vick SHW (2022b) A dual enrichment strategy provides soil and digestate competent nitrous oxide-respiring bacteria for mitigating climate forcing in agriculture. *mBio*, 13.3: e00788-22.
23. Kanter DR, Brownlie WJ (2019) Joint nitrogen and phosphorus management for sustainable development and climate goals. *Environmental Science and Policy* 92:1–8
24. Kaminsky LM, Trexler RV, Malik RJ, Hockett KL, Bell TH (2019). The inherent conflicts in developing soil microbial inoculants. *Trends Biotechnol*, 37:140–51.
25. Kim D, Han H, Yun T, Song MJ, Terada A, Laureni M, Yoon S (2022) Identification of nosZ-expressing microorganisms consuming trace N₂O in microaerobic chemostat consortia dominated by an uncultured Burkholderiales. *The ISME Journal* 16:2087–2098;
<https://doi.org/10.1038/s41396-022-01260-5>
26. Klimasmith IM, Kent AD (2022) Micromanaging the nitrogen cycle in agroecosystems. *Trends in Microbiology*, 30:1045-1055
27. Lawson CE, Harcombe WR, Hatzenichler R et al (2019) Common principles and best practices for engineering microbiomes. *Nature Rev Micr* 17:725-741
28. Liu B, Frostegård Å, Bakken LR (2014) Impaired Reduction of N₂O to N₂ in Acid Soils Is Due to a Posttranscriptional Interference with the Expression of nosZ. *mBio* 5: 01383-14
<https://doi.org/10.1128/mbio.01383-14>

29. Lycus P, Bøthun KL, Bergaust L, Shapleigh JP, Bakken LR, Frostegård Å (2017) Phenotypic and genotypic richness of denitrifiers revealed by a novel isolation strategy. *The ISME Journal* 11:2219-2232.
30. Lycus P, Soriano-Laguna MJ, Kjos M, Richardson DJ, Gates AJ, Milligan DA, Frostegård F, Bakken LR (2018) A bet-hedging strategy for denitrifying bacteria curtails their release of N₂O. *PNAS* 115: 1182-11825.
31. Mallon CA, Le Roux X, van Doorn GS, Dini-Andreote F, Poly F, Salles JF (2018) The impact of failure: unsuccessful bacterial invasions steer the soil microbial community away from the invader's niche. *ISME J.* 2018;12:728-41.
32. Nadeem, S., Bakken, L. R., Frostegard, Å., Gaby, J. C. & Dörsch, P. (2020). Contingent Effects of Liming on N₂O-Emissions Driven by Autotrophic Nitrification. *Frontiers in Environmental Science*, 8: 8-16. doi: 10.3389/fenvs.2020.598513.
33. Pessi IS, Viitamäki S, Virkkala AM, Eronen-Rasimus E, Delmont TO, Luoto M, Hultman J (2020) Truncated denitrifiers dominate the denitrification pathway in tundra soil metagenomes. *BioRxiv* doi: <https://doi.org/10.1101/2020.12.21.419267>
34. Qu Z, Bakken LR, Molstad L, Frostegård Å, Bergaust LL (2016). Transcriptional and metabolic regulation of denitrification in *Paracoccus denitrificans* allows low but significant activity of nitrous oxide reductase under oxic conditions. *Environ Microbiol.* 18:2951-63.
35. Reay DS, Davidson EA, Smith KA, Smith P, Melillo JM, Dentener F, Crutzen PJ (2012) Global agriculture and nitrous oxide emissions. *Nature Climate Change* 2:410-416.
36. Roco CA, Bergaust LL, Bakken LR, Yavitt JB, Shapleigh JP (2017) Modularity of nitrogen-oxide reducing soil bacteria: linking phenotype to genotype. *Environmental Microbiology* 19:2507-2519 doi:10.1111/1462-2920.13250
37. Rockström J, Steffen W, Noone K, Persson Å, Chapin III FS, Lambin E, Lenton TM, Scheffer M, Folke C, Schnellhuber HJ, Nykvist B, de Wit CA, Hughes T, van der Leuw S, Rodhe H, Sörlin S, Snyder PK, Costanza R, Svedin U, Falkenmark M, Karlberg L, Corell RW, Fabry VJ, Hansen J, Walker B, Liverman D, Richardson K, Crutzen P, Foley J (2009) Planetary Boundaries: Exploring the Safe Operating Space for Humanity. *Ecology and Society* 14(2): 32. Online: <http://www.ecologyandsociety.org/vol14/iss2/art32/>
38. Seitzinger SP, Phillips L (2017) Nitrogen stewardship in the Anthropocene. *Science* 357:350-351.
39. Simon, J (2021) Mitigation of laughing gas emissions by nitrous oxide respiring microorganisms. In *Enzymes for Solving Humankind's Problems: Natural and Artificial Systems in Health, Agriculture, Environment and Energy*; Moura, J.J.G., Moura, I., Maia, L.B., Eds.; Springer International Publishing: Cham, Switzerland, pp. 185-211.
40. Shan J, Sanford RA, Chee-Sanford J, Ooi SK, Löffler FE, Konstantinidis KT, Yang WH (2021) Beyond denitrification: The role of microbial diversity in controlling nitrous oxide reduction and soil nitrous oxide emissions. *Glob Change Biol.*, 27:2669-2683.
41. Suenaga T, Hori T, Riya S, Hosomi M, Smets BF, Terada A (2019) Enrichment, Isolation, and Characterization of High-Affinity N₂O-Reducing Bacteria in a Gas-Permeable Membrane Reactor. *Environ. Sci. Technol.* 2019, 53, 12101-12112
42. Sutton MA, Onema O, Erisman W, Leip A, van Grinsven H, Winiwarter W (2011) Too much of a good thing. *Nature* 472:160-161
43. Taylor JP, Wilson B, Mills MS, Burns RG (2002) Comparison of microbial numbers and enzymatic activities in surface soils and subsoils using various techniques. *Soil Biol Biochem* 34:387-401.
44. Tian H, Xu R, Canadell JG, Thompson RL, Winiwarter W, Suntharalingam P, Davidson EA, Ciais P, Jackson RB, Janssens-Maenhout G, Prather MJ, Regnier P, Pan N, Pan S, Peters GP, Shi H,

- Tubiello FN, Zaehle S, Zhou F, Arneith A, Battaglia G, Berthet S, Bopp L, Bouwman AF, Buitenhuis ET, Chang J, Chipperfield MP, Dangal SRS, Dlugokencky E, Elkins JW, Eyre BD, Fu B, Hall B, Ito A, Joos F, Krummel PB, Landolfi A, Laruelle GG, Lauerwald R, Li W, Lienert S, Maavara T, MacLeod M, Millet DB, Olin S, Patra PK, Prinn RG, Raymond PA, Ruiz DJ, van der Werf GR, Vuichard N, Wang J, Weiss RF, Wells KC, Wilson C, Yang J, Yao Y (2020) A comprehensive quantification of global nitrous oxide sources and sinks. *Nature* 586:248-256.
45. van Spanning RJM, Richardson DJ, Ferguson SJ (2007) Introduction to the Biochemistry and Molecular Biology of Denitrification. In *Biology of the Nitrogen Cycle*. Bothe, H., Ferguson, S.J., and Newton, W.E. (eds). Amsterdam, the Netherlands: Elsevier, pp. 3–20.
 46. van Veen JA, van Overbeek LS, van Elsas JD (1997) Fate and activity of microorganisms introduced into soil. *Micr and Molec Biol Rev* 61:121-135.
 47. Vatn A (2015) *Environmental Governance: institutions, Policies and Actions*. Edward Elgar Publishing Ltd, Cheltenham UK. ISBN 978 1 78100 725 9
 48. Vick SHW, Jonassen KR, Arntzen MØ, Lycus P, Bakken LR (2023) Meta-omics analyses of dual substrate enrichment culturing of nitrous oxide respiring bacteria suggest that attachment and complex polysaccharide utilisation contributed to the ability of *Cloacibacterium* strains to reach dominance. *BiorXiv* : <https://doi.org/10.1101/2023.06.04.543644>
 49. Wang Y, Yao Z, Zhan Y, Zheng X, Zhou M, Yan G, Wang L, Werner C, Butterbach-Bahl K (2021) Potential benefits of liming to acid soils on climate change mitigation and food security. *Glob Change Biol.*,27:2807–2821.
 50. Winiwarter W, Höglund-Isaksson L, Klimont Z, Schöpp W, Amann M (2018) Technical opportunities to reduce global anthropogenic emissions of nitrous oxide. *Environ. Res. Lett.* 13 014011. DOI: 10.1088/1748-9326/aa9ec9
 51. Yoon S, Nissen S, Park D, Sanford R, Løffler FE (2016) Nitrous Oxide Reduction Kinetics Distinguish Bacteria Harboring Clade I NosZ from Those Harboring Clade II NosZ. *Appl Env Micr* 82:3793-3800.
 52. Zhang X, Davidson EA, Mauzerall DL, Searchinger TD, Dumas P, Shen Y (2015) Managing nitrogen for sustainable development. *Nature* 528: 51-59
 53. Zumft, WG (1997) Cell biology and molecular basis of denitrification. *Microbiol Mol Biol Rev.* 61, 533–616.

Unlocking bacterial potential: A solution to farmland N₂O-emissions

Elisabeth G Hiis, Silas H W Vick, Lars Molstad, Kristine Røsdal, Kjell Rune Jonassen, Wilfried Winiwarter, Lars R Bakken

Supplementary Information

Contents:

1 The respiratory phenotype of CB-01

2 Methods

2A Determining the phenotype by robotized batch-cultivation

2B Culturing CB-01 in digestate for fertilization experiment

2C Monitoring N₂O emission by a field robot

2D Field-bucket experiments

2E Field plot experiments

2F Calculations of emissions and statistical analyses

2G Tracing CB-01 in digestate and soil

2H Survival of CB-01 in soil, laboratory incubation

2J Extrapolating to national emission reductions

3 Effect of CB-01 on the soil microbiome

4 Survival of CB-01 in the field plot experiments

1 The respiratory phenotype of CB-01

Non-denitrifying N₂O-respiring bacteria (NNRB) have attracted much interest recently as net sinks for N₂O in soils, potentially curbing N₂O emissions if abundant (Hallin et al. 2018, Simon 2021). NNRB-strains vary grossly in their apparent capacity to act as N₂O-sinks, assessed by determining their biokinetic parameters: NNRB strains are commonly assumed to be strong N₂O sinks if they have strong affinity (low apparent k_m) for N₂O and a high maximal rate of N₂O reduction (V_{max}), or simply a high catalytic efficiency, i.e. a high V_{max}/k_m (Yoon et al. 2016). Another desirable, albeit speculative feature would be to reduce N₂O under aerobic or at least hypoxic conditions (Kim et al. 2022).

To assess *Cloacibacterium* sp. CB-01 along these criteria, and to compare it with other strains, we conducted an in-depth investigation of its respiratory phenotype using a robotized incubation system (Molstad et al. 2016) which provides high resolution gas kinetics (CH₄, CO₂, O₂, NO, N₂O and N₂) in batch cultures in gas tight vials, as they deplete oxygen and switch to anaerobic respiration, reducing N₂O to N₂. Combined with adequate calculation of gas transport, this approach has proven powerful in unravelling novel regulatory features such as *bet-hedging* (Hassan et al. 2014, Lycus et al. 2018), as well as characterizing key enzyme parameters *in vivo* (Hassan et al. 2016a).

Fig S1: Growth yield by aerobic and anaerobic respiration. The growth yield of CB-01 by aerobic and anaerobic respiration was assessed by batch cultivation in 50 mL nutrient broth in gas tight 120 mL serum vials, stirred by magnetic bars. Prior to inoculation, the 18 vials were He-washed by repeated evacuation and He-filling (Molstad et al. 2007) and provided with different amounts of N₂O and O₂ by injection with a syringe. They were then placed in the thermostatic water-bath (temperature = 23 °C) of the robotized incubation system (Molstad et al. 2016). After temperature equilibration and subsequent release of overpressure due to N₂O- and O₂-injection, the vials were inoculated with 3.5*10¹⁰ cells (7*10⁸ cells mL⁻¹). The incubation system monitors the O₂, N₂O and N₂ concentration in the headspace, and these measurements were used to estimate the cumulated reduction of O₂ and N₂O throughout the incubation for each vial. The inserted panel shows an example for a single vial with 5 mL N₂O and 5 mL O₂ injected. The cumulated O₂- and N₂O-reduction (to N₂) do not add up to 100 % of the initial amounts because each sampling removes a fraction of the headspace gas (replaced by helium, see Molstad et al. 2007).

When all the electron acceptors were depleted, the cell density was measured by OD at 600 nm. The relationship between cell density and OD₆₀₀ was measured in separate experiments with suspensions of CB-01 with a range of densities (microscopic counts), which showed a linear relationship for OD₆₀₀ ≤ 0.5 (cell density = 3.34*10⁹ mL⁻¹ OD⁻¹). Thus, any sample with OD₆₀₀>0.5 was diluted to reach OD₆₀₀<0.5 for determination of cell density. In the same experiment, the cell dry weight of CB-01 was determined by weighing dry cells (cells washed three times in distilled water by dispersion and centrifugation, then dried at 105°C). The dry weight was 108 fg cell⁻¹ ± SE= 7.5 (n=9).

The measured yield per mol of N₂O and O₂ was found by using the Generalized Reduced Gradient Solver in Excel for the entire dataset. The panel shows the result for individual vials, as a plot of the predicted cell density (based on the yields given below) against measured cell density. The estimated yields were Y_{N₂O}= 1.7*10¹⁴ cells mol⁻¹ N₂O and Y_{O₂}=4*10¹⁴ cells mol⁻¹ O₂. The yields per mol electrons are Y_{e-N₂O}= 0.85*10¹⁴ mol⁻¹ e⁻ to N₂O and Y_{e-O₂}= 1*10¹⁴ cells mol⁻¹ e⁻ to O₂. The yields in terms of dry weight g (given the dry weight= 108 (± SE= 7.5) fg cell⁻¹), are Y_{e-N₂O}= 9.2 ± 0.6 g mol⁻¹ e⁻ to N₂O and Y_{e-O₂}= 11 ± 0.8 g mol⁻¹ e⁻ to O₂.

In comparison, Bergaust et al. (2010) found *Paracoccus denitrificans* to have Y_{e-O₂}= 3.75*10¹³ cells mol⁻¹ e⁻ to O₂ = 11.2 g cell dry weight mol⁻¹ e⁻ to O₂ (cell dry weight=298 fg), which is practically identical with Y_{e-O₂} determined for CB-01. Y_{e-N₂O} was 85% of Y_{e-O₂} ratio for CB-01, which is surprisingly high compared to that measured for *P. denitrificans* (53%), and compared to the expectations (~0.6) based on the charge separation per electron for aerobic and anaerobic respiration for denitrifying organisms carrying NosZ clade I (van Spanning et al. 2007). However, there is mounting evidence that some organisms with NosZ Clade II have higher yields per electron to N₂O than organisms with NosZ clade I (Yoon et al. 2016), suggesting that the electron pathway to NosZ Clade II generates more charge separations than the pathway to NosZ Clade I, which is thermodynamically possible (Hein and Simon 2019, Simon 2021).

Fig S2. Aerobic and anaerobic growth rates. The aerobic and anaerobic growth rates were determined by nonlinear regression of the rates of O_2 - and N_2O -reduction against time during unrestricted growth in nutrient broth in stirred batch cultures in 120 mL serum vials (23 °C). **Panel A** shows the rate of oxygen consumption in three replicate vials with 6 vol% O_2 in the headspace, inoculated with $\sim 7 \cdot 10^8$ cells mL^{-1} . The oxygen concentration in the liquid was 80 μM initially, declining to 50 μM at the end, thus growth was not restricted by the availability of O_2 . The growth rate in each vial was estimated by nonlinear regression (equations and R^2 values shown in the panel) and the average aerobic growth rate is $0.29 h^{-1}$ (stdev = 0.006). **Panel B** shows the rate of N_2O reduction in three replicate vials after O_2 -depletion, when the cell density had reached $\sim 1.7 \cdot 10^{10}$ vial $^{-1}$. The N_2O concentration in the liquid was 280 μM initially, declining to 110 μM at the end (14 h), thus growth was not restricted by availability of N_2O . The anaerobic growth rate was calculated for each vial as in panel A, and the average is $0.11 h^{-1}$ (stdev = 0.001). Given these aerobic and anaerobic unrestricted growth rates, we can calculate V_{max} per cell for O_2 and N_2O ($V_{max} = \mu/Y$). The estimated values are $V_{maxN_2O} = 0.6$ fmol N_2O cell $^{-1} h^{-1}$, $V_{maxO_2} = 0.72$ fmol O_2 cell $^{-1} h^{-1}$. The dry weight of the CB-01-cells is 108 fg, thus V_{maxN_2O} expressed on a dry weight basis is 0.0059 mol N_2O g $^{-1}$ dry weight h^{-1} . The maximal electron transport rates for aerobic and anaerobic respiration are $V_{e_{maxO_2}} = 2.9$ fmol e^- to O_2 cell $^{-1} h^{-1}$, $V_{e_{maxN_2O}} = 1.2$ fmol e^- to N_2O cell $^{-1} h^{-1}$. Thus, the aerobic respiratory rate of CB-01 is more than twice the anaerobic respiration rate.

Fig S3 Onset of N_2O -reduction during O_2 depletion.

Denitrifying bacteria vary as to how early they initiate anaerobic respiration during O_2 -depletion (Bergaust et al. 2011). To explore this for *Cloacibacterium* sp. CB-01, we ran several experiments, both in nutrient broth (Panel A-C) and in digestate (panel D). **Panel A&B:** show the results for experiments where the inoculum was raised through >10 generations under strict aerobic conditions, thus diluting out any N_2O reductase that might be present in the cells. **Panel A** shows the gas measurements, the O_2 -concentration in the liquid as calculated from the O_2 transport rate (see Molstad et al. 2007), and the rate of N_2O -reduction (V_{N_2O}). **Panel B** shows the ratio $V_{e_{N_2O}}/V_{e_{tot}}$, i.e. the fraction of total electron flow that goes to N_2O (for each time increment), plotted against the O_2 concentration in the liquid. **Panel C** shows $V_{e_{N_2O}}/V_{e_{tot}}$ (plotted against $[O_2]$) for an experiment where the inoculum had been exposed to hypoxia, thus with NosZ expressed already. **Panel D** shows the result for growth in digestate, inoculated with cells raised aerobically.

Fig S4 The electron flow kinetics during transition from aerobic to anaerobic respiration suggests *bet-hedging*. To investigate the characteristic denitrification regulatory phenotype of CB-01, 120 mL vials with 50 mL nutrient broth and $O_2 + N_2O$ in He-atmosphere were inoculated with $2.7 \cdot 10^8$ cells per vial, which had been raised under strict aerobic conditions ($n=3$ replicate vials). The vials were monitored for gas kinetics as the cultures grew by oxygen initially, and then switched to respiring N_2O in response to O_2 -depletion. The experiment included four treatments, all with 1 mL N_2O ($50 \mu\text{mol } N_2O \text{ vial}^{-1}$), but five different amounts of O_2 (0, 0.8, 1.4, 3 and 4.6 mL O_2 (3 replicate vials for each O_2 level). The panel shows the result for the vials with 0.8 mL. **Panel A** shows the measured amounts of O_2 and N_2O per vial. **Panel B** shows the electron flow rates to O_2 and N_2O (and total electron flow rate as a dashed line) as calculated from the measured O_2 and N_2O . Two phenomena stand out here: as oxygen was depleted, the electron flow rate declined to very low values and the subsequent electron flow rate to N_2O increased exponentially, with an apparent growth rate of 0.12 h^{-1} (which is slightly higher than the anaerobic growth rate of CB-01 determined previously). This is the typical pattern for a *bet-hedging* denitrifying organism, i.e. an organism which expresses denitrification enzymes only in a fraction of the cells (Hassan et al. 2014, Lycus et al. 2017). Assuming this, we investigated the possible fraction of cells that express NosZ and engaged in anaerobic respiration and growth: **Panel C** shows the estimated total number of cells (based on the cumulated O_2 and N_2O -consumption and the yields per mol O_2 and N_2O , **Fig S1**), and the number of N_2O -respiring cells (Nos-active) calculated from the measured N_2O -reduction rate (V_{N_2O} , mol $N_2O \text{ vial}^{-1} \text{ h}^{-1}$) and the assumption that $V_{\text{max}N_2O} = 0.65 \text{ fmol } N_2O \text{ cell}^{-1} \text{ h}^{-1}$ (as determined previously): $N_{\text{Nos-active cells vial}^{-1}} = V_{N_2O} / V_{\text{max}N_2O}$. The blue dashed line is the estimated number of cells without Nos (Nos inactive), assumed to be cells entrapped in anoxia without Nos, hence unable to synthesize Nos. **Panel D** shows the number of Nos-active cells as numbers per vial, and as fraction of the total number of cells in the vial. This fraction increases with time due to growth by N_2O -respiration, and the fraction at the time of O_2 depletion is a crude estimate of the fraction of cells which were able to express Nos before O_2 is completely exhausted.

Fig S5 Bet-hedging depending on cell density at the time of O₂ depletion. Lycus et al. (2018) provided experimental proof for *bet-hedging* in *Paracoccus denitrificans*, hypothesized by Hassan et al. (2014) based on modelling of the type of electron flow kinetics throughout the transition from oxic to anoxic respiration as shown in Fig S4. In *P. denitrificans*, the fraction of cells that were able to switch to anaerobic respiration (F_{den}), thus avoiding entrapment in anoxia (Kellermann et al. 2022), was proportional to the time length of the hypoxic phase preceding complete anoxia. This was ascribed to a stochastic initiation of transcription (of *nirS*, coding for nitrite reductase) once the cells experience hypoxia. To investigate if the apparent *bet-hedging* in CB-01 shows the same pattern, we estimated F_{nosZ} for all the 15 vials in the experiment reported in Fig S4: the vials were all provided with 1 mL N₂O but five different amounts of O₂ (0, 0.8, 1.4, 3 and 4.6 mL O₂, n= 3 replicate vials for each O₂-level). The cell density at the time of O₂-depletion increased with increasing initial O₂ (Panel A), while the time length of exposure to hypoxia (arbitrarily defined as 0.2-4 μM O₂) declined (Panel B). A simplified version of the *bet-hedging* model (Hassan et al. 2016b), assuming instantaneous expression of NosZ in a fraction of the cells (F_{NosZ}) as O₂ reached below 0.5 μM, was fitted to observed gas kinetics (O₂ and N₂O) for each vial to estimate F_{NosZ} (all other parameters were as determined previously (Fig S1-3). Contrary to our expectations, the estimated fraction of cells expressing *nosZ* (F_{NosZ}) decreased with increasing time length of exposure to hypoxia (Panel C) and increased with cell density at the time of O₂ depletion (Panel D). Interestingly, practically all cells appeared to become entrapped in anoxia ($F_{den} = 0.002-0.006$) in the vials without any O₂ injected. The results warrant further investigations to provide direct evidence for the cell differentiation (*bet-hedging*), and the mechanism causing F_{den} to increase with cell density. A tantalizing hypothesis is that quorum sensing induction of NosZ expression is involved.

Fig S6. Apparent affinity for O₂ and N₂O. To assess the affinity for O₂ and N₂O, we measured the rates of O₂- and N₂O-reduction as the batch cultures depleted the two electron acceptors. The rates as measured (mol vial⁻¹ h⁻¹) were converted to rates per actively respiring cell (V_{O_2} and V_{N_2O} , fmol cell⁻¹ h⁻¹) based on the numbers of active cells in the vial at each time point (= the midpoint between two samplings). The number of active cells were the numbers of cells in the inoculum + new-grown cells as calculated from the cumulated consumption of O₂ and N₂O (and the growth yield per mol, Fig S1), as done previously for determining the affinity for NO (Hassan et al. 2016a). For V_{N_2O} , the number of actively N₂O-respiring cells was only a fraction of the total (as shown in Fig S5 panels C&D). The maximum rates V_{max} and the apparent k_m values were found by fitting the Michaelis Menten model $V = V_{max} * S / (k_m + S)$ to the data (S is the concentration of O₂ and N₂O in the liquid) by least square, using the Generalized Reduced Gradient Solver in Excel. This was done for each individual vial, and for the collective datasets. Panel A and B show the results for O₂, for a single vial (A) and for the entire dataset (B). Embedded in the panel is the estimated k_m for each individual vial. Panel C and D show the results for N₂O, for a single vial (A) and for the entire dataset (B). Embedded in the panel is the estimated k_m for each individual vial. These results show a relatively strong affinity for O₂ ($k_m \sim 1 \mu\text{M O}_2$), and rather weak affinity for N₂O ($k_m \sim 13 \mu\text{M N}_2\text{O}$).

Table S1. Comparison of biokinetic parameters for CB-01 and other N₂O-respiring bacteria. The maximum rates of N₂O-respiration (V_{max}), and the affinity for N₂O, normally expressed as the half saturation concentration (K_m), have been measured in various N₂O-respiring strains to assess and compare their capacity to scavenge N₂O in soil. A plausible way to rank strains according to their capacity to reduce N₂O emission at low N₂O-concentrations is to calculate their V_{max}/K_m ratios, since this approximates the slope of the rate against N₂O-concentration ($[N_2O]$) at very low concentrations ($[N_2O] \ll K_m$). At very high N₂O-concentrations, the capacity to reduce emissions is proportional with V_{max} . In the table below, we have listed V_{max} and K_m determined for a range of N₂O-respiring strains. Some report V_{max} as fmol N₂O cell⁻¹ h⁻¹, while others report it as mol g⁻¹ cell dry weight h⁻¹. To enable a comparison, we have converted all V_{max} values to mol N₂O g⁻¹ cell dry weight h⁻¹, assuming a cell volume = ~0.6 μm³ and 200 fg dry weight per cell (Bakken and Olsen 1983), and converted all values to V_{max} at 20 °C assuming the rates increase exponentially with temperature by a factor of 2.5 per 10 °C increase in temperature ($Q_{10}=2.5$). The calculated V_{max20}/K_m values are plotted against V_{maxN20} in Figure 1 (Panel B) in the main paper.

	nosZ	temp	Vmax	Vmax	Km	V _{max20}	V _{max20} /K _m	Reference
Organism	Clade	(°C)	fmol cell ⁻¹ h ⁻¹	Mol g ⁻¹ DW h ⁻¹	μM N ₂ O	mmol g ⁻¹ h ⁻¹ at 20°C	L μg ⁻¹ h ⁻¹ at 20°C	
Pseudomonas stutzeri DCP1	I	30		0.250	35.5	99.840	2.812	Yoon et al. 2016
Shewanella loihica PV4	I	30		0.027	7.07	10.704	1.514	Yoon et al. 2016
Paracoccus denitrificans DSM413	I	20		0.027	0.59	26.760	45.356	Hassan et al. 2016b
Pseudomonas stutzeri JCM5965	I	30	1.64	0.008	4.01	3.280	0.818	Suenaga et al. 2018
Paracoccus denitrificans NBRC102528	I	30	0.51	0.003	34.8	1.020	0.029	Suenaga et al. 2018
Paracoccus denitrificans NBRC102528	I	30	0.51	0.003	1.1	1.020	0.927	Qi et al. 2022
Pseudomonas stutzeri JCM5965	I	30	2.66	0.013	1.01	5.320	5.267	Qi et al. 2022
Alicyclophilus denitrificans I51	I	30	3.78	0.019	8.98	7.560	0.842	Suenaga et al. 2019
Dechloromonas aromatica RCB	II	30		0.028	0.324	11.064	34.148	Yoon et al. 2016
Anaeromyxobacter dehalogenans 2CPC	II	30		0.001	1.34	0.410	0.306	Yoon et al. 2016
Cloacibacterium sp. CB-01	II	23		0.006	12.9	4.558	0.353	this study
Azospira sp. I09	II	30	0.634	0.003	0.868	1.268	1.461	Suenaga et al. 2018
Azospira sp. I13	II	30	5.8	0.029	3.76	11.600	3.085	Suenaga et al. 2018
Azospira sp. I09	II	30	1.24	0.006	0.54	2.480	4.593	Qi et al. 2022
Azospira sp. I13	II	30	18.84	0.094	2.12	37.680	17.774	Qi et al. 2022
Dechloromonas sp. I20	II	30	18	0.090	2.04	36.000	17.647	Suenaga et al. 2019
Azospira sp. I09	II	30	4.23	0.021	1.55	8.460	5.458	Suenaga et al. 2019
Azospira sp. I13	II	30	17.9	0.090	2.1	35.800	17.048	Suenaga et al. 2019
Dechloromonas aromatica RCB	II	30	7.748	0.039	0.324	15.496	47.827	Suenaga et al. 2019
Anaeromyxobacter dehalogenans 2CP-C	II	30	0.287	0.001	1.34	0.574	0.428	Suenaga et al. 2019

Commented [P1]: Fix the sort - fmol N2O

Commented [P2R1]: Fix km in last column - suggest small k for half sat constant

2 Methods

2A Robotized batch cultivations for determination of respiratory phenotypes

The respiratory phenotypic parameters of CB-01 were determined by batch culturing in the robotized incubation system designed and described by Molstad et al. (2007, 2016). The system hosts up to 30 parallel stirred batch cultures (normally 50 mL) in 120 mL gas tight serum vials with He-atmosphere (with or without N₂O and O₂), which are sampled frequently for measuring the concentrations of O₂, N₂, N₂O, NO and CO₂. Robust routines are established for calculating the rates of production/consumption of all the gases (taking sampling-loss and leakage into account), and for calculating gas concentrations in the liquid as a function of gas concentrations and the rate of transport between liquid and headspace. These routines are included in a spreadsheet which is publically available, including a set of instruction videos (Bakken 2021). The system has been used in numerous investigations of the respiratory phenotypes of denitrifying bacteria (Bergaust et al. 2008, 2010; Hassan et al. 2016ab, Jonassen et al. 2022ab, Gao et al. 2021, Kellerman 2022, Qu et al. 2016).

To enable refined analyses of the respiratory phenotype of CB-01, we initially determined the cell dry weight (fg cell⁻¹), and the growth yields for aerobic (Y_{O_2} , cells mol⁻¹ O₂) and anaerobic (Y_{N_2O} , cells mol⁻¹ N₂O) respiration by measuring the cell yields in batches provided with various amounts of O₂ and N₂O. This enabled inspection of the cell specific respiration rates (fmol cell⁻¹ h⁻¹) throughout subsequent batch incubations, based on measured rates (mol O₂ & N₂O vial⁻¹ h⁻¹) for each time interval between two gas samplings, and the estimated cell number in the vial for the same time interval ($=N_{ini} + Y_{O_2} * CumO_2 + Y_{N_2O} * CumN_2O$, where N_{ini} is the initial number of cells at time=0, $CumO_2$ and $CumN_2O$ are the cumulated consumption of the two gases). The cell specific rates calculated this way allowed an analysis of the affinity for O₂ and N₂O by plotting cell specific rates of O₂ and N₂O against the concentrations of the two gases in the liquid as the cultures depleted the gases, and fitting the Michaelis-Menton function to these data (least square). Batch cultures provided with both N₂O and O₂ in the headspace were monitored as they depleted O₂ and switched to respiring N₂O, thus determining the critical concentration of O₂ (in the liquid) at which the cells started to respire N₂O. The kinetics of electron flow throughout such transitions from aerobic to anaerobic respiration were used to assess the fraction of cells expressing N₂O-reductase in response to O₂-depletion, using a simplified version of the model developed by Hassan et al. (2016b).

2B Culturing CB-01 in digestate for field experiments

For each field experiment, fresh digestate was collected from a wastewater treatment plant close to Oslo (VEAS), described in Jonassen et al. (2022a). Averaged values of the quality parameters for the period of digestate collection were: dry matter content = 3.97 weight % (stdev=0.16), ignition loss of dry matter = 55.6 % (stdev=2), pH = 7.72 (stdev=0.07) and $\text{NH}_3+\text{NH}_4^+$ = 1.71 g N L⁻¹ (stdev=0.12).

Prior to cultivation of CB-01, the digestate was heat-treated, aerated and pH-adjusted. For the field bucket experiments (Chapter 2B), the digestate was autoclaved (121 °C for 20 min), and then sparged with air (while stirred) for 48 hours to secure chemical oxidation of Fe²⁺ to Fe³⁺, then autoclaved again. Oxidation of Fe²⁺ by air sparging was considered necessary to avoid abiotic oxygen consumption, as the digestate had high concentrations of Fe²⁺ originating from the Fe³⁺ used as precipitation chemicals in the primary wastewater treatment, and reduced to Fe²⁺ in the anaerobic digesters (Jonassen et al. 2022a). The sparging caused the pH to increase to 9.4 due the removal of CO₂, requiring a final pH adjustment to 7.3 (with HCl). The same procedure was used for the field plot experiment, except that autoclaving was replaced by heat treatment: 70°C for 4 hours.

CB-01 was then grown aerobically in the pretreated digestates, inoculated to an initial cell density of $\sim 5 \cdot 10^7$ cells mL⁻¹, which were stirred and sparged with sterile air (filtered) at 23 °C. To monitor the growth of CB-01, we transferred subsamples of each batch (after inoculation) to 120 mL vials (50 mL vial⁻¹) with teflon coated magnetic stirring bars, which were placed in the incubation robot system for monitoring the O₂ consumption (Fig S7).

Fig S7. Cultivation of CB-01 in digestate. The panels show the kinetics of O₂-consumption as measured in the 50 mL subsamples placed in the incubation robot (4 replicate vials with CB-01, and 4 vials with sterile digestate).

Panel A shows the rates of O₂ consumption in vials with CB-01 and in the sterile controls, and the net consumption by CB-01 (CB-01 minus sterile control). This increased exponentially during the first 24 hours, with apparent growth rate 0.12 h⁻¹, which is much slower than in nutrient broth (0.29 h⁻¹ Fig S2).

Panel B shows the cumulated O₂ consumption by CB-01, and the estimated cell density assuming $Y_{O_2} = 4.06 \cdot 10^{14}$ cells mol⁻¹ O₂ (Fig S1), reaching $1.2 \cdot 10^{10}$ cells mL⁻¹. The cell density quantified by qPCR for a similar experiment only 47% of the density based on O₂ (Table S2), suggesting that Y_{O_2} for growth to high cell densities in digestate is $\sim 50\%$ of Y_{O_2} for optimal growth in nutrient broth (Fig S1).

Panel C shows estimated cell specific O₂ consumption (V_{O_2} , fmol O₂ cell⁻¹ h⁻¹), estimated growth rate, μ (h⁻¹) = $V_{O_2} \cdot Y$, where Y = the measured growth yield by aerobic respiration ($4.06 \cdot 10^{14}$ cells mol⁻¹ O₂), and the estimated fraction of organic C in the digestate (10 mg C mL⁻¹) consumed by CB-01 (sum of CO₂ and assimilated C). This suggests that $\sim 1\%$ of the organic C in the digestate was easily available monomers, supporting rapid growth of CB-01 to a cell density of $\sim 1 \cdot 10^9$ cells mL⁻¹ after 20 h, while subsequent growth was gradually declining as the organism utilized increasingly recalcitrant substrates, plausibly with a lower growth yield (Y_{O_2}).

Table S2. Growth yield for CB-01 when growing to high cell densities. The estimated growth of CB-01 in the digestate (Fig S7) was based on the growth yield as measured in batch cultures growing exponentially until limited by electron acceptors ($4 \cdot 10^{10}$ cells per mol O₂ consumed, Fig S1), and the final cell density in these cultures was $\leq 2.7 \cdot 10^9$ cells mL⁻¹. The growth yield is plausibly lower for cultures that grow to higher cell densities, be it in nutrient broth or in digestate, due to a gradually declining growth rate induced by limitation of substrate supply. To assess the true growth yields under these

conditions, we quantified the cell densities by real-time PCR (qPCR), and compared this with the estimated cell densities based on the oxygen consumption. The table shows this comparison both for growth to high cell densities in nutrient broth and digestate, confirming the hypothesis that growth yield is lower when the cultures reach high cell densities, and more so in digestate than in nutrient broth. Thus, the true cell density in the digestate used in the field experiments were approximately 50% of that estimated by the O₂-consumption.

Medium	Estimated cell density (10 ⁹ cells mL ⁻¹)		qPCR-estimate as % of estimate based on O ₂ -consumption
	Based on O ₂ - consumption*	Based on qPCR	
Nutrient broth	20.1	15.1 (stdev=0.7)**	75
Digestate	10.5	4.92 (stdev=0.08)**	47

* No standard deviation available for O₂ consumption. The sample was taken from a single vial

** n=3 subsamples

2C Monitoring N₂O emissions with a field robot

Emissions of N₂O in all outdoor experiments were monitored by the “dynamic chamber” technique (Cowan et al. 2014, Hensen et al. 2006), operated by an autonomous Field Flux Robot (FFR) described by Molstad et al. (2014) and Molstad et al. (in prep.)

Fig S8. The Field Flux Robot. FFR (Panel A) operates two chambers (diameter 50 cm, height 50 cm), which are lowered over field buckets (Panel A) or onto the soil surface (Panel B). The chambers are equipped with cellular foam with gas tight flexible rubber coating which is compressed by deployment, and a circular brush skirting to function as a wind break (Panels C&D). The concentrations of N₂O and CO₂ in the two chambers are measured by circulating the chamber-air (intermittently for each chamber) via a Tunable Diode Laser N₂O/CO-analyzer (DLT-100, Los Gatos Research, California, USA) and a CO₂/H₂O infrared gas analyzer (LI-840A, LI-COR Biosciences, Nebraska, USA), throughout a deployment time of 3 minutes.

2D Field-bucket experiments

Soils for the bucket experiments were collected from agricultural fields in southern Norway, spanning a range of soil characteristics:

Table S2 Origin and characteristics of the soils. The acid sandy silt soil (**S**) was taken from an agricultural field in Solør, Norway, dominated by fluvial sandy silt soils. The clay loam soils **L**, **I** and **N** were from different plots within a liming experiment near the Norwegian University of Life Sciences ((59°39'48.2"N 10°45'44.8"E), limed in 2014 (Nadeem et al. 2020), while **O** was a clay loam soil from the same area (hence with similar mineral components), but with a much higher content of organic C because it had been a peat-/wetland prior to cultivation. Soils **S**, **L**, **N** and **O** were used in the field bucket experiments. Soil **I** is the soil of the plots used for the field plot experiment.

Soil:	pH (CaCl ₂) *	Tot C (%) **	Tot N (%) **	[NO ₃] mg N kg ⁻¹ **
S : sandy silt soil	4.15	0.75	0.07	30.2
L : low-pH clay loam	4.50	3.23	0.24	55.6
I : intermediate pH clay loam	6.13	3.23	0.24	-
N : Neutral-pH clay loam	6.70	3.21	0.25	21.1
O : Organic-rich clay loam	5.26	15.8	0.78	11.2

¹ pH_{CaCl₂} was measured after dispersing 10 g soil in 25 mL of 0.01 M CaCl₂

² Tot C = total organic C, Tot N = total organic N, and [NO₃] = mg NO₃-N kg⁻¹ soil dry weight.

Soil **L**, **I** and **N** were taken from different plots of a liming experiment on clay loam soil established and limed in 2014 (Nadeem et al., 2020). The low-pH clay loam (**L**) received no lime, the intermediate pH clay loam (**I**) was limed with 2.3 kg m⁻² of dolomite, and the neutral-pH clay loam (**N**) was limed with 3 kg m⁻² of finely ground calcite.

The soils used in the bucket experiments (**S**, **L**, **N** and **O**) were sieved (10 mm) in moist condition and mixed thoroughly before filling into the buckets. The conically shaped buckets (h = 21.5 cm, top diam. = 23.5 cm, bottom diam. = 21.5 cm) had a total volume of 8.6 L. A ~1 cm layer of gravel (4-8 mm diam.) was placed at the bottom, covered with a nylon fiber cloth to prevent eluviation of the soil by drainage. For soils **S**, **L** and **N**, 8 kg soil dry weight were filled into each bucket, packed by thumping the bucket on the ground till the soil had reached a bulk density of 1 kg L⁻¹. For the organic rich clay loam soil, each bucket was filled with only 5.92 kg soil dry weight, reaching a bulk density of 0.74 kg L⁻¹ after being packed to 8 L. The soil surface area of the buckets was 0.043 m².

To secure equal initial amounts of NO₃ m⁻² for all soils, we mixed an amount of KNO₃ to each soil to reach a level of 12 g N m⁻² soil surface = 516 mg NO₃-N bucket⁻¹ (soil surface area = 0.043 m²). Digestate (480 mL bucket⁻¹ = 11 L m⁻² soil surface area) was mixed into the top ~10 cm of the soil by "harrowing", using a small hand-held rake. We used autoclaved digestates in which CB-01 had been grown to ~6*10⁹ cells mL⁻¹, and as the control treatment we heat-treated this digestate (70 °C, 2 h), which effectively killed the CB-01 cells (tested by measuring respiration, results not shown). As an additional control treatment, buckets received water only. The density of CB-01 cells per soil surface area immediately after application was 6.6*10¹³ cells m⁻². The cell density in the upper 10 cm of the soil was ~6*10⁸ cells g⁻¹ soil dry weight for the soils **S**, **L**, and **N** (bulk density = 1 kg L⁻¹), and ~8*10⁸ g⁻¹ for soil **O**.

The buckets were placed on 1 m² plexiglass plates (1.5 mm), to avoid gas exchange with the soil below. The soil moisture (volumetric water content, m³/m³) and temperature (°C) in the upper 5.5 cm of the soil were monitored by four Teros 11 sensors, connected to an EM50 logger (Meter Group, Inc., WA, USA).

In the first experiment, using only **SSoil N** (Table S2), starting 14.07.2021, rye grass was sown the day after the incorporation of the digestate, and the emissions were monitored for 90 days. Within this time span, we added 200 mL autoclaved and pH-adjusted digestate (4.6 L m^{-2}) without CB-01 three times (after 19, 33 and 89 days), to induce transient bursts of N_2O emission. By the end of each burst of N_2O -emission induced by applying digestates, the upper 10 cm of the soil was sampled with an auger (diam. 1 cm) and stored in the freezer ($-4 \text{ }^\circ\text{C}$) until DNA extraction and subsequent molecular work. The auger was washed and sterilised with 70 % ethanol between each sampling.

In a follow up experiment, all soils were included and monitored for 10 days, with no re-fertilisation. Soil sampling was performed after the first peak of N_2O emissions, as described for the 90 days bucket experiment.

The digestate application's influence on soil pH was tested in the lab by mixing soil with the same type and amount of digestate as applied to the 0-10 cm soil layers of the field buckets (0.11 mL g^{-1} soil) $\pm 50 \%$ to show the potential pH in pockets with higher or lower than average concentration of digestate. Water was added (if needed) together with digestate to reach the same water-filled pore space (WFPS, %) as in the field bucket experiment. The most prominent increase in soil pH was seen in the sandy silt soil, reflecting its low buffer capacity due to low content of clay and organic material (Table S1), both known to be crucial factors determining the soils' buffer capacities (Curtin et al. 1996).

Table S3. Soil pH changes due to application of digestate. The table shows pH(CaCl_2) in the four soils as affected by applying 0.055, 0.11 and 0.165 g digestate g^{-1} soil, which is 50, 100 and 150 % of the amounts added to the soils in the field bucket experiment ($0.11 \text{ mL digestate g}^{-1}$ soil). Water was added (if needed) to reach a WFPS (%) equivalent to that in the buckets after digestate application (last column).

Soil type	no digestate	0.055 mL digestate g^{-1} soil	0.11 mL digestate g^{-1} soil	0.165 mL digestate g^{-1} soil	WFPS (%)
S: Sandy silt soil	4.15	5.02	5.33	5.59	49
L: Low-pH clay loam	4.50	4.87	5.00	5.11	53
N: Neutral-pH clay loam	6.70	6.75	6.75	6.74	54
O: Organic-rich clay loam	5.26	5.42	5.52	5.59	45

2E Field plot experiments

To assess the effect of CB-01 in digestate on soil N_2O emission under field conditions, we established small (0.5 m^2) test plots within larger field plots ($8 \text{ m} \times 3 \text{ m}$) of a soil liming experiment (limed in 2014) on clay loam soil (Nadeem et al. 2020, Byers et al. 2021) and re-limed with $174 \text{ g dolomite m}^{-2}$ in 2019. We used the plots with soil I (Table S2) that were previously limed with dolomite to $\text{pH}(\text{CaCl}_2) = 6.13$ (stdev = 0.10), and within each of the 6 replicate plots, we established two $0.7 \text{ m} \times 0.7 \text{ m}$ test plots side by side (distance = 30 cm), fertilized with autoclaved digestate in which CB-01 had been grown to a cell density of $\sim 6 \times 10^9$ cells mL^{-1} . We applied $4.5 \text{ L digestate per plot}$ ($= 9 \text{ L m}^{-2}$), which was mixed into the upper $\sim 10 \text{ cm}$ of the soil by a hand-held cultivator. The initial density of CB-01 was 5.4×10^{13} cells m^{-2} . If distributed throughout the the soil layer that was sampled for analyses ($0\text{-}10 \text{ cm depth} = 125 \text{ kg soil dry weight m}^{-2}$, assuming a bulk density of 1.25 kg L^{-1}), the initial cell density in the soil would be 4.3×10^8 cells g^{-1} soil. Soil samples for determining CB-01 abundance were taken from each plot (3 replicate samples) before incorporation of digestate with CB-01, 9 days later, and after 10 months. The soil samples were stored in the freezer ($-20 \text{ }^\circ\text{C}$) until DNA extraction and following quantification by PCR.

The 0.5 m² test plots were situated along the boardwalk for the autonomous field flux robot (FFR), which was used to monitor the N₂O emission.

Fig S9. Field plot experiments. The top ~10 cm of the soil was first loosened by a cultivator (Panel A). After evening out the surface with a rake, we placed a 0.7 X 0.7 m frame onto the surface (Panel B), and poured digestate onto the surface within the frame (Panel C). One day later the top soil (0-10 cm) was harrowed by a hand held tool (Panel D), and N₂O emissions were monitored by the robot (Panel E).

2F Calculations of emissions and statistical analyses

Fig S10 Recorded signal of N₂O concentration from a single measurement with the FFR. The jagged shape of the curve is due to valves in the robot switching every 20 seconds, alternating circulating air from the left and right chamber through the laser instrument. The two straight lines are regression lines, the slopes of which is used to calculate the emission fluxes. The difference in the steepness of the regression lines is caused by a difference between the fluxes from the soils the chambers have been lowered onto.

Emissions: From the slope of the N₂O regression lines the flux of N₂O is calculated by the equation

$$q_{N_2O} = \frac{10^{-6} a h p}{RT},$$

where q_{N_2O} is the flux of N₂O (mol m⁻² s⁻¹), a is the slope of the regression line (ppm s⁻¹), h is the height (i.e., the volume divided by the ground surface area) of the chamber (m), p is the pressure (Pa), R is the universal gas constant (J mol⁻¹ K⁻¹) and T is the temperature (K).

For graphic presentation of the emissions, we used the Gaussian Kernel smoother (Hastie et al 2009) to plot floating averages for each treatment (continuous curves) together with individual measurements (as dots); Figs 2,4,5 in main paper.

Cumulative N₂O emissions over a period of time are approximated by using the trapezoidal rule on the estimated fluxes. ($\int q_{N_2O}(t) dt \approx \sum (q_{N_2O}(t_i) + q_{N_2O}(t_{i+1}))(t_{i+1} - t_i)/2$). This was done for each individual bucket and field plot.

The field plot experiment yielded paired data – six pairs (X_i, Y_i), $i = 1 \dots 6$, where X_i are cumulative emissions from plots treated with NNRB, and Y_i are cumulative emissions for control plots. This gives six ratios $R_i = X_i/Y_i$. Confidence intervals for the mean of the ratios, $1/6 \sum R_i$, for two time periods were made with a Student's t distribution (assuming that the ratios were normally distributed.). These confidence were similar to confidence intervals found by the Fieller method for ratios of paired data and also by simple nonparametric bootstrapping (Efron and Tibshirani, 1994).

Since the field bucket experiments did not yield paired data, flux reduction statistics are calculated as ratios of means, rather than means of ratios, of cumulative fluxes. Confidence intervals of these ratios were made by the Fieller method for unpaired data (Motulsky, 1995, p 285) and by simple nonparametric bootstrapping (the results were similar). The 95% coverage of the Fieller confidence intervals were tested by numerical simulations and a bootstrap-calibration of the confidence level was made, with negligible effects on the confidence intervals.

Commented [P3]: Clarify - frequency of measurement with each 20 sec interval - that two different level are from different treatments or reps? Correct?

2G Tracing CB-01 in digestate and soil

To measure growth of CB-01 in digestate, and its survival in soil, we used quantitative PCR (qPCR), with primers that are specific to *Cloacibacterium* strains, developed by Allen et al. (2006). The primers target the complementary parts of the following sequence of the 16S rRNA gene: 5'-TATTGTTTCTTCGGAAATGA (Cloac-001f) and 5'-ATGGCAGTTCTATCGTTAAGC (Cloac-001r).

DNA was extracted with the DNeasy PowerSoil Pro Kit (Qiagen) according to the manufacturers protocol, except for the first step: Bead beating of the cells was done at 4.5 m s⁻¹ for 45 s in a FastPrep-24™ (MP Biomedicals, LLC, CA, USA), instead of a vortex. To measure the concentration of DNA in the extract, we used a Qubit dsDNA HS Assay Kit (Thermo Fisher Scientific, USA). The number of CB-01 16S gene copies in extracted DNA was quantified using a CFX96 Touch™ Real-Time PCR Detection System (Bio-Rad, USA), running for 15 min at 95 °C followed by 40 cycles of denaturation (30 s at 95 °C), annealing (30 s at 55 °C) and elongation (45 s at 72 °C). The master mix contained 0.2 μM of each primer (Cloac-001f and Cloac-001r), and 1x HOT FIREPol® EvaGreen® qPCR Supermix (Solis BioDyne).

For calibration, we used DNA extracted suspensions of washed cells containing 10³, 10⁴, 10⁵, 10⁶, 10⁷ and 10⁸ cells mL⁻¹, resulting in 2.4*10¹-2.4*10⁶ 16S templates per PCR tube (taking dilution into account, and the fact that each genome of CB-01 contains three 16S gene). To enable the use of the Cq values to estimate copy numbers, we used Generalized Reduced Gradient Solver in Excel to fit the model (equation 1) to the data:

$$N = \frac{N_T}{(2 \cdot e)^{Cq}} \quad (1)$$

where **N** is the initial number of 16S templates in the PCR tube, **N_T** is the number of amplicons per tube needed for signal detection (above background), **e** is the efficiency of the PCR amplification, and **Cq** is the number of cycles needed for detection of a signal. The fitted parameters were **N_T = 7.68*10¹⁰ copies per tube and e = 0.85 (85 % efficiency)**.

An independent dataset was provided by running qPCR with the same primers on extracted DNA from suspensions of unwashed CB-01 cells (in nutrient broth) with densities 10⁴, 10⁵, 10⁶, 10⁷ and 10⁸ cells mL⁻¹. The log₁₀ values of cell densities estimated by the Cq values were on average 104 % of the expected value, with a standard deviation of 6 %.

When using qPCR to estimate the CB-01 abundance in soil and digestate, inhibition of the polymerase can result in too high Cq numbers, hence resulting in underestimation of the gene abundance (Lim et al. 2016). To investigate this, we spiked the different soils and the digestate with 10⁹ CB-01 cells g⁻¹ soil dry weight and mL⁻¹ digestate, respectively, extracted DNA from 0.2 g soil and 0.2 mL digestate, eluted to a 50 μL DNA solution for each material, which was then diluted in 10-fold steps from 0 (undiluted) down to 1/10⁷. The results (**Figure S10**) show a reasonable fit between model (predicted) and measured Cq values for all materials if diluting the extracted DNA to ≤ 1/10, except for the clay loam pH 6.13 soil, which required dilution to ≤ 1/100 to eliminate inhibition.

Fig S11 Inhibition of qPCR. Soils and digestate was spiked with CB-01 cells (10^9 cells g^{-1} soil dw and mL^{-1} digestate, respectively). DNA was extracted (0.2 g soil and 0.2 mL digestate was extracted, eluted in a 50 μL volume. From undiluted, and 10-fold dilutions of this, 2 μL were transferred to PCR tubes for amplification of CB-01 16S genes. The plot shows the Cq values plotted against dilution, together with the model $N = N_T / (2 * e)^{Cq}$ as parameterized (see text): $N_T = 7.68 * 10^{10}$ copies per tube and $e = 0$.

The result can be used to approximate the lower limit for detection of CB-01 in soils and digestate: A cautious upper limit for Cq values to be trusted is 40, i.e. 34 templates per PCR tube (equation 1). The polymerases were evidently inhibited by using undiluted DNA in the reaction (Fig S10), hence a 1/10 dilution of the extracted DNA is needed for all soils except soil I, for which 1/100 dilution is required. This means that the PCR tube can maximally be loaded with DNA from 0.8 mg soil (0.08 mg for soil I) and 0.8 μL digestate. This implies a limit of detection around $4.3 * 10^4$ templates g^{-1} soil ($4.3 * 10^5$ for soil I due to dilution to 1/100) and mL^{-1} digestate, or $1.4 * 10^4$ CB-01-genomes g^{-1} soil and mL^{-1} digestate (since the genome contains 3 copies of the 16S gene).

The real limit of detection for a CB-01 inoculum in soil and digestate could be higher than this, if indigenous genes are amplified with the primers. This was tested by running PCR on soil and digestates which had not been spiked with CB-01, along with analyzing spiked samples in various experiments. The results are summarized in Table S4. Since there were several tubes with a negative result ($Cq > 40$), average values cannot be calculated. A cautious judgment would be that the “background” PCR signal of the soil is $Cq = 39-38$, which is equivalent to 67-107 templates per PCR tube, or 21-36 CB-01-genomes per tube. For all soils except I, we used the Cq values for the PCR tubes loaded with 1/10 dilutions, which were thus loaded with DNA from 0.8 mg soil. For these, the background PCR signal is equivalent to $2.6-4.8 * 10^4$ CB-01 genomes g^{-1} , while 10 times higher for soil I (due to 1/100 dilution of the DNA from this soil). For digestate, the average Cq was 31.98 (Table S4), which means that the untreated digestate contains $3.2 * 10^6$ CB-01 16S-templates mL^{-1} , or $1.1 * 10^6$ CB-01 genomes mL^{-1} .

Table S4 PCR-results for soils and digestate without spiking with CB-01. The table shows the total number of samples analyzed for each soil and for the digestate, the number of samples with $Cq > 40$ (estimate not available), and the average for those < 40 . The results suggest that the background Cq values are > 38 , which implies that the background is $< 1.25 * 10^5$ 16S-templates g^{-1} , hence $< 4.2 * 10^4$ CB-01 genomes g^{-1} soil. The background value of digestate is a bit higher, with a Cq of 31.98, corresponding to about $3.2 * 10^6$ 16S-templates mL^{-1} , or $1.1 * 10^6$ CB-01 genomes mL^{-1} digestate.

Material	Total number of samples	Number of samples with $Cq < 40$	Average Cq for samples with $Cq < 40$
N: Neutral-pH clay loam	8	2	39.33
I: Intermediate-pH clay loam	12	7	39.48
L: Low-pH clay loam	3	1	37.78
S: Sandy silt soil	3	0	-
O: Organic-rich clay loam	3	2	39.8
Digestate	3	3	31.98

2H Survival of CB-01 in soil, laboratory incubation

A soil incubation experiment was designed to assess the survival of CB-01 in soil, vectored by digestate, under constant temperature and moisture conditions, and without any subsequent incorporation of digestate (thus contrasting the field bucket experiment, Fig 2 in main paper). CB-01 was first grown to $\sim 6 \times 10^9$ cells mL⁻¹ in autoclaved, aerated and pH-adjusted digestate (as for the field experiments). Neutral-pH clay loam soil (soil N **Table S2**) was portioned into a set of 50-mL Falcon tubes (9.4 g soil dry weight, moisture content = 0.5 mL g⁻¹ soil dry weight). To each tube, 4.2 mL sterile water and 0.85 g digestate (with CB-01) was dripped onto the soil. The tubes were stored in a dark moist chamber at 15 °C, with loose lids to allow exchange of air. Control tubes received only sterile water. At intervals, 2 replicate tubes were frozen (-20 °C) for quantification of CB-01 gene abundance by qPCR as described above.

2I Extrapolating to national emission reductions

We use the emissions quantified with the GAINS model (Winiwarter et al. 2018, Amann et al. 2011) for 2030 in Europe to estimate the possible reductions of the measure.

The experiments described in this paper demonstrate marked emission reductions on all soils tested, over extended periods. The strongest reductions have been seen for the initial N₂O peak immediately after fertilization, but NNRB has shown to remain active over a period of 90 days. Cumulative emissions over the whole period have been reduced by at least 41% (for clay loam soils), up to 95% reduction. We may disregard the case of smallest reduction as also the emissions from these soils are rather small, but the organic loam soils (55% reductions) need to be considered. Consistent with the uniform emission factor used in GAINS (from IPCC, 2006) of 1% of N applied to be emitted as N₂O for all conditions of crops, soil or type of fertilizer added, we also choose to apply a uniform reduction factor of 60% of emission reductions due to NNRB which we consider a conservative estimate. In Table S5, emission reductions are shown by EU country for 2030 if emissions from application of liquid manure only is reduced by 60%. This assumption is based on the understanding that liquid manure can easily be treated in biodigesters. Höglund et al. (2020) assume, for purpose of methane abatement, anaerobic digestion becomes profitable only for large agricultural entities of at least 100 livestock units. According to GAINS numbers, this concerns 70% of all farms in the EU, which more probably reflect liquid than solid manure systems, so the above estimate remains valid for the major fraction of liquid manure available. Indirect emissions as well as other soil emissions due to grazing, mineral fertilizer additions or application of farmyard manure (solid manure systems) have been left unchanged. Note that the GAINS model (in agreement with IPCC, 2006) does not account for potentially increased emissions due to dry periods or freeze-thaw cycles (the latter considered to potentially contribute as much as 17-28% to global soil emissions: Wagner Riddle et al. 2017) while covering increased emissions from cropping histosols.

Under these assumptions, total N₂O emissions from Europe decrease by 2.7% due to NNRB introduced. This figure is higher in countries that have a high share of liquid manure systems in their agriculture, hence for EU27 (27 EU member countries) the corresponding figure is 4.0%, if NNRB were used for all manure nitrogen applied from liquid manure systems.

If it were possible to extend the NNRB-technology, using solid manure and plant residues as substrates and vectors, we speculate emission reductions could be achieved for all mineral and natural fertilizer actively applied on fields. Ongoing work has shown that while *Cloacibacter* CB-01

Commented [P4]: Calculations are unclear. If 60% emission reduction is assumed for all anthropogenic ag soil emissions (based on reactive N added) then overall abatement should be 60%!

grows to high cell densities in plant residues, new strains which grow in manure have been enriched and isolated (unpublished results). Although further development will be needed to implement this, it is relevant to estimate their impacts. Applying NNRB also to these other substrates at the same reduction efficiency could decrease European emissions as well as EU27 emissions by about a quarter (24% and 23%, respectively). For agricultural emissions only, this means that roughly a third (31%) could be eliminated.

It needs to be pointed out that an emission reduction of 60% as derived here for NRB is much larger than emission reductions typically reported for N₂O abatement measures. E.g., GAINS assumes nitrification inhibitors to be able to reduce emissions by as much as 38%, and high-tech mechanical fertilizer saving technologies (“variable rate application”) to be able to save 24% of the emissions only (Winiwarter et al. 2018). Of note, the percent reduction of N₂O-emission by the NNRB-technology is plausibly unaffected by “variable rate application” and nitrification inhibitor, since the target for NNRB is to reduce the N₂O/N₂ product ratio of denitrification, while the two others target the concentration of NO₃⁻ and nitrification, respectively.

Table S5: Potential emission reductions as a consequence of implementing NNRB measures (anthropogenic emissions in kt N₂O per year projected for 2030)

	Total N ₂ O emissions Kt N ₂ O y ⁻¹	Reduction by NNRB kt N ₂ O y ⁻¹		% Reduction of total emissions		% reduction of agricultural emissions
		NNRB in liquid manure only	NNRB with all N- fertilizers	NNRB in liquid manure only	NNRB with all N- fertilizers	NNRB with all N- fertilizers
Albania	4.3	0.10	0.85	2.3%	20%	24%
Austria	13.4	0.80	2.99	5.9%	22%	34%
Belarus	44.7	0.20	8.29	0.4%	19%	23%
Belgium	23.0	0.96	5.11	4.2%	22%	33%
Bosnia-Herzegovina	3.9	0.12	0.75	3.0%	19%	25%
Bulgaria	13.6	0.09	4.44	0.6%	33%	40%
Croatia	6.0	0.12	1.67	2.0%	28%	34%
Cyprus	0.9	0.05	0.15	5.9%	16%	21%
Czech Republic	19.2	0.18	5.11	0.9%	27%	37%
Denmark	18.8	1.83	5.24	9.8%	28%	33%
Estonia	3.2	0.06	0.73	1.8%	23%	31%
Finland	17.6	0.45	2.93	2.5%	17%	25%
France	143.3	5.16	36.76	3.6%	26%	31%
Germany	132.5	7.46	30.09	5.6%	23%	31%
Greece	14.0	0.18	2.60	1.3%	19%	25%
Hungary	17.4	0.18	5.85	1.1%	34%	41%
Iceland	1.0	0.04	0.24	4.3%	24%	28%
Ireland	30.6	1.40	5.73	4.6%	19%	20%
Italy	60.0	3.08	13.28	5.1%	22%	33%
Kosovo	1.2	0.02	0.18	1.5%	14%	28%
Latvia	4.9	0.08	0.86	1.6%	17%	20%
Lithuania	13.2	0.30	2.58	2.2%	20%	22%
Luxembourg	1.2	0.05	0.18	4.3%	15%	26%
Malta	0.2	0.01	0.04	2.3%	17%	32%
Moldavia	2.6	0.03	0.58	1.0%	22%	28%
Montenegro	0.5	0.01	0.06	1.5%	11%	19%
Netherlands	36.0	2.69	5.28	7.5%	15%	23%
Northern Macedonia	1.9	0.04	0.29	2.0%	15%	25%
Norway	11.1	0.41	2.17	3.7%	20%	30%
Poland	81.3	2.32	20.65	2.8%	25%	33%
Portugal	11.4	0.32	1.91	2.8%	17%	23%
Romania	28.8	0.48	8.00	1.7%	28%	35%
Russia	291.4	2.22	79.37	0.8%	27%	33%
Serbia	10.7	0.31	2.78	2.9%	26%	37%
Slovakia	7.3	0.08	2.09	1.1%	29%	40%
Slovenia	2.4	0.16	0.46	6.8%	19%	31%
Spain	62.1	2.47	14.75	4.0%	24%	32%
Sweden	18.7	0.49	3.47	2.6%	19%	29%
Switzerland	9.3	0.57	2.01	6.1%	22%	31%

Turkey	109.4	0.81	23.45	0.7%	21%	28%
Ukraine	73.4	0.34	18.61	0.5%	25%	34%
United Kingdom	89.4	1.69	19.39	1.9%	22%	29%
All Europe (incl. non-European parts of Russia and Turkey)	1435.9	38.51	341.97	2.7%	24%	31%
sum EU27	781.0	31.46	182.97	4.0%	23%	31%

3 Effect of CB-01 on the soil microbiome

Microbial community composition was examined by amplicon sequencing of the 16S rRNA gene V3-V4 region. Purified DNA from soil samples was sent to Novogene Europe for amplification, library preparation and sequencing to generate 250 bp paired-end reads using the Illumina Novoseq platform. Reads, after primer removal, were processed using GHAP (V2.4), an in-house amplicon clustering and classification pipeline built around Usearch (V11.0.66) (Edgar, 2010), the RDP classifier (V2.13) (Wang et al. 2007) and locally written tools for generating OTU tables. Reads were processed using default quality control and trimming parameters. Clustering was performed at both 97% and 100% similarity to generate OTUs and zOTUs, respectively. The 16S rRNA gene sequence of *Cloacibacterium* sp. CB-01 (GCA_907163125) was then matched against the OTU and zOTU representative sequences using the Usearch `usearch_global` command at 97% similarity and 99% similarity, respectively, to determine which OTU and zOTUs circumscribe the *Cloacibacterium* sp. CB-01 inoculant. From visual inspection it appeared that two zOTUs (zotu45 and zotu611) may circumscribe *Cloacibacterium* sp. CB-01 due to shared abundance profiles and taxonomic classifications. To confirm that these two zOTUs both matched to *Cloacibacterium* sp. CB-01 the two representative sequences were BLAST searched (Altschul et al, 1990) against the *Cloacibacterium* sp. CB-01 genome, where it was observed that both zOTU sequences matched closely to two separate regions of the genome, presumably harbouring multiple slightly divergent copies of the 16S rRNA gene. To confirm, the two 16S rRNA genes from the *Cloacibacterium* sp. CB-01 genome were matched back against the zOTU representative sequences using the `usearch_global` command at 99% similarity where they matched to both zotu45 and zotu611, separately. Due to this zotu45 and zotu611 were combined for downstream analyses.

To assess the impact of the various treatments on the soil microbial communities alpha and beta diversity measures were calculated for microbial communities from all samples using the OTU tables generated above. OTU tables were first modified by removing the OTU circumscribing *Cloacibacterium* sp. CB-01 (OTU_27) before rarifying the tables to 72 846 reads per sample using the Usearch `otutab_rare` command. Shannon's (Shannon, 1948) and Simpson's (Simpson, 1949) diversity indices were calculated using the Usearch `-alpha_div` command and beta diversity measures calculated using the Usearch `-beta_div` command. Jaccard's dissimilarity measures (Jaccard, 1912) were then used to generate MDS plots using the Scikitlearn MDS module (Pedregosa, et al. 2011).

The beta diversity as shown by Jaccard's dissimilarity measures indicated that early during the soil incubation period there is greater between sample variation both within treatments as well as between soils treated with live CB-01 and those treated with water or dead CB-01. Indicating an effect of CB-01 on the soil microbial communities (Fig S11). This effect, however, disappears by the final time point where samples from live CB-01, dead CB-01 and water treated soils cluster together, suggesting the effect of live CB-01 on native soil microbial communities is transient and microbial soil communities are not affected in the longer term by live CB-01 addition. It should be noted that the effect over time throughout the experiment is also a much larger source of microbial community

variation than the addition of live CB-01 cells. Presumably due to disturbances to the soil from digging, sieving and packing of pots. Similarly, no systematic effects are observed on the alpha diversity of soil microbial communities throughout the experiment indicating that the CB-01 treatment does not reduce the complexity or evenness of soil microbial communities when added to soils with digestate organic matter as can be seen in the Shannon's and Simpson's diversity measures of samples taken throughout the experiment (Fig S12 and S13).

Fig S12 MDS plot based on community composition of soils inoculated with CB-01. This panel shows an MDS plot based on Jaccard's dissimilarity measures for microbial communities in the soil of the field bucket experiment treated with live CB-01 in digestate, killed CB01 in digestate and water treatment only.

Fig S12 Box and whisker plot showing the Shannon's diversity indices for soil microbial communities. This panel shows the Shannon's diversity indices for microbial soil communities sampled throughout the field bucket experiment treated with live CB-01 in digestate, killed CB-01 in digestate and water treatment only.

Fig S13 Box and whisker plot showing the Simpson's diversity indices for soil microbial communities. This panel shows the Simpson's diversity indices for microbial soil communities sampled throughout the field bucket experiment treated with live CB-01 in digestate, killed CB-01 in digestate and water treatment only.

4 Survival of CB-01 in the field plot experiment

Table S6 and Figure S14 show the estimated average abundance of CB-01 genomes (unit= 1000 g⁻¹ soil dry-weight) in the paired field plots (a-f), i.e. adjacent plots inoculated with live and heat-killed Cloacibacter. The number of cells added with the digestate was 4.3*10⁹ g⁻¹ soil dryweight. The genome abundance in the soil samples taken 9 days after fertilization are remarkably variable, suggesting a fast initial decline in two of the plots with live CB-01 (plot c&e), at rates comparable to the rapid initial decline in the laboratory incubation experiment (Figure 6 in the main paper), tentatively ascribed to protozoal grazing. The variability suggests a patchy distribution of protozoa within the field.

Of note, the genome abundance after 280 days in the plots with live CB01 are in some agreement with the first order decline rates estimated for the field bucket experiment: The decline in genome abundance from 4.3*10⁹ at time 0 to 1.5*10⁶ at time 280 days implies an average first order decay rate of 0.028 d⁻¹, or an average half life = 25 days.

Table S6. Estimated abundance of CB-01 genomes in the field plot experiment, 9 and 280 days after fertilization with CB-01 containing digestate (live or heat killed). The numbers are the average genome abundance based on qPCR analysis of 3 replicate samples for each individual plot. Unit is thousand genomes g⁻¹ soil dry-weight. Initial numbers of CB-01 (at time 0) was ~6*10⁹ cells g⁻¹ soil dry-weight, based on analysis of the digestate (Fig S7).

plot-pair	Plots with live CB-01				plots with heat-killed CB-01			
	Time 9 days		time=280 days		Time 9 days		time=280 days	
	avg	se	avg	se	avg	se	avg	se
average	698 007	194 735	2 331	1 891	7 064	3 446	70	33
b	Na*	-	3 510	1 321	98 195	90 261	203	17
c	1 531	533	652	469	424	196	160	79
d	471 431	189 461	1 229	650	58 446	29 613	176	14
e	2 298	1 619	814	531	43 173	27 419	81	44
f	608 617	599 881	677	447	599	281	191	82
average	356 377	149 141	1 536	471	34 650	16 092	147	23

*samples lost

Fig S14 Graphic presentation of the data in Tale S6. Standard error shown as vertical lines (n=3).

References:

- Allen TD, Lawson PA, Collins MD, Falsen E, Tanner RS (2006). *Cloacibacterium normanense* gen. nov., sp. nov., a novel bacterium in the family Flavobacteriaceae isolated from municipal wastewater. *International journal of systematic and evolutionary microbiology* 56: 1311-1316.
- Altschul SF, Gish W, Miller W, Myers EW, Lipman DJ (1990). Basic local alignment search tool. *Journal of molecular biology*, 215(3), 403-410.
- Amann M, Bertok I, Borken-Kleefeld J, Cofala J, Heyes C, Höglund-Isaksson L, Klimont Z, Nguyen B, Posch M, Rafaj P, Sandler R, Schöpp W, Wagner F, Winiwarter W (2011) Cost-effective control of air quality and greenhouse gases in Europe: Modeling and policy applications. *Environmental Modelling & Software* 26: 1489–1501. <https://doi.org/10.1016/j.envsoft.2011.07.012>
- Bakken LR (2021) Spreadsheet for gas kinetics in batch cultures: Kincalc. Researchgate (2021) doi:10.13140/RG.2.2.19802.23809
- Bakken LR, Olsen RA (1983) Buoyant Densities and Dry-Matter Contents of Microorganisms: Conversion of a Measured Biovolume into Biomass *Appl Env Micro* 45, 1188-1195. <https://doi.org/10.1128/aem.45.4.1188-1195.1983>
- Bergaust L, Shapleigh J, Frostegård Å, Bakken LR (2008) Transcription and activities of NO_x reductases in *Agrobacterium tumefaciens*: the influence of nitrite, nitrate and oxygen availability. *Environmental Microbiology* 10:3070-3081.
- Bergaust L, Mao Y, Bakken LR, Frostegård Å (2010) Denitrification Response Patterns during the Transition to Anoxic Respiration and Posttranscriptional Effects of Suboptimal pH on Nitrogen Oxide Reductase in *Paracoccus denitrificans*. *Appl Env Micro* 76:6387-6393
- Bergaust L, Bakken LR, Frostegård Å (2011) Denitrification regulatory phenotype, a new term for the characterization of denitrifying bacteria. *Biochem. Soc. Trans.* 39, 207–212; doi:10.1042/BST0390207
- Byers E, Bleken MA, Dörsch P (2021). Winter N₂O accumulation and emission in subboreal grassland soil depend on clover proportion and soil pH. *Environ. Res. Commun.* 3, 015001. <https://doi.org/10.1088/2515-7620/abd623>
- Cowan NJ, Famulari D, Levy P E, Anderson M, Bell MJ, Rees RM, Reay DS and Skiba UM (2014) An improved method for measuring soil N₂O fluxes using a quantum cascade laser with a dynamic chamber: dynamic chamber method *Eur. J. Soil Sci.* 65 643–52
- Curtin D, Campbell CA, Messer D (1996) Prediction of titratable acidity and soil sensitivity to pH Change. *J Env Qual* 25:1280-1284.
- Edgar RC (2010). Search and clustering orders of magnitude faster than BLAST. *Bioinformatics*, 26(19), 2460-2461.
- Efron B, Tibshirani RJ (1994). *An introduction to the bootstrap*. CRC press.
- Gao Y, Mania D, Mousavi SA, Lycus P, Arntzen M, Woliy K, Lindström K, Shapleigh JP, Bakken LR, Frostegård Å (2021) Competition for electrons favors N₂O reduction in denitrifying *Bradyrhizobium* isolates. *Environmental Microbiology* 23:2244-2259. doi: 10.1111/1462-2920.15404.
- Hallin S, Philippot L, Löffler FE, Sanford RA, Jones CM (2018) Genomics and ecology of novel N₂O-reducing microorganisms. *Trends Microbiol.* 26:43–55.

Hassan J, Bergaust LL, Wheat D, Bakken LR (2014) Low Probability of Initiating nirS Transcription Explains Observed Gas Kinetics and Growth of Bacteria Switching from Aerobic Respiration to Denitrification. *PLoS Comput Biol* 10(11): e1003933. doi:10.1371/journal.pcbi.1003933

Hassan J, Bergaust L, Molstad L, de Vries S, Bakken LR (2016a) Homeostatic control of nitric oxide (NO) at nanomolar concentrations in denitrifying bacteria – modelling and experimental determination of NO reductase kinetics in vivo in *Paracoccus denitrificans*. *Env Microbiol* 18:2964-2978.

Hassan J, Bergaust L, Qu Z, Bakken LR (2016b) Transient accumulation of NO₂- and N₂O during denitrification explained by assuming cell diversification by stochastic transcription of denitrification genes. *PLoS Computational Biology*. DOI: 10.1371/journal.pcbi.1004621

Hastie, T., Tibshirani, R., & Friedman, J. H. (2009). *The elements of statistical learning: data mining, inference, and prediction*. 2nd ed. New York, Springer. (chapt. 6)

Hein S, Simon J (2019) Bacterial nitrous oxide respiration: electron transport chains and copper transfer reactions. *Advances in Microbial Physiology*, 75:137-169

Hensen A, Groot TT, van den Bulk WCM, Vermeulen AT, Olesen JE, Schelde K (2006) Dairy farm CH₄ and N₂O emissions, from one square metre to the full farm scale. *Agriculture, Ecosystems and Environment* 112: 146–152

Höglund-Isaksson L, Gómez-Sanabria A, Klimont Z, Rafaj P, Schöpp W (2020). Technical potentials and costs for reducing global anthropogenic methane emissions in the 2050 timeframe –results from the GAINS model. *Environ. Res. Commun.* 2, 025004. <https://doi.org/10.1088/2515-7620/ab7457>

IPCC (2006) 2006 IPCC Guidelines for National Greenhouse Gas Inventories. Institute for Global Environmental Strategies, Hayama, Japan

Jaccard P. (1912) The distribution of the flora in the alpine zone *New phytologist*. 1912;11(2):37-50.

Jonassen KR, Hagen LH, Vick SHW, Arntzen MØ, Eijsink VGH, Frostegård Å, Lycus P, Molstad L, Bakken LR (2022a) N₂O-respiring bacteria in biogas digestates for reduced agricultural emissions. *The ISME Journal* 16:580-590. <https://doi.org/10.1038/s41396-021-01101-x>

Jonassen KR, Ormåsén I, Duffner C, Hvidsten TR, Bakken LR, HW Vick SHW (2022b) A dual enrichment strategy provides soil and digestate competent nitrous oxide-respiring bacteria for mitigating climate forcing in agriculture. *mBio* 13 e0078-22 <https://journals.asm.org/doi/10.1128/mbio.00788-22>

Kellermann R, Hauge K, Tjåland R, Thalmann S, Bakken LR, Bergaust L (2022) Preparation for Denitrification and Phenotypic Diversification at the Cusp of Anoxia: a Purpose for N₂O Reductase Vis-à-Vis Multiple Roles of O₂. *Appl Env Micr* : e01053-22 <https://doi.org/10.1128/aem.01053-22>

Kim DD, Han H, Yun MJ, Terada A, Lureni M, Yoon S (2022) Identification of nosZ-expressing microorganisms consuming trace N₂O in microaerobic chemostat consortia dominated by an uncultured Burkholderiales. *The ISME Journal* 16:2087-2098.

Lim NYN, Roco C, Frostegård Å (2016) Transparent DNA/RNA Co-extraction Workflow Protocol Suitable for Inhibitor-Rich Environmental Samples That Focuses on Complete DNA Removal for Transcriptomic Analyses. *Front. Microbiol.* 7:1588. doi: 10.3389/fmicb.2016.01588

- Lycus P, Bøthun KL, Bergaust L, Shapleigh JP, Bakken LR, Frostegård Å (2017) Phenotypic and genotypic richness of denitrifiers revealed by a novel isolation strategy. *The ISME Journal* 11:2219-2232. doi: 10.1038/ismej.2017.82.
- Lycus P, Soriana-Laguna, Kjos M, Richardson DJ, Gates AJ, Milligan DA, Frostegård Å, Bergaust L, Bakken LR (2018) A bet-hedging strategy for denitrifying bacteria curtails their release of N₂O. *PNAS* 115: 11820–11825. <https://doi/10.1073/pnas.1805000115>
- Molstad L, Dörsch P, Bakken LR (2007). Robotized incubation system for monitoring gases (O₂, NO, N₂O N₂) in denitrifying cultures. *J Microbiol Methods* 71:202–211. <https://doi.org/10.1016/j.mimet.2007.08.011>
- Molstad, L., Köster, J. R., Bakken, L., Dörsch, P., Lien, T., Overskeid, Ø., Utstumo, T., Løvås, D. & Brevik, A. (2014). A field robot for autonomous laser-based N₂O flux measurements. *EGU General Assembly*.
- Molstad L, Dörsch P, Bakken L.R (2016). Improved robotized incubation system for gas kinetics in batch cultures. *Researchgate*. 2016. <https://doi.org/10.13140/RG.2.2.30688.07680>
- Motulsky H (1995) *Intuitive Biostatistics*, 1995 - Oxford University Press, New York.
- Nadeem, S., Bakken, L. R., Frostegård, Å., Gaby, J. C. & Dörsch, P. (2020). Contingent Effects of Liming on N₂O-Emissions Driven by Autotrophic Nitrification. *Frontiers in Environmental Science*, 8: 8-16. doi: 10.3389/fenvs.2020.598513.
- Pedregosa F, Varoquaux G, Gramfort A, Michel V, Thirion B, Grisel O, et al. *Scikit-learn: Machine learning in Python*. *the Journal of machine Learning research*. 2011;12:2825-30.
- Qi C, Zhou Y, Suenaga T, Oba K, Wang G, Zhang L, Yoon S, Terada A (2022) Organic carbon determines nitrous oxide consumption activity of clade I and II nosZ bacteria: Genomic and biokinetic insights. *Water Research* 209, 11791.
- Qu Z, Bakken LR, Frostegård Å, Bergaust L (2016) Transcriptional and metabolic regulation of denitrification in *Paracoccus denitrificans* allows low but significant activity of nitrous oxide reductase under oxic conditions. *Environmental Microbiology* DOI 10.1111/1462-2920.13128
- Simon J (2021) Mitigation of laughing gas emissions by nitrous oxide respiring microorganisms. In *Enzymes for Solving Humankind's Problems: Natural and Artificial Systems in Health, Agriculture, Environment and Energy* In: Moura, J.J.G., Moura, I., Maia, L.B., Eds.; Springer International Publishing: Cham, Switzerland, 2021; pp. 185–211.
- Shannon CE (1948) A mathematical theory of communication. *The Bell system technical journal*. 27(3):379-423.
- Simpson EH (1949). Measurement of diversity. *Nature* 163:688.
- Suenaga T, Riya S, Hosomi M, Terada A (2018) Biokinetic Characterization and Activities of N₂O-Reducing Bacteria in Response to Various Oxygen Levels. *Front. Microbiol.* 9:697.
- Suenaga T, Hori T, Riya S, Hosomi M, Smets BF, Terada A (2019) Enrichment, Isolation, and Characterization of High-Affinity N₂O-Reducing Bacteria in a Gas-Permeable Membrane Reactor. *Environ. Sci. Technol.* 2019, 53, 12101–12112

van Spanning RJM, Richardson DJ, Ferguson SJ (2007) Introduction to the Biochemistry and Molecular Biology of Denitrification. In *Biology of the Nitrogen Cycle*. Bothe, H., Ferguson, S.J., and Newton, W.E. (eds). Amsterdam, the Netherlands: Elsevier, pp. 3–20.

Wagner-Riddle C, Congreves KA, Abalos D, Berg AA, Brown SE, Ambadan JT, Gao X, Tenuta M (2017). Globally important nitrous oxide emissions from croplands induced by freeze–thaw cycles. *Nature Geosci* 10, 279–283. <https://doi.org/10.1038/ngeo2907>

Winiwarter W, Höglund-Isaksson L, Klimont Z, Schöpp W, Amann M (2018) Technical opportunities to reduce global anthropogenic emissions of nitrous oxide. *Environ. Res. Lett.* 13 014011. DOI: 10.1088/1748-9326/aa9ec9

Wang Q, Garrity GM, Tiedje JM, Cole JR (2007). Naive Bayesian classifier for rapid assignment of rRNA sequences into the new bacterial taxonomy. *Applied and environmental microbiology*, 73(16), 5261-5267.

Yoon S, Nissen S, Park D, Sanford RA, Löffler FE (2016). Nitrous oxide reduction kinetics distinguish bacteria harboring clade I NosZ from those harboring clade II NosZ. *Appl Environ Microbiol.* 82:3793–800.

Referee #2 (Remarks to the Author):

Agricultural intensification increases nitrous oxide (N₂O) emissions, a climate active gas. As a culmination of prior laboratory-based accomplishments, the authors present an innovative strategy for increasing bacterial N₂O consumption in soil, potentially applicable for curbing N₂O emissions to the atmosphere. The concept is clearly presented and the data support of the conclusions. The demonstration of a successful bioaugmentation strategy to reduce harmful N₂O emissions from fertilized land has far-reaching implication for agribusiness and reaching greenhouse gas emission goals. Although additional work is needed prior to broad implementation, the work demonstrates technological feasibility, has groundbreaking character, and potential to transform agricultural land management globally. The new approach (i.e., bioaugmentation of agricultural soils with a specialized bacterium using a waste product as substrate and vector) embraces circularity and illustrates how innovative biotechnologies can promote the circular bioeconomy.

The script has been carefully prepared, and the data are robust and support the conclusions. The pertinent literature has been cited. Essential control experiments were performed but should be highlighted to emphasize that the data strongly support a cause-and-effect relationship (i.e., reduced N₂O emissions in response to bioaugmentation).

The specific comments below address mostly technical issues and comprehension. The authors should be able to address these concerns.

59: Most scientist would agree that bacterial denitrification is a major source of N₂O in soil agroecosystems. Chemodenitrification should be mentioned as another N₂O-generating process. The contributions of denitrifying fungi and ammonia-oxidizing microbes to N₂O emissions from soils are less certain. My suggestion is to mention the major processes without judging their relative contributions, as soil type and geochemistry will have impact.

61: Not all denitrifying bacteria have nosZ. Such microbes would also be net sources of N₂O.

65: Has microbial respiratory growth with nitric oxide as electron acceptor been conclusively demonstrated?

65-72: The authors argue that a denitrifying bacterium can lack 1-3 denitrification genes. Do the authors believe that a microbe only possessing nar/nap should be called a denitrifying bacterium? The authors' argument that soil denitrification is a community process is valid; however, their definition of what genes define a denitrifying bacterium is not.

83: N₂O reductase is found in diverse taxa, and a community, rather a population, governs the reduction process.

90: In addition to expression, enzyme kinetic features play a role.

95-98: Are the (environmental) conditions that influence electron flow known?

101: Consider a ferric iron-reducing bacterium harboring nosZ. In the presence of ferric iron and nitrite (the latter generated by other microbes), this ferric iron-reducing bacterium could fuel chemodenitrification and contribute to N₂O formation (and likely to N₂O reduction). Also, geochemical factors other than oxygen can impact NosZ activity. "Unconditional" is too strong.

Box: The authors argue that all bacteria with nosZ are N₂O-respiring bacteria, what is problematic. There are examples showing that bacteria expressing NosZ can reduce N₂O without energy

conservation (i.e., uncoupled from growth). The term respiration implies chemiosmotic energy conservation, and the authors should refine the definition of “non-denitrifying N₂O-respiring bacteria” accordingly. NNRB may reduce N₂O without respiratory energy conservation. A simple solution would be “non-denitrifying N₂O-reducing bacteria”.

121: Growth in digestate occurred under oxic conditions, without any selection pressure for denitrification, correct?

125: “16S” is jargon. Replace with “16S rRNA gene” throughout.

139-146: The authors define NNRB as non-denitrifying N₂O-respiring bacteria (Box 1). This definition includes strain CB-01. Unclear why NO kinetic data are needed for classification. The authors state that strain CB-01 respire N₂O but evidence for this claim is lacking. Data showing N₂O-dependent growth should be included as supplementary material. Note that growth with N₂O does not prove respiration, and additional experimental efforts would be needed to distinguish chemiosmotic energy conservation (i.e., respiration) versus substrate-level phosphorylation.

150: Based on thermodynamics, N₂O is a more favorable electron acceptor than oxygen.

157: The reader needs to learn about the medium composition, and the authors ought to specify the composition of the nutrient broth. What was the electron donor? If rich, undefined medium was used, N₂O could serve as an electron sink for the re-oxidation of reduced electron carriers.

Enhanced growth may not be a result of respiratory N₂O reduction.

213: Do the data shown in Figure S7 demonstrate exponential growth?

257: How were strain CB-01 cells inactivated?

290: Both, Figure 2 and Figure 4 show N₂O emission data following biostimulation and bioaugmentation with live/dead strain CB-01 cells. To avoid redundancy, the authors should consider presenting Figure 4 data in text form (i.e., move one figure to SI).

297-305: In prior work, the authors demonstrated that strain CB-01 does not reduce N₂O under acidic pH conditions and speculated that the bacterium forms biofilms in alkaline microniches (reference 22). This concept should be mentioned here and the prior work cited.

342: The authors recognize that the inoculated strain CB-01 may not compete with the native microflora (“inoculants are invariably found to die out rapidly”). Does strain CB-01 occur in soil microbiomes prior to inoculation (i.e., is this bacterium native to soils)? If not, the decline of a non-native microbe following fertilizer application (and associated N₂O production) may be a desirable trait. This issue deserves attention in the context of general (public) acceptability of the new biotechnology, and potential risks associated with the introduction of non-native species.

348: How many *Cloacibacterium* genomes have been sequenced? At least some soil samples yielded amplicons with the *Cloacibacterium*-targeted primers, suggesting this bacterium occurs in soils. Do all sequenced *Cloacibacterium* genomes contain *nosZ*?

390-396: The authors used 16S rRNA gene amplicon sequencing to assess the impact of bioaugmentation on the microbial community structure. They conclude the impact was transient and not lasting. What was the impact of digestate addition without bioaugmentation on the soil microbiome? The authors should probably discuss any safety concerns directly associated with *Cloacibacterium* strain CB-01. Has the potential pathogenicity of this bacterium been assessed?

401: Why are nitrification inhibitors mentioned here considering ammonia-oxidizing microbes are minor contributors to N₂O formation (line 60)? Should liming be mentioned?

426: Do soils emit most N₂O only during the crop growing season?

Supplementary Information

The entire SI document should be carefully reviewed, and misspellings corrected.

39: Do the authors mean bacteria harboring nosZII?

42: Should read "oxic" conditions. Change throughout. Processes can be aerobic or anaerobic, conditions are oxic or anoxic.

53: The authors ought to provide details about the composition of the growth medium. How were the serum bottles closed?

54: He-flushed?

67: Did the authors verify that cell integrity was maintained in distilled water?

Figure S3: The abbreviation for micro is μ , not u. Check for misspellings.

273: How was exponential growth determined? What electron acceptors became limiting?

352: The application of digestate without strain CB-01 resulted in transient bursts of N₂O emissions. What was the nitrogen loading of the digestate? What was the dominant form of nitrogen in the digestate?

Table S3: Please verify units (mass versus volume) for digestate and harmonize.

410: "due" is repeated.

Referee #3 (Remarks to the Author):

The authors of this very well written manuscript present compelling results that demonstrate the augmentation of a previously isolated nitrous-oxide reducing bacterium, *Cloacibacterium* 19 sp. CB-01, reduced the N₂O emissions from soil 50% - 90% depending on the soil type. This organism expresses the NosZII nitrous oxide reductase and lacks the ability to reduce nitrate or nitrite. It is thus identified as a non-denitrifying N₂O respiring bacterium (NNRB). Its cell yield on N₂O, is demonstrated to be higher than microorganisms with a NosZI enzyme, coming in at 85% of the aerobic growth yield. Remarkably the data collected also suggest that this organism is bet-hedging during the transition to N₂O consumption; that is not all the members of the population are expression NosZ.

They also conducted field plot experiments that showed bioaugmentation with this strain yielded lower N₂O emissions. The authors demonstrate that with the use of organic waste as an application matrix, that strain CB-01 had a half-life of 23 days in their bucket experiment, demonstrating significant survivability in the different soil matrices. They lastly determined that the European Union (EU) scale benefit of application to agricultural systems could result in a 5-20% reduction in N₂O emissions.

B. This work presents a comprehensive study of the demonstrated feasibility of reducing N₂O emissions from N fertilizer applied to agricultural fields using a bioaugmentation approach. To my knowledge no one else has demonstrated such a remediation approach to the N₂O emission problem.

C. This lab is known for conducting detailed studies in the area of N-cycle research. This study shows no short-comings in the validity of the approach, the quality of data or the detailed nature of the presentation. I am impressed with what was presented.

D. Statistics are appropriately applied throughout the manuscript. I only note that some of the supplemental figure captions fail to define the error bars included in the Figure.

E. The conclusions that bioaugmentation by strain CB-01 to a variety of soil types, including those with relatively low pH, will yield consistently lower N₂O emission rates is robust. All of the data presented support these conclusions.

F. Suggestions and Observations: Although the authors do introduce the issue of how increased fertilization has led to an increase in N₂O emissions in agriculture, I believe they could further emphasize the significance by noting that the amount of fixed N on terrestrial earth has doubled since the development of the Haber-Bosch process. That is over all earth history we have never had this much fixed N. Thus, although N₂O generation and consumption has occurred over earth history, the flux potential has never been greater and organisms have not been naturally adapted to this situation. In my opinion, a sentence or two on this would provide an additional reason that some sort of bioaugmentation is needed to reduce the increased release of N₂O.

Figure 2 and Figure 5 seem to show that there is a diurnal temperature influence on the N₂O flux. It would seem that strain CB-01 might be directly influenced by this ΔT . What is the potential effect of daily temperature variation on the N₂O reduction rate? Since the V_{max} was determined at a fixed temperature, it would not apply over the observed temperature range that occurs in the soil which seems to shift by about 20° C on a daily basis. I also note in Figure 5 that the mean T is lowering as time goes on. How is strain CB-01 impacted by lowering seasonal temperatures? Does N₂O reduction cease at some minimal T? I note that the temperature reaches 0° C in Figure 5.

Figure 4 is not referenced in the text.

G. The reference list is thorough and comprehensive. There were several recent papers that I had not been aware of. I did note that one reference had been updated. Pessi, I.S., Viitamäki, S., Virkkala, AM. et al. In-depth characterization of denitrifier communities across different soil ecosystems in the tundra. *Environmental Microbiome* 17, 30 (2022). <https://doi.org/10.1186/s40793-022-00424-2>

H. The manuscript is very well written. Its message is very clear and the flow of the text is very logical. It is not overly verbose and therefore makes it very easy for any reader to comprehend what is being presented without reducing the rigor of the scientific approach used.

Reviewed by: Robert Sanford

Referee #4 (Remarks to the Author):

A. This manuscript contains a partial literature review and experimental evaluation of a microbial strain, CB-01, to be utilized as a soil inoculant to control N₂O emissions across European soils. However, the findings are contradictory and inconclusive as presented.

B. This is an original and significant research area into a microbial isolate that respire N₂O but lacks the ability to reduce nitrate and nitrite. These microorganisms are well known and have other types

of metabolism. They are not restricted to N₂O consumption. The authors indicate that these microbes should "always" be a sink for N₂O under low oxygen, but this is not true as other activities are usually present and regulated based on factors beyond O₂ availability.

C. The approach involves lab and field examinations of high density cultures of the strain *Cloacibacterium* sp. CB-01 cultivated in sludge digestate. The experiments show moderate and transient N₂O consumption over time based on O₂ and N₂O availability. The N₂O consumption activity of the CB-01 strain was evaluated against other microorganisms and shown to have a moderate V_{max} and low K_m. It showed a strong "bet-hedging" activity whereby only a small subset of cells showed N₂O consumption activity, suggesting that this is a weak activity for the cells. It is unknown how this feature contributes to the role the authors are claiming - that this strain could be a strong mitigator of N₂O emissions in all soils and throughout the growing season. The data and the author interpretation are contradictory.

D. There are controlled laboratory and field applications, but the authors did not use other microbial strains as positive or negative controls. They also did not control for digestate addition as an independent variable or the effect of digestate inoculation on soil pH. They only reported the many variables were uncontrolled in their field tests.

E. The conclusions of the study are contradictory in several sections. The physiology and relative activity of strain CB-01 indicates that it will not be effective at reducing N₂O emissions from soils, yet the authors suggest the opposite. The data are far too preliminary and many variables were uncontrolled, suggesting low reliability of the main conclusion.

F. Suggested improvements for the text: The statement that "bacteria equipped only with the gene for N₂O reduction will be unconditional net sinks for N₂O whenever deprived of O₂" is overly simplistic and neglects to mention the other N-metabolism pathways and conditions that occur under low O₂ for these same bacteria. For instance, many bacterial strains that encode NosZII can also perform DNRA, and both occur under O₂ deprivation, but the relative expression is dependent on other variables aside from O₂. Furthermore, conditions like copper deprivation will prevent N₂O removal, even under O₂ deprivation. This statement should be removed, revised, or followed by conditional statements.

The "sensus stricto" definition provided for NNRB is misleading as the presence of NO reductases suggest the possibility that the microbe can generate N₂O from. NRB that lack NirK/S can still generate N₂O using NOR enzymes or other N₂O-generating processes (e.g. cyt P460, cryptic nitrite reductase, globins, etc.). They can also have NosZ downregulated in favor of other processes (e.g. DNRA) that will prevent them from acting as N₂O sinks. Thus, the NNRB as defined are not always net sinks for N₂O, and their definition should thus be qualified.

Figure 1: The kinetic parameters for N₂O reduction imply that the microbe would need to be present at a high density and can only mitigate relatively high concentrations of N₂O due to its low affinity, impairing its utility in soils that are lower emitters of N₂O. This figure also seems to negate the potential for this strain in mitigating N₂O emissions universally in soil. The values reported in Table S5 are modelled based on the few experiments from this study under ideal conditions of high-

density growth. While it is nice to see projected emissions reductions across Europe, there are a number of variables (regulation, other microbes, "bet-hedging" activity model, type of crop, physicochemical parameters) that will result in substantial error bars on these values.

The idea of growing CB-01 to high enough density throughout European soils to mitigate N₂O emissions does not take into account the consequences to the ecology of the soils systems. Table S5 did not consider AMF or variability in rhizosphere microbiota. Would encouraging resilient high-density growth of a foreign microbe be a positive factor for preserving other soil services (e.g. AMF, rhizosphere microbiome, nutrient cycles, etc.)? What are the possible unintended consequences of this proposal? The authors should have a section with caveats for soil engineering at this magnitude.

Problematic statements: "Given the number of CB-01 cells added with the digestate (6.6×10^{13} cells m⁻² soil surface), and the $V_{max} = 0.6$ fmol N₂O cell⁻¹ h⁻¹ (Fig S2), the potential N₂O-reduction rate, if all the added CB-01 cells were respiring N₂O at maximum rate, is 1.1 g N₂O-N m⁻² h⁻¹"

According to the bet-hedging model, only a subset of cells can respire N₂O, so how can this calculation be accurate?

The peak N₂O-flux 1-2 days after fertilization was reduced by ~85 mg N₂O-N m⁻² h⁻¹, which is ~8 % of the estimated potential.

Estimated potential based on what?

"For the subsequent peaks of N₂O-flux, the apparent N₂O-respiration by CB-01 (i.e. the reduction of the flux) was ≤ 4 mg N₂O-N m⁻² h⁻¹, which is ≤ 0.36 % of the initial potential."

This measurement does not seem to indicate that these microbes are effective at reducing N₂O to a large extent. If both reductions of estimated potential are taken into account, then far less than 92% N₂O emissions control is possible.

"This decline in apparent N₂O-respiration by CB-01 was plausibly a result of two factors: a gradually declining rate of N₂O provision by the indigenous microbiome, and a gradually declining number of CB-01 cells."

Doesn't this data indicate that CB-01 cells are not effective at controlling N₂O emissions if they are not able to take up N₂O and if they decline in number after the first N₂O peak? It seems the authors validate that CB-01 will be a weak controller of N₂O following the initial high emission peak in the paragraph following the one above.

The experiments with different soils, digestate, and CB-01 were not a good control for pH since the digestate increased the pH. Thus, the effect of CB-01 alone was not measurable independent from the digestate.

The experiments from Fig. 5 seem to show that CB-01 is not effective over a long period of time. Even if the PCR experiment showed persistence of CB-01 genomes, their activity does not seem sustainable. The further caveats of differences between lab and field, soil moisture effects, and digestate application are additional complications that cast a shadow over the claimed benefits of CB-01 in mitigating N₂O flux. These problems should be addressed prior to making broad-based claims regarding the efficacy of this microbe for N₂O mitigation.

Analysis of the microbiome only considered other bacteria and maybe archaea, but not fungi. This is a major oversight.

The calculations of national emission reductions are missing essential information. How did the authors assume that 60% emission reductions was a conservative estimate? Was this emission reduction from the first pulse? Based on annual reductions? Based on reduction from a single application of CB-01? Based on reduction from a single amendment of digestate? Based on reduction in moist or dry soils? Based on what pH regime? The authors say that this is NRB (shouldn't this be NNRB?) being introduced and applied to all liquid manure systems. So are these estimates only for soils that receive liquid manure? What about the other application issues mentioned above?

The future development section does not consider the rapid decline in activity of CB-01 over time or the expression in a sub-set of the applied population. Thus, the statement that CB-01 application can curb N₂O emissions throughout an entire growth season does not seem accurate based on the results from this study. The call for future NNRB strains with different, yet unknown, characteristics validates that this technology is not well understood and cannot be presented at this time as a viable means to reduce N₂O emissions.

Fig. S4 -- there seems to be an extra "N₂O" in the legend for panel A

Fig. S5 -- based on the bet-hedging model for N₂O consumption demonstrated for this microbe, expression of NosZ is both transient and only for in a sub-population of cells. It is not understandable, then, how these physiological features would enable sustainable N₂O reduction in situ and across land-landscape scale in a predictable fashion. The authors should explain why bet-hedging activity is a positive feature for broad scale N₂O consumption.

G. References for N₂O emissions from soils, global sources of N₂O, and denitrifier genetics and biochemistry are outdated and should be updated.

H. Clarity and context: The rationale for this study is sound and the mini-literature review in the beginning is compelling. However, the stated major outcome -- that CB-01 can mitigate N₂O emissions from all soils over full growing seasons -- is not substantiated by the data.

Author Rebuttals to Initial Comments:

Response to referee comments on Hiis et al., Unlocking bacterial potential: A solution to farmland N₂O emissions

Here we provide a point-by-point response to referees' comments (red text). Where needed, we refer to publications, which are listed under each paragraph.

Referee #1 (Remarks to the Author):

Summary of results

The manuscript reports on a set of interlinked laboratory and field experiments to assess the capacity for introducing a bacteria inoculum with a high capacity to reduce N₂O to N₂, thereby reducing emissions of a potent greenhouse gas from agricultural soils. Full growing season field experiment results are used to extrapolate the potential value for abating soil N₂O emissions at EU scale, by using this (or similar) species of bacteria together with a waste-derived organic amendment, as a management practice.

Originality and significance

Both originality and significance are very high. Of ALL anthropogenic greenhouse gas emissions, one can make an argument that N₂O emissions are THE most difficult to effectively mitigate. The technology outlined in the ms suggests a substantially greater abatement potential, compared to existing interventions, and thus has enormous relevance for environmental and land use policy and management. What is particularly impressive is the degree of detail and thoroughness of the research going from the cellular level, in terms of gene expression and specific metabolic potentials, to field-scale biogeochemical investigations of gas flux rates with and without the inoculum as a function key biogeochemical and biophysical process controls, at the ecosystem scale, for different soils.

Methodology, data and statistics

As stated above, I find the methodology to be very comprehensive and well-executed and have no suggestions for improvements in the statistical analysis or presentation of the data.

From what I can tell the main conclusions are very robust and well supported. The one major suggestion for improvement, aside from some narrative text revisions to improve clarity, is to provide additional explanation for the country- and EU-scale of estimated abatement potential (based on data in Table S5. As described in the supplemental, country-scale emissions (as reported in national inventories), are shown based on a simple emission factor model of 1% of applied N to soil. The baseline emissions (2nd column totals) are not broken out by subsurface category so it is unclear if they represent only ag soil N₂O, or all agricultural N₂O emissions, or ALL N₂O emissions (including industrial, transportation, etc) for the country. Using the 1% emission factor model (GAINS) implies that the emissions are from agricultural soils only. If that is the case, then the assumption of a 60% abatement by use of non-denitrifying N₂O-respiring bacteria (NNRB) inoculum for all N amendments

to soil should yield a much higher reduction than the overall 31% emission reduction that is estimated.

Response: Data used in Table S5 (now in Extended Data Table 2) allow to quantify both the total anthropogenic emissions of a country as well as its agricultural emissions. The 60% abatement is considered to apply to agricultural soil emissions caused by anthropogenic N inputs only. Other categories of agricultural emissions (especially indirect emissions i.e., agricultural N lost as ammonia or leached as nitrate and contributing to N₂O emissions elsewhere, but also manure management emissions and emissions from histosols), at least in this estimate, are assumed not to be affected. Hence the reduction is not 60% but only 31% of agricultural emissions, or 23% of total anthropogenic emissions. In the updated Supplementary Information we make sure to carefully explain this.

In case it will be useful to the authors, I include some grammatical edits and/or suggested edits to improved clarity in the attached (anonymized) pdfs

Response: Thanks for this, manuscript revised accordingly.

Referee #2 (Remarks to the Author):

Agricultural intensification increases nitrous oxide (N₂O) emissions, a climate active gas. As a culmination of prior laboratory-based accomplishments, the authors present an innovative strategy for increasing bacterial N₂O consumption in soil, potentially applicable for curbing N₂O emissions to the atmosphere. The concept is clearly presented and the data support of the conclusions. The demonstration of a successful bioaugmentation strategy to reduce harmful N₂O emissions from fertilized land has far-reaching implication for agribusiness and reaching greenhouse gas emission goals. Although additional work is needed prior to broad implementation, the work demonstrates technological feasibility, has groundbreaking character, and potential to transform agricultural land management globally. The new approach (i.e., bioaugmentation of agricultural soils with a specialized bacterium using a waste product as substrate and vector) embraces circularity and illustrates how innovative biotechnologies can promote the circular bioeconomy.

The script has been carefully prepared, and the data are robust and support the conclusions. The pertinent literature has been cited. Essential control experiments were performed but should be highlighted to emphasize that the data strongly support a cause-and-effect relationship (i.e., reduced N₂O emissions in response to bioaugmentation).

Response: To assess the effect of *Cloacibacterium* sp. CB-01 on N₂O emission from soil, we fertilized the soil with digestate in which CB-01 had been grown to high cell densities, and as a control treatment we fertilized with the same digestate, in which CB-01 had been killed by a mild heat treatment. The implicit reasoning being that to assess the effect of CB-01 metabolism on N₂O-emissions, the control treatment must receive identical amount and quality of organic fertilizer as the treatment with live CB-01. In the revised manuscript, we have added a couple of sentences explaining this rationale.

The specific comments below address mostly technical issues and comprehension. The authors

should be able to address these concerns.

59: Most scientist would agree that bacterial denitrification is a major source of N₂O in soil agroecosystems. Chemodenitrification should be mentioned as another N₂O-generating process. The contributions of denitrifying fungi and ammonia-oxidizing microbes to N₂O emissions from soils are less certain. My suggestion is to mention the major processes without judging their relative contributions, as soil type and geochemistry will have impact.

Response: Agree; done.

61: Not all denitrifying bacteria have nosZ. Such microbes would also be net sources of N₂O.

Response: Good point, and our solution is to write that “DB can be either sinks, sources, or both...”.

65: Has microbial respiratory growth with nitric oxide as electron acceptor been conclusively demonstrated?

Response: Yes and no (!):

- Garrido-Amador et al.¹ enriched and characterized organisms which grew by formate as electron donor, and NO as the sole available electron acceptor.
- We ran anaerobic coculturing nir- and nor- mutants of *Paracoccus denitrificans* (fed with nitrate), and found growth by both mutants².

That said: neither of these studies prove that the organisms grow by NO-reduction to N₂O alone, because the bacteria in both studies reduced N₂O further to N₂, thus the experiments are no direct proof of charge separation (pmf generation), hence growth, by the electron flow to NO-reductase. On the other hand, electrogenic nature of the electron flow to NOR is plausible, given that the electrons are delivered from the BC1 complex (via cytochromes).

¹ Garrido-Amador et al. (2023) Enrichment and characterization of a nitric oxide-reducing microbial community in a continuous bioreactor. Nature Microbiol 8:1574-1586 <https://doi.org/10.1038/s41564-023-01425-8>

² Bergaust L, Bakken LR, Spiro S et al., (in prep)

65-72: The authors argue that a denitrifying bacterium can lack 1-3 denitrification genes. Do the authors believe that a microbe only possessing nar/nap should be called a denitrifying bacterium? The authors' argument that soil denitrification is a community process is valid; however, their definition of what genes define a denitrifying bacterium is not.

Response: Denitrifying organisms *sensu stricto* are defined by their ability to grow by anaerobic respiration using nitrite as a terminal electron acceptor (converting it to Nitric oxide). The “denitrification pathway” is commonly defined as the stepwise reduction NO₃⁻→NO₂⁻→NO→N₂O→N₂, catalyzed by the four enzymes Nar/Nap, NirS/K, Nor and NosZ, sustaining anaerobic respiration and growth. Many organisms are equipped with truncated denitrification pathways, lacking 1-3 of the four genes coding for the full pathway. Any organism lacking NirS/K is by definition not a denitrifying organism *sensu stricto*. Our text could be taken to suggest otherwise, and we have revised it to avoid this confusion:

1. A sentence has been added at the end the second paragraph of “Mitigation”: Of note, an organism with a truncated denitrification pathway lacking nirK/S is not a denitrifying bacterium *sensu stricto*.
2. The text in the Box (and adjacent text) has been revised accordingly.

83: N₂O reductase is found in diverse taxa, and a community, rather a population, governs the reduction process.

Response: Our phrasing was clumsy, and we have rephrased this sentence to make it clear that it is the abundance of all NRB-populations (“Increasing the abundance of N₂O-respiring bacteria (NRB, BOX 1) could decrease the emission of N₂O (Simon 2021)”).

90: In addition to expression, enzyme kinetic features play a role.

Response: This is correct, although it has limited implications (in comparison to regulation) in determining whether the organism is a net source or sink for N₂O. N₂O is a free intermediate, as is NO, and we have shown earlier¹ that the steady state concentration of NO depends the organism’s rate of NO-production, and on V_{max}/K_m of the enzyme reducing the intermediate (NOR in this case). While K_m is enzyme-characteristic, V_{max} depends both on the turnover rate of the enzyme-substrate complex (k_{cat}) and the amount of enzyme produced. The latter depends on regulation.

We found it difficult to include enzyme kinetics as a factor in the brief summary here. The enzyme kinetics is thoroughly discussed later in the manuscript (Fig 1 and adjacent text).

¹ Hassan J, Bergaust L, Molstad L, deVries S, Bakken LR (2016) Homeostatic control of nitric oxide (NO) at nanomolar concentrations in denitrifying bacteria – modelling and experimental determination of NO reductase kinetics in vivo in *Paracoccus denitrificans*. *Environmental Microbiology* 18(9), 2964–2978 doi:10.1111/1462-2920.13129

95-98: Are the (environmental) conditions that influence electron flow known?

Response: Some are, such as the concentrations of nitrate and nitrite, NO, and copper¹, while the list will probably be longer as we increase our understanding of the transcriptional regulation of the denitrification pathway.

¹ Gaimster, H, Alston M, Richardson DJ, Gates AJ, Rowley G (2018) Transcriptional and environmental control of bacterial denitrification and N₂O emissions. *FEMS Microbiology Letters*, 365, 2018, fnx277. doi: 10.1093/femsle/fnx277

101: Consider a ferric iron-reducing bacterium harboring nosZ. In the presence of ferric iron and nitrite (the latter generated by other microbes), this ferric iron-reducing bacterium could fuel chemodenitrification and contribute to N₂O formation (and likely to N₂O reduction). Also, geochemical factors other than oxygen can impact NosZ activity. “Unconditional” is too strong.

Response: We assume that the referee has the paper by Onley et al.¹ in mind. They elegantly demonstrate that the iron reducing nitrate-ammonifying *Anaeromyxobacter dehalogenans* can denitrify despite the lack of nirK/S, when grown axenically with Fe³⁺. It produces NO₂⁻ and Fe²⁺, which react (chemically) to form N₂O, which the organism then reduces to N₂, catalyzed by its N₂O reductase. We have moderated the statement, and referred to Onley et al.¹.

¹ Onley JR, Ahshan S, Sanford RA, Löffler FE (2018) Denitrification by *Anaeromyxobacter dehalogenans*, a Common Soil Bacterium Lacking the Nitrite Reductase Genes *nirS* and *nirK*. *Appl Env Microbiology* 84: e01985-17 <https://doi.org/10.1128/AEM.01985-17>

Box: The authors argue that all bacteria with *nosZ* are N₂O-respiring bacteria, what is problematic. There are examples showing that bacteria expressing *NosZ* can reduce N₂O without energy conservation (i.e., uncoupled from growth). The term respiration implies chemiosmotic energy conservation, and the authors should refine the definition of “non-denitrifying N₂O-respiring bacteria” accordingly. NNRB may reduce N₂O without respiratory energy conservation. A simple solution would be “non-denitrifying N₂O-reducing bacteria”.

Response: We accept the point, even though we consider it to be a minor issue. We have resolved the issue by writing NRB: N₂O respiring bacteria.

121: Growth in digestate occurred under oxic conditions, without any selection pressure for denitrification, correct?

Response: No, the *enrichment culturing* in the study by Jonassen et al.¹ was anoxic, with N₂O as the sole electron acceptor, except for a small dose of O₂ consumed in the early phase of each batch cultivation. We have added this information in the revised manuscript.

¹Jonassen, K. R. et al. A Dual Enrichment Strategy Provides Soil-and Digestate-Competent Nitrous Oxide-Respiring Bacteria for Mitigating Climate Forcing in Agriculture. *mBio* 13, e00788-22 (2022).

125: “16S” is jargon. Replace with “16S rRNA gene” throughout.

Response: Done.

139-146: The authors define NNRB as non-denitrifying N₂O-respiring bacteria (Box 1). This definition includes strain CB-01. Unclear why NO kinetic data are needed for classification. The authors state that strain CB-01 respire N₂O but evidence for this claim is lacking. Data showing N₂O-dependent growth should be included as supplementary material. Note that growth with N₂O does not prove respiration, and additional experimental efforts would be needed to distinguish chemiosmotic energy conservation (i.e., respiration) versus substrate-level phosphorylation.

Response: We have demonstrated beyond any doubt that CB-01 grows by respiring N₂O (Extended Data Fig. 1, and in the main text). In terms of cell dry-weight per mole respired N₂O, CB-01 had higher growth yield than the canonical denitrifying bacterium *Paracoccus denitrificans*, while the two organisms had similar growth yield per mole O₂ (Extended Data Fig. 1 and its legend).

The energy conservation (as *pmf*) of anaerobic respiration in denitrifying bacteria is often assessed by comparing their aerobic and anaerobic growth yield per electron: Y_{eO_2} = cells (or g dry-weight) produced per electron to O₂ and Y_{eN_2O} = cells (or g dry-weight) produced per mole electrons to N₂O. For canonical denitrifying bacteria, such as *Paracoccus denitrificans*, Y_{eN_2O} is around 60% of Y_{eO_2} , and the reasons for $Y_{eN_2O} / Y_{eO_2} = \sim 0.6$ is well understood¹. For CB-01, Y_{eN_2O} was 85% of Y_{eO_2} . Such high anaerobic growth yield could not possibly be obtained by substrate level phosphorylation alone. In conclusion: the presented data leave little doubt regarding the ability of CB-01 to grow by anaerobic respiration, using N₂O as the terminal electron acceptor.

¹ van Spanning, R. J., Richardson, D. J. & Ferguson, S. J. Introduction to the Biochemistry and Molecular Biology of Denitrification. in *Biology of the Nitrogen Cycle* (eds. Bothe, H., Ferguson, S. J. & Newton, W. E.) 3–20 (Elsevier, 2007).

150: Based on thermodynamics, N₂O is a more favorable electron acceptor than oxygen.

Response: Yes, we are aware of that, and this is the reason why the observed high Y_{N_2O}/Y_{O_2} for CB-01 (0.85!) is not in conflict with the thermodynamics of the two reactions (N₂O- and O₂-reduction).

157: The reader needs to learn about the medium composition, and the authors ought to specify the composition of the nutrient broth. What was the electron donor? If rich, undefined medium was used, N₂O could serve as an electron sink for the re-oxidation of reduced electron carriers.

Enhanced growth may not be a result of respiratory N₂O reduction.

Response: We are sorry for not reporting this. The medium was nutrient broth, consisting of meat peptone and meat extract. This information is now reported in the revised paper and in Supplementary Information 1.

The growth yield per mole O₂ (g dry-weight mole⁻¹ O₂) for CB-01 was on par with the canonical denitrifying organism *Paracoccus denitrificans*, while the yield per mole N₂O (g dry-weight mole⁻¹ N₂O) was higher. Thus, there is little doubt that CB-01 obtained energy by respiration, plausibly sustained by oxidizing a variety of C-sources in the complex medium used (Nutrient broth).

213: Do the data shown in Figure S7 demonstrate exponential growth?

Response: This is explained in the legend: The rate of O₂-consumption increased exponentially during the first 24 hours. Thereafter, the growth rate declined gradually.

257: How were strain CB-01 cells inactivated?

Response: This is explained in the legend of Fig. 1 (heated to 70°C for 2 hours).

290: Both, Figure 2 and Figure 4 show N₂O emission data following biostimulation and bioaugmentation with live/dead strain CB-01 cells. To avoid redundancy, the authors should consider presenting Figure 4 data in text form (i.e., move one figure to SI).

Response: Yes, we thought about this, but find that Figure 4 is important to keep because it is the only experiment comparing different soils, and not the least: the panels do not only present emissions, but also temperature and soil moisture throughout.

297-305: In prior work, the authors demonstrated that strain CB-01 does not reduce N₂O under acidic pH conditions and speculated that the bacterium forms biofilms in alkaline microniches (reference 22). This concept should be mentioned here and the prior work cited.

Response: Done.

342: The authors recognize that the inoculated strain CB-01 may not compete with the native microflora (“inoculants are invariably found to die out rapidly”). Does strain CB-01 occur in soil microbiomes prior to inoculation (i.e., is this bacterium native to soils)? If not, the decline of a non-

native microbe following fertilizer application (and associated N₂O production) may be a desirable trait. This issue deserves attention in the context of general (public) acceptability of the new biotechnology, and potential risks associated with the introduction of non-native species.

Response: CB-01 plausibly originates from digestate¹. The qPCR results, using *Cloacibacterium*-specific primers, showed a “background” of templates in the pristine soil, amounting to ~10⁴ genomes per g soil dry-weight (Supplementary Information 5). This suggests that the soils contain small populations of organisms that are phylogenetically close to *Cloacibacterium*, but not necessarily identical.

¹ Jonassen, K. R. et al. A Dual Enrichment Strategy Provides Soil-and Digestate-Competent Nitrous Oxide-Respiring Bacteria for Mitigating Climate Forcing in Agriculture. *mBio* 13, e00788-22 (2022).

348: How many *Cloacibacterium* genomes have been sequenced? At least some soil samples yielded amplicons with the *Cloacibacterium*-targeted primers, suggesting this bacterium occurs in soils. Do all sequenced *Cloacibacterium* genomes contain *nosZ*?

Response: In previous work¹, we isolated two *Cloacibacterium* strains, with very similar genomes, both contained *NosZ*.

Regarding abundance in soil, see the former paragraph.

¹ Jonassen, K. R. et al. A Dual Enrichment Strategy Provides Soil-and Digestate-Competent Nitrous Oxide-Respiring Bacteria for Mitigating Climate Forcing in Agriculture. *mBio* 13, e00788-22 (2022).

390-396: The authors used 16S rRNA gene amplicon sequencing to assess the impact of bioaugmentation on the microbial community structure. They conclude the impact was transient and not lasting. What was the impact of digestate addition without bioaugmentation on the soil microbiome? The authors should probably discuss any safety concerns directly associated with *Cloacibacterium* strain CB-01. Has the potential pathogenicity of this bacterium been assessed?

Response: The investigation of the community structure included soil amended with digestate without live CB-01.

The potential “game-stoppers” for CB-01 are pathogenicity and antibiotic resistance genes (ARG). We searched the genome for genes for pathogenicity and ARG, found none. This information has now been added to the main text and in Supplementary Information 9.

401: Why are nitrification inhibitors mentioned here considering ammonia-oxidizing microbes are minor contributors to N₂O formation (line 60)? Should liming be mentioned?

Response: because nitrification inhibitors (NI) reduce N₂O emissions beyond their direct effect on N₂O-production by nitrification: High nitrification rates in soil induce denitrification by consuming O₂, thus inducing hypoxia¹. Retardation by NI reduces this nitrification-induced denitrification rates, hence N₂O emission from denitrification.

¹ Nadeem S, Bakken LR, Frostegård Å, Gaby JC, Dörsch P (2020) Contingent effects of liming on N₂O emission driven by autotrophic nitrification. *Front. Environ. Sci.* 8:598513. doi: 10.3389/fenvs.2020.598513

426: Do soils emit most N₂O only during the crop growing season?

Response: Indeed, most N₂O is released during the cropping season, often in close temporal

proximity to fertilizing events. But quantities also depend on the climate. In the northern climates, off-season can be substantial (1/5-1/3 of annual emissions) due to freeze-thaw induced emissions¹.

¹ Wagner-Riddle C, Congreves KA, Abalos D et al (2017) Globally important nitrous oxide emissions from croplands induced by freeze-thaw cycles. *Nature Geosci* 10, 279–283 (2017). <https://doi.org/10.1038/ngeo2907>

Supplementary Information

The entire SI document should be carefully reviewed, and misspellings corrected.

Response: SI is profoundly revised, and split up (Extended Data and Supplementary Information)

39: Do the authors mean bacteria harboring nosZII?

Response: As shown in Fig 1, we compare it with bacteria harboring NosZI as well as those harboring NosII.

42: Should read “oxic” conditions. Change throughout. Processes can be aerobic or anaerobic, conditions are oxic or anoxic.

Response: Thanks for this correction. The manuscript has been scrutinized and corrected throughout.

53: The authors ought to provide details about the composition of the growth medium. How were the serum bottles closed?

Response: The composition of the medium is now reported. The vials were closed as for any other experiments: crimp-sealed with butyl rubber caps. This information is included in the revised text.

54: He-flushed?

Response: Good idea. He-washed is a clumsy term. Changed to He-flushed.

67: Did the authors verify that cell integrity was maintained in distilled water?

Response: Yes, they remained intact.

Figure S3: The abbreviation for micro is μ , not u. Check for misspellings.

Response: Thanks for spotting this.

273: How was exponential growth determined? What electron acceptors became limiting?

Response: As stated in the legend, the rate of oxygen consumption increased exponentially throughout the first 24 hours, with an apparent growth rate = 0.12 h^{-1} ($r^2 = 0.99$), as written in the panel. The experiment referred to was oxic (oxygen was the only electron acceptor provided), hence oxygen became limiting.

352: The application of digestate without strain CB-01 resulted in transient bursts of N₂O emissions.

What was the nitrogen loading of the digestate? What was the dominant form of nitrogen in the digestate?

Response: As written, the secondary doses of digestate (without CB-01) were 4.6 L digestate m⁻². The digestate contained 1.71 g NH₄⁺-N L⁻¹ (reported above this: Line 231). Thus, the mineral N load of each dose was 7.2 g NH₄⁺-N m⁻².

Table S3: Please verify units (mass versus volume) for digestate and harmonize.

Response: The units are mL digestate and g soil dry-weight (now specified).

410: "due" is repeated.

Response: Corrected.

Referee #3 (Remarks to the Author):

The authors of this very well written manuscript present compelling results that demonstrate the augmentation of a previously isolated nitrous-oxide reducing bacterium, Cloacibacterium 19 sp. CB-01, reduced the N₂O emissions from soil 50% - 90% depending on the soil type. This organism expresses the NosZII nitrous oxide reductase and lacks the ability to reduce nitrate or nitrite. It is thus identified as a non-denitrifying N₂O respiring bacterium (NNRB). Its cell yield on N₂O, is demonstrated to be higher than microorganisms with a NosZI enzyme, coming in at 85% of the aerobic growth yield. Remarkably the data collected also suggest that this organism is bet-hedging during the transition to N₂O consumption; that is not all the members of the population are expression NosZ.

They also conducted field plot experiments that showed bioaugmentation with this strain yielded lower N₂O emissions. The authors demonstrate that with the use of organic waste as an application matrix, that strain CB-01 had a half-life of 23 days in their bucket experiment, demonstrating significant survivability in the different soil matrices. They lastly determined that the European Union (EU) scale benefit of application to agricultural systems could result in a 5-20% reduction in N₂O emissions.

B. This work presents a comprehensive study of the demonstrated feasibility of reducing N₂O emissions from N fertilizer applied to agricultural fields using a bioaugmentation approach. To my knowledge no one else has demonstrated such a remediation approach to the N₂O emission problem.

C. This lab is known for conducting detailed studies in the area of N-cycle research. This study shows no short-comings in the validity of the approach, the quality of data or the detailed nature of the presentation. I am impressed with what was presented.

D. Statistics are appropriately applied throughout the manuscript. I only note that some of the supplemental figure captions fail to define the error bars included in the Figure.

E. The conclusions that bioaugmentation by strain CB-01 to a variety of soil types, including those with relatively low pH, will yield consistently lower N₂O emission rates is robust. All of the data presented support these conclusions.

F. Suggestions and Observations: Although the authors do introduce the issue of how increased fertilization has led to an increase in N₂O emissions in agriculture, I believe they could further emphasize the significance by noting that the amount of fixed N on terrestrial earth has doubled since the development of the Haber-Bosch process. That is over all earth history we have never had this much fixed N. Thus, although N₂O generation and consumption has occurred over earth history, the flux potential has never been greater and organisms have not been naturally adapted to this situation. In my opinion, a sentence or two on this would provide an additional reason that some sort of bioaugmentation is needed to reduce the increased release of N₂O.

Response: We agree that this is an important backdrop, and have added a sentence to the introduction, and one reference¹.

¹ Canfield, D. E., Glazer, A. N. & Falkowski, P. G. The Evolution and Future of Earth's Nitrogen Cycle. *Science* 330, 192–196 (2010).

Figure 2 and Figure 5 seem to show that there is a diurnal temperature influence on the N₂O flux. It would seem that strain CB-01 might be directly influenced by this ΔT . What is the potential effect of daily temperature variation on the N₂O reduction rate? Since the V_{max} was determined at a fixed temperature, it would not apply over the observed temperature range that occurs in the soil which seems to shift by about 20° C on a daily basis. I also note in Figure 5 that the mean T is lowering as time goes on. How is strain CB-01 impacted by lowering seasonal temperatures? Does N₂O reduction cease at some minimal T? I note that the temperature reaches 0° C in Figure 5.

Figure 4 is not referenced in the text.

Response: Valid point. We have not investigated the temperature response of CB-01. Probably a good idea to explore this in more detail in a follow up study.

G. The reference list is thorough and comprehensive. There were several recent papers that I had not been aware of. I did note that one reference had been updated. Pessi, I.S., Viitamäki, S., Virkkala, AM. et al. In-depth characterization of denitrifier communities across different soil ecosystems in the tundra. *Environmental Microbiome* 17, 30 (2022). <https://doi.org/10.1186/s40793-022-00424-2>

Response: Thanks for this information. The reference has been updated.

H. The manuscript is very well written. Its message is very clear and the flow of the text is very logical. It is not overly verbose and therefore makes it very easy for any reader to comprehend what is being presented without reducing the rigor of the scientific approach used.

Reviewed by: Robert Sanford

Referee #4 (Remarks to the Author):

NB: We have added letters to each paragraph of Referee #4, to make it easier to do cross-referencing.

Of note, referee#4 repeatedly questions our evidence for N₂O-mitigation, without providing much of an explanation, beyond implicitly claiming that Cloacibacterium sp. CB-01 would be unable to mitigate N₂O emissions, given the observed bet-hedging. This forced us to inspect bet-hedging for CB-01 when growing in digestate. The results show that in digestate, all cells switched to N₂O-respiration!

This adds to the scientific impact of our paper: The majority of research groups searching for effective N₂O-reducing bacteria use the strains' phenotypes in laboratory media to predict their comparative strength as N₂O sinks under natural conditions. Our observation of the medium-dependent bet-hedging shows how wrong this can be. Great thanks to referee#4 for drilling down on this issue!

A. This manuscript contains a partial literature review and experimental evaluation of a microbial strain, CB-01, to be utilized as a soil inoculant to control N₂O emissions across European soils. However, the findings are contradictory and inconclusive as presented.

Response: We hope that our response to the more specific points (below) will help.

B. This is an original and significant research area into a microbial isolate that respire N₂O but lacks the ability to reduce nitrate and nitrite. These microorganisms are well known and have other types of metabolism. They are not restricted to N₂O consumption. The authors indicate that these microbes should "always" be a sink for N₂O under low oxygen, but this is not true as other activities are usually present and regulated based on factors beyond O₂ availability.

Response: We refer to Non-Denitrifying N₂O-respiring Bacteria (NNRB) as microbes that can respire N₂O (catalyzed by NosZ), but do not have the genes for the dissimilatory reduction of NO₂⁻ to NO (NirK/S), which is the defining step of denitrification *sensu stricto*, similar to other authors¹⁻³.

Several such organisms are also equipped with genes for Nitrate-Ammonification (NA), which is an alternative pathway supporting respiratory metabolism in anoxia by reducing NO₃⁻ to NO₂⁻ by dissimilatory nitrate reductases, and NO₂⁻ to NH₄⁺ by a single six-electron transfer reaction (NirB or nrfA)⁴. NA-organisms may also have fermentative pathway(s)⁵. Although NO is not a free intermediate in the NA pathway, NA-organisms can produce detectable amounts of NO during nitrate ammonification, plausibly by a "side-reaction" of one or several enzymes^{5,6}. They can reduce NO to N₂O, be it by dissimilatory NO-reductase, such as qCu_ANOR, or by other NO reductases whose role appear to be to protect the cell against NO^[7].

To add to the complexity of this, reviewer #2 pointed out that organisms without nirK/S but with nitrate ammonifying enzymes and N₂O-reductase can still reduce NO₃⁻ all the way to N₂ if using Fe³⁺ as electron donor: N₂O is produced chemically by a reaction between Fe²⁺ and NO₂⁻ [8].

In conclusion, reviewer 4 is right: our statement in Box 1 (and in the text) that NNRB are unconditional net sinks for N₂O is too strong. We have rephrased this to claim that NNRB are net sinks for N₂O, unless equipped with the genes for nitrate ammonification.

This said, the issue is irrelevant for *Cloacibacterium* sp. CB-01, which was used in our study, because it lacks the genes for NA⁷.

- ¹ Hallin S, Philippot L, Löffler FE, Sanford RA, Jones CM (2018) Genomics and Ecology of Novel N₂O-Reducing Microorganisms. Trends in Microbiology, 26:43-55 <http://dx.doi.org/10.1016/j.tim.2017.07.003>
- ² Domeignoz-Horta LA, Putz M, Spor A, Bru D, Breuil MC, Hallin S, Philippot L (2016) Non-denitrifying nitrous oxide-reducing bacteria— An effective N₂O sink in soil. Soil Biol Biochem 103:376-379.
- ³ Conthe M, Wittorf L, Kuenen JG, Kleerebezem R, vanLoosdrechet MCM, Hallin S (2018) Life on N₂O: deciphering the ecophysiology of N₂O respiring bacterial communities in a continuous culture The ISME Journal (2018) 12:1142–1153 <https://doi.org/10.1038/s41396-018-0063-7>
- ⁴ Sun Y, deVos P, Heylen K (2016) Nitrous oxide emission by the nondenitrifying, nitrate ammonifier *Bacillus licheniformis*. BMC Genomics (2016) 17:68 DOI 10.1186/s12864-016-2382-2 cite: “Despite the absence of genes for dissimilatory nitrite reductase (NirK/S), these organisms can produce some N₂O, tentatively ascribed to some ability of other enzymes to catalyze the reduction of nitrite to NO, and NO to N₂O.”
- ⁵ Mania D, Heylen K, vanSpanning RJM, Frostegård A (2014) The nitrate-ammonifying and nosZ-carrying bacterium *Bacillus vireti* is a potent source and sink for nitric and nitrous oxide under high nitrate conditions. Environmental Microbiology (2014) 16(10), 3196–3210. Comment: As for Sun et al (2016), NO production is catalyzed by an unknown enzyme, while the reduction of NO to N₂O was ascribed to copper-NOR (qCuANor).
- ⁶ Mania D, Heylen K, vanSpanning RJM, Frostegård A (2015) Regulation of nitrogen metabolism in the nitrateammonifying soil bacterium *Bacillus vireti* and evidence for its ability to grow using N₂O as electron acceptor. Environmental Microbiology 18: 2937–2950 doi:10.1111/1462-2920.13124
- ⁷ Jonassen KR, Ormåsen I, Duffner C, Hvidsten TR, Bakken LR, HW Vick SHW (2022) A dual enrichment strategy provides soil and digestate competent nitrous oxide-respiring bacteria for mitigating climate forcing in agriculture. mBio, 13.3: e00788-22.
- ⁸ Onley JR, Ahshan S, Sanford RA, Löffler FE (2018) Denitrification by *Anaeromyxobacter dehalogenans*, a Common Soil Bacterium Lacking the Nitrite Reductase Genes nirS and nirK. Appl Env Microbiology 84: e01985-17 <https://doi.org/10.1128/AEM.01985-17>

C. The approach involves lab and field examinations of high density cultures of the strain *Cloacibacterium* sp. CB-01 cultivated in sludge digestate. The experiments show moderate and transient N₂O consumption over time based on O₂ and N₂O availability. The N₂O consumption activity of the CB-01 strain was evaluated against other microorganisms and shown to have a moderate V_{max} and low K_m. It showed a strong “bet-hedging” activity whereby only a small subset of cells showed N₂O consumption activity, suggesting that this is a weak activity for the cells. It is unknown how this feature contributes to the role the authors are claiming— that this strain could be a strong mitigator of N₂O emissions in all soils and throughout the growing season. The data and the author interpretation are contradictory.

Response: Our point is that the field experiments proved that CB-01 reduces N₂O emissions effectively, despite its apparently inferior biokinetic parameters, as determined in pure culture (nutrient broth), plausibly due to its tenacity and ability to remain active in soil.

Biokinetic parameters determined in pure culture when growing in artificial laboratory media (as we did for CB-01), are commonly used to judge (and compare) bacterial strains’ capacity to reduce N₂O emissions^{1,2}. Our results suggest that tenacity and ability to be metabolically active in soil are more important. Of note, the dual substrate enrichment culturing used to “find” CB-01 was designed to select for such tenacity in soil³.

Regarding bet-hedging: The referee is right that bet-hedging as demonstrated when CB-01 was growing in nutrient broth would severely limit its “N₂O-sink-strength”. In the original manuscript, we suspected that bet-hedging could be less pronounced when growing in digestate/soil, and follow-up experiments verified this. When growing in digestate, practically all CB-01 cells switched to respiring N₂O in response to hypoxia. This information is now included in the manuscript, and the kinetics proving the case are shown in Extended Data Fig. 6d-g.

Of note, the meagre effect of CB-01 in periods with low emissions, as demonstrated in Figure 3, could possibly be ascribed to CB-01's low affinity for N₂O, suggesting that strains with stronger affinity would have a better effect in such periods with low emissions. We refrain from drawing this conclusion, however, because the meagre effect of CB-01 in periods with low emissions could also be due to shut-down of the N₂O-respiration in CB-01 due to effective oxygenation of the entire soil matrix.

¹ Suenaga T, Hori T, Riya S, Hosomi M, Smets BF, Terada A (2019) Enrichment, Isolation, and Characterization of High-Affinity N₂O-Reducing Bacteria in a Gas-Permeable Membrane Reactor. *Environ. Sci. Technol.* 2019, 53, 12101–12112

² Jonassen KR, Ormåsén I, Duffner C, Hvidsten TR, Bakken LR, HW Vick SHW (2022) A dual enrichment strategy provides soil and digestate competent nitrous oxide-respiring bacteria for mitigating climate forcing in agriculture. *mBio*, 13.3: e00788-22.

D. There are controlled laboratory and field applications, but the authors did not use other microbial strains as positive or negative controls. They also did not control for digestate addition as an independent variable or the effect of digestate inoculation on soil pH. They only reported the many variables were uncontrolled in their field tests.

Response: The capacity of CB-01 to reduce N₂O emissions from denitrification in soil was compared with several other isolates in the foregoing paper and found to be superior¹. In the field experiments, we measured N₂O emissions from soils fertilized with digestate with CB-01, and the control buckets/plots were fertilized with the same CB-01-containing digestate in which CB-01 had been killed by a mild heat treatment (70 °C for 2 h), thus securing that the control-treatment received the same amounts and quality of organic C and ammonium as the treatment with live CB-01. This secured a stringent assessment of how metabolically active CB-01 cells affected the N₂O emissions.

¹ Jonassen KR, Ormåsén I, Duffner C, Hvidsten TR, Bakken LR, HW Vick SHW (2022) A dual enrichment strategy provides soil and digestate competent nitrous oxide-respiring bacteria for mitigating climate forcing in agriculture. *mBio*, 13.3: e00788-22.

E. The conclusions of the study are contradictory in several sections. The physiology and relative activity of strain CB-01 indicates that it will not be effective at reducing N₂O emissions from soils, yet the authors suggest the opposite. The data are far too preliminary and many variables were uncontrolled, suggesting low reliability of the main conclusion.

Response: We would appreciate more specifications here. Regarding the contradictions, we have already corrected some misunderstanding (paragraph C).

Regarding preliminary: CB-01 proved to be superior to a range of other NRB's in the foregoing laboratory incubations of soils¹. In the present study, we scaled up to field conditions, and demonstrated consistent strong effects through several experiments, including a range of soils. We would argue that this is a strong proof of concept.

^[1] Jonassen KR, Ormåsén I, Duffner C, Hvidsten TR, Bakken LR, HW Vick SHW (2022) A dual enrichment strategy provides soil and digestate competent nitrous oxide-respiring bacteria for mitigating climate forcing in agriculture. *mBio*, 13.3: e00788-22.

F. Suggested improvements for the text: The statement that "bacteria equipped only with the gene for N₂O reduction will be unconditional net sinks for N₂O whenever deprived of O₂" is overly simplistic and neglects to mention the other N-metabolism pathways and conditions that occur under low O₂ for these same bacteria. For instance, many bacterial strains that encode NosZII can

also perform DNRA, and both occur under O₂ deprivation, but the relative expression is dependent on other variables aside from O₂. Furthermore, conditions like copper deprivation will prevent N₂O removal, even under O₂ deprivation. This statement should be removed, revised, or followed by conditional statements.

Response: We agree, and we have revised the introduction accordingly (see paragraph B, and responses to the other reviewers).

G. The "sensus stricto" definition provided for NNRB is misleading as the presence of NO reductases suggest the possibility that the microbe can generate N₂O from. NRB that lack NirK/S can still generate N₂O using NOR enzymes or other N₂O-generating processes (e.g. cyt P460, cryptic nitrite reductase, globins, etc.). They can also have NosZ downregulated in favor of other processes (e.g. DNRA) that will prevent them from acting as N₂O sinks. Thus, the NNRB as defined are not always net sinks for N₂O, and their definition should thus be qualified.

Response: The argument is correct, and the issue about NOR was already addressed in the original manuscript: CB-01 (and any other NNRB with NOR) can produce N₂O from NO delivered by other organisms. But this production is probably marginal due to the limited amounts of NO available in the environment. For CB-01, we also argue that since the previous investigations suggested a marginal NO-reductase activity¹, N₂O-production by reducing external NO would be even more marginalized.

That said, the absence of genes for a respiratory NO-reductase (*NorB/Q*) is no guarantee that the organism cannot produce N₂O from NO. Many organisms are equipped with one or several other types of NO-scavenging enzymes, some oxidizing NO, some reducing it to N₂O. Their role appears to be protection against NO. We characterized one such NO-reductase (among several) in *E. coli*².

¹ Jonassen KR, Ormåsén I, Duffner C, Hvidsten TR, Bakken LR, HW Vick SHW (2022) A dual enrichment strategy provides soil and digestate competent nitrous oxide-respiring bacteria for mitigating climate forcing in agriculture. *mBio*, 13.3: e00788-22.

² Wang J, Vine CE, Balasiny BK, Rizk J, Bradley CL, Trejo MT, Poole RK, Bergaust L, Bakken LR, Cole JA (2018) The roles of the hybrid cluster protein, Hcp and its reductase, Hcr, in high affinity nitric oxide reduction that protects anaerobic cultures of *Escherichia coli* against nitrosative stress. *Molecular Microbiology*(2016)100(5), 877–892

H. Figure 1: The kinetic parameters for N₂O reduction imply that the microbe would need to be present at a high density and can only mitigate relatively high concentrations of N₂O due to its low affinity, impairing its utility in soils that are lower emitters of N₂O. This figure also seems to negate the potential for this strain in mitigating N₂O emissions universally in soil. The values reported in Table S5 are modelled based on the few experiments from this study under ideal conditions of high-density growth. While it is nice to see projected emissions reductions across Europe, there are a number of variables (regulation, other microbes, "bet-hedging" activity model, type of crop, physicochemical parameters) that will result in substantial error bars on these values.

Response: Yes, there is a need for high cell density in soil to achieve a reduction of N₂O, this is really the *raison d'être* of our approach. By using organic wastes that are destined for soil anyway, both as a substrate and vector, we can achieve high cell densities in soil with low costs.

I. The idea of growing CB-01 to high enough density throughout European soils to mitigate N₂O emissions does not take into account the consequences to the ecology of the soils systems. Table S5

did not consider AMF or variability in rhizosphere microbiota. Would encouraging resilient high-density growth of a foreign microbe be a positive factor for preserving other soil services (e.g. AMF, rhizosphere microbiome, nutrient cycles, etc.)? What are the possible unintended consequences of this proposal? The authors should have a section with caveats for soil engineering at this magnitude.

Response: We assume that AMF is Arbuscular Mycorrhizal Fungi (?). Indeed, there are a number of soil functions that could be affected by heavy inoculation with NNRB's, but to investigate all such perils would be a daunting task. We believe it is essential to focus on the most likely scenarios. In this instance, we particularly highlight two potential barriers: pathogenicity and the transmission of antibiotic resistance genes (ARG's). The genome of CB-01 contains no known ARG's, however, and neither any indications of pathogenicity. We have included this information in the revised version, and in Supplementary Information 9.

That said, we did a feeble attempt to investigate the effect of CB-01 on the soil microbiome by analyzing the composition based on 16S rRNA gene amplicon sequencing throughout the bucket experiment, which showed a strong effect of the digestate as such (with or without live CB-01), but transient: all treatments converged towards the end of the experiment (Extended Data Fig. 8).

J. Problematic statements: "Given the number of CB-01 cells added with the digestate (6.6×10^{13} cells m^{-2} soil surface), and the $V_{max} = 0.6$ fmol N_2O cell $^{-1}$ h $^{-1}$ (Fig S2), the potential N_2O -reduction rate, if all the added CB-01 cells were respiring N_2O at maximum rate, is 1.1 g N_2O-N m^{-2} h $^{-1}$ " According to the bet-hedging model, only a subset of cells can respire N_2O , so how can this calculation be accurate?

Response: This statement was meant to illustrate that the actual N_2O -reduction rate by CB-01 in the soil is only a fraction of the potential, if all cells were highly active and with ample provision of N_2O . The validity of our assessment is strengthened by the fact that our follow up experiments demonstrated that all cells switched to N_2O -respiration in response to hypoxia when growing in digestate (see our response above, paragraph C). This is made clear in the revised manuscript.

K. The peak N_2O -flux 1-2 days after fertilization was reduced by ~ 85 mg N_2O-N m^{-2} h $^{-1}$, which is ~ 8 % of the estimated potential.

Estimated potential based on what?

"For the subsequent peaks of N_2O -flux, the apparent N_2O -respiration by CB-01 (i.e. the reduction of the flux) was ≤ 4 mg N_2O-N m^{-2} h $^{-1}$, which is ≤ 0.36 % of the initial potential."

This measurement does not seem to indicate that these microbes are effective at reducing N_2O to a large extent. If both reductions of estimated potential are taken into account, then far less than 92% N_2O emissions control is possible.

Response: In this paragraph we take the observed reduction of N_2O emissions (i.e., the emissions in the control treatment minus that in the CB-01 treatment, mg N_2O-N m^{-2} h $^{-1}$) as a measure of the actual N_2O -reduction rate by the CB-01 in soil, and express this as % of the potential N_2O -reduction rate if all cells were actively respiring N_2O at maximum rate. This percentage is bound to be low, because the majority of the CB-01 cells in the soil are probably exposed to oxygen, thus shutting down their N_2O -respiration.

L. "This decline in apparent N₂O-respiration by CB-01 was plausibly a result of two factors: a gradually declining rate of N₂O provision by the indigenous microbiome, and a gradually declining number of CB-01 cells."

Doesn't this data indicate that CB-01 cells are not effective at controlling N₂O emissions if they are not able to take up N₂O and if they decline in number after the first N₂O peak? It seems the authors validate that CB-01 will be a weak controller of N₂O following the initial high emission peak in the paragraph following the one above.

Response: The data speak for themselves: there was a sustained reduction of N₂O emissions throughout the 93 days bucket experiment (Fig 2). This indicates that the initial CB-01 cell density exceeded the minimum for effectively reducing N₂O emissions. The slowly declining abundance of CB-01 speaks for itself: as for any other soil inoculant tested till now, they die out, fast or slow, and so will the potential for N₂O-mitigation. It would be naive to expect that one inoculation lasts much longer than a growth season.

M. The experiments with different soils, digestate, and CB-01 were not a good control for pH since the digestate increased the pH. Thus, the effect of CB-01 alone was not measurable independent from the digestate.

Response: It is not clear what is meant by the last sentence, but we assume that the concern here is that the soil-pH-effects on the capacity of CB-01 to reduce N₂O emissions is obscured by the fact that the mixture of soil and digestate (with and without CB-01) had higher pH than the original soil, and more so for the sandy soil than the clay loams (Table S3 in the original submission, now shown in Extended Data Fig.7). We agree. In fact, the experimental design effectively confounds pH-effects and any other soil specific effect on CB-01. We discussed this at some length in original manuscript.

N. The experiments from Fig. 5 seem to show that CB-01 is not effective over a long period of time. Even if the PCR experiment showed persistence of CB-01 genomes, their activity does not seem sustainable. The further caveats of differences between lab and field, soil moisture effects, and digestate application are additional complications that cast a shadow over the claimed benefits of CB-01 in mitigating N₂O flux. These problems should be addressed prior to making broad-based claims regarding the efficacy of this microbe for N₂O mitigation.

Response: We disagree. The inspection of CB-01-effects throughout the first field bucket experiment showed that CB-01 was effectively reducing emissions during periods with high emissions in the control soil (Figure 3), but had marginal effect in periods with moderate/low emissions. The period in question (20-290 days, Figure 5) had generally low emissions (< 0.4), in this case, the very low soil temperature could be of importance (this is commented in the revised manuscript).

O. Analysis of the microbiome only considered other bacteria and maybe archaea, but not fungi. This is a major oversight.

Response: We agree that this would be worth a study.

P. The calculations of national emission reductions are missing essential information. How did the authors assume that 60% emission reductions was a conservative estimate? Was this emission reduction from the first pulse? Based on annual reductions? Based on reduction from a single application of CB-01? Based on reduction from a single amendment of digestate? Based on reduction in moist or dry soils? Based on what pH regime? The authors say that this is NRB (shouldn't this be NNRB?) being introduced and applied to all liquid manure systems. So are these estimates only for soils that receive liquid manure? What about the other application issues mentioned above?

Response: We consider 60% to be a conservative estimate, given the emission reductions observed (Figs 2, 4 and 5).

Q. The future development section does not consider the rapid decline in activity of CB-01 over time or the expression in a sub-set of the applied population. Thus, the statement that CB-01 application can curb N₂O emissions throughout an entire growth season does not seem accurate based on the results from this study. The call for future NNRB strains with different, yet unknown, characteristics validates that this technology is not well understood and cannot be presented at this time as a viable means to reduce N₂O emissions.

Response: The observed effects of CB-01 on N₂O emissions are well understood, providing proofs of concept, and indications that the effect of the technology could be improved by isolating organisms with "stronger" biokinetic parameters than CB-01, such as N₂O-reduction at higher oxygen concentrations, and higher V_{max}/K_m .

R. Fig. S4 -- there seems to be an extra "N₂O" in the legend for panel A

Response: Thanks for spotting this error.

S. Fig. S5 -- based on the bet-hedging model for N₂O consumption demonstrated for this microbe, expression of NosZ is both transient and only for in a sub-population of cells. It is not understandable, then, how these physiological features would enable sustainable N₂O reduction in situ and across land-landscape scale in a predictable fashion. The authors should explain why bet-hedging activity is a positive feature for broad scale N₂O consumption.

Response: In the original manuscript, we emphasized that the bet-hedging, as observed, would weaken the capacity of CB-01 as an N₂O sink. However, we also expressed suspicion that bet-hedging would be less pronounced in digestate/soil. Follow-up experiments, as explained previously (Paragraph C) proved that all cells switched to N₂O-respiration (hence no bet-hedging) when growing in digestate. This is documented in the revised version.

T. References for N₂O emissions from soils, global sources of N₂O, and denitrifier genetics and biochemistry are outdated and should be updated.

Response: We assume that Zumft 1997 is considered outdated, and have replaced it with a more recent and comprehensive review of genes and enzymes of microbial N-transformations¹. We have also added a new reference regarding the global nitrogen cycle as affected by humans².

¹ Kuypers MMM, Marchant HK, Kartal B (2018) The microbial nitrogen-cycling network. *Nat Rev Micro* 16:263-276. Doi: 10-1038/nrmicro.2018.9

² Canfield DE, Glazer AN, Falkowski PG (2010) The evolution and future of the earth's nitrogen cycle. *Science* 330:192-196.

U. Clarity and context: The rationale for this study is sound and the mini-literature review in the beginning is compelling. However, the stated major outcome -- that CB-01 can mitigate N₂O emissions from all soils over full growing seasons -- is not substantiated by the data.

Response: Our data lend support to the claims, although we agree that more experiments are needed to assess the effects of our biotechnology under a range of agronomic and climatic conditions. To make the biotechnology robust, we will probably need more strains because there is a significant risk that repeated inoculation with the same strain over several years will result in proliferation of bacteriophages, thus accelerating the decline. Our research group is working along these lines and foresee that the present paper will inspire other research groups to do the same, thus securing progress.

Reviewer Reports on the First Revision:

Referees' comments:

Referee #1 (Remarks to the Author):

After the revision and addressing review comments the paper is improved. The originality and significance of the research remains novel and of high quality. The authors have made a number of revisions to the narrative that provides more clarity and sharpness and a number of grammatical errors have been fixed. The expansion and revisions to the supplemental materials are noted and address well the previous review comments and additional details of value.

I have no further suggestion for improvements.

Referee #2 (Remarks to the Author):

I have read the reviews, the authors' responses, and the revised manuscript files. All referees brought up valid points and the authors convincingly responded to the referees' input, which resulted in a refined manuscript. I have no comments for further improvement of the technical portion. I point out a few minor things that can be easily corrected.

The authors have conducted outstanding fundamental research on bacterial N₂O-reduction, and they transitioned these findings to an important real-world application. This is an important contribution with transformative character and a biotechnology to curb N₂O emissions has far-reaching implications. The authors' work is pioneering, will trigger many more investigations, and receive many citations.

Referees #1, 2, and 3 are supportive, referee #4 expresses concerns. Referee #4 brought up a several good points and the authors responded to the extent possible. Referee #4 rightfully raised the question about bet hedging, and the authors conducted an additional experiment. They now demonstrate that strain CB-01 cells switch to N₂O respiration during growth in digestate, illustrating that bet hedging is not of concern.

The authors' work extends fundamental science into the realm of applied research, what is excellent because it is at this interface where real progress occurs. I have editor experience, and I appreciate constructive referee criticism. I also developed an understanding for an 'the glass is half empty' versus 'the glass is half full' mentality, and it appears that Referee #4 does not fully acknowledge the opportunities the authors' work generates to address a pressing environmental problem. The fundamental science and applied research fields do not always speak the same language. Some of the questions Referee #4 raises, the authors can simply not address at this point. The authors' work is pioneering, it 'breaks the ice', and their work will trigger more investigations and field trials, which will ultimately answer outstanding questions, some of which Referee #4 asks. Of course, there are remaining challenges, but the authors' findings are convincing, well documented, and lay a strong foundation for additional efforts aimed at developing a robust biotechnology that addresses a climate grand challenge under a variety of soil geochemical and environmental conditions.

Minor corrections/suggestions:

1. BOX 1: NNRB. Genes should be formatted lower case and italicized.

Line 118: Consider replacing "they" with "these authors".

Line 120" Maybe say "The isolates obtained" ...

Lines 385-387: There may be other legislative hurdles, not just the presence of antibiotic resistance genes or pathogenicity. I suggest being more general as countries have different rules and regulations.

The citation list has formatting inconsistencies.

Referee #3 (Remarks to the Author):

I stated the following summary in my first review and I still agree with it after going through all the reviewer comments: *"The authors of this very well written manuscript present compelling results that demonstrate the augmentation of a previously isolated nitrous-oxide reducing bacterium, Cloacibacterium 19 sp. CB-01, reduced the N₂O emissions from soil 50% - 90% depending on the soil type. This organism expresses the NosZII nitrous oxide reductase and lacks the ability to reduce nitrate or nitrite. It is thus identified as a non-denitrifying N₂O respiring bacterium (NNRB). Its cell yield on N₂O, is demonstrated to be higher than microorganisms with a NosZI enzyme, coming in at 85% of the aerobic growth yield. Remarkably the data collected also suggest that this organism is bet-hedging during the transition to N₂O consumption; that is not all the members of the population are expression NosZ."*

In a response to a reviewer's comments that bet-hedging would limit the N₂O reducing ability of strain CB-01, the authors went back and showed that in the digestate used to add this strain to soil, all the cells were active in terms of N₂O reduction ability. I agree with the authors that this strengthens their paper. I would also add a few comments about interpretation of the bet-hedging results obtained in pure culture.

1. Figure 1 actually shows that N₂O is actually being depleted in the presence of O₂ (although I don't know if this loss is statistically supported). This implies to me that even in an artificial medium, that a subset of the population is ready to consume N₂O and does so when aerobic respiration is occurring.

2. The data shown fit a $F_{NosZ}=0.03$, which I interpret to be the fraction of cells with active NosZ when O₂ is depleted. It is important to note, however, in my opinion that this fraction is not a static number as depicted in this figure. The figure shows that the rate of N₂O reduction increases reflecting growth coupled to N₂O reduction. This would increase the fraction of N₂O reducers. Since the authors establish that the $F_{NosZ}=1$ in the digestate this point doesn't matter, but it does show that these organisms are not only active but they are capable of growth with N₂O.

I found that the author's responses to the four reviews and subsequent revisions to the manuscript are exceptional and as a result yield an even better paper than before.

Referee #4 (Remarks to the Author):

The authors took great care to answer each of my reviewer comments in great detail, which is appreciated. In particular, I appreciate the clarification of the types of physiological groupings of nitrogen metabolizing groups and when/where they become active in production or consumption of N₂O. The manuscript reads better than the original version. I have no further comments for the authors to consider.

Author Rebuttals to First Revision:**Referee #1 (Remarks to the Author):**

After the revision and addressing review comments the paper is improved. The originality and significance of the research remains novel and of high quality. The authors have made a number of revisions to the narrative that provides more clarity and sharpness and a number of grammatical errors have been fixed. The expansion and revisions to the supplemental materials are noted and address well the previous review comments and additional details of value.

I have no further suggestion for improvements.

Referee #2 (Remarks to the Author):

I have read the reviews, the authors' responses, and the revised manuscript files. All referees brought up valid points and the authors convincingly responded to the referees' input, which resulted in a refined manuscript. I have no comments for further improvement of the technical portion. I point out a few minor things that can be easily corrected.

The authors have conducted outstanding fundamental research on bacterial N₂O-reduction, and they transitioned these findings to an important real-world application. This is an important contribution with transformative character and a biotechnology to curb N₂O emissions has far-reaching implications. The authors' work is pioneering, will trigger many more investigations, and receive many citations.

Referees #1, 2, and 3 are supportive, referee #4 expresses concerns. Referee #4 brought up a several good points and the authors responded to the extent possible. Referee #4 rightfully raised the question about bet hedging, and the authors conducted an additional experiment. They now demonstrate that strain CB-01 cells switch to N₂O respiration during growth in digestate, illustrating that bet hedging is not of concern.

The authors' work extends fundamental science into the realm of applied research, what is excellent because it is at this interface where real progress occurs. I have editor experience, and I appreciate constructive referee criticism. I also developed an understanding for an 'the glass is half empty' versus 'the glass is half full' mentality, and it appears that Referee #4 does not fully acknowledge the opportunities the authors' work generates to address a pressing environmental problem. The fundamental science and applied research fields do not always speak the same language. Some of the questions Referee #4 raises, the authors can simply not address at this point. The authors' work is pioneering, it 'breaks the ice', and their work will trigger more investigations and field trials, which will ultimately answer outstanding questions, some of which Referee #4 asks. Of course, there are remaining challenges, but the authors' findings are convincing, well documented, and lay a strong foundation for additional efforts aimed at developing a robust biotechnology that addresses a climate grand challenge under a variety of soil geochemical and environmental conditions.

Minor corrections/suggestions:

1. BOX 1: NNRB. Genes should be formatted lower case and italicized.

Thanks for spotting this. Text revised accordingly

Line 118: Consider replacing “they” with “these authors”.

Agreed and changed.

Line 120” Maybe say “The isolates obtained” ...

Agreed and changed.

Lines 385-387: There may be other legislative hurdles, not just the presence of antibiotic resistance genes or pathogenicity. I suggest being more general as countries have different rules and regulations.

We agree, but retain the mentioning of pathogenicity- and antibiotic resistance -genes because we find it worthwhile to inform that we were unable to find such genes in our *Cloacibacter* strain.

The citation list has formatting inconsistencies.

We have used the Nature formatting style to the best of our abilities. Please let us know if there are any specific references that should be changed and we would be happy to improve it.

Referee #3 (Remarks to the Author):

I stated the following summary in my first review and I still agree with it after going through all the reviewer comments: *"The authors of this very well written manuscript present compelling results that demonstrate the augmentation of a previously isolated nitrous-oxide reducing bacterium, Cloacibacterium 19 sp. CB-01, reduced the N₂O emissions from soil 50% - 90% depending on the soil type. This organism expresses the NosZII nitrous oxide reductase and lacks the ability to reduce nitrate or nitrite. It is thus identified as a non-denitrifying N₂O respiring bacterium (NNRB). Its cell yield on N₂O, is demonstrated to be higher than microorganisms with a NosZI enzyme, coming in at 85% of the aerobic growth yield. Remarkably the data collected also suggest that this organism is bet-hedging during the transition to N₂O consumption; that is not all the members of the population are expression NosZ."*

In a response to a reviewer's comments that bet-hedging would limit the N₂O reducing ability of strain CB-01, the authors went back and showed that in the digestate used to add this strain to soil, all the cells were active in terms of N₂O reduction ability. I agree with the authors that this strengthens their paper. I would also add a few comments about interpretation of the bet-hedging results obtained in pure culture.

1. Figure 1 actually shows that N₂O is actually being depleted in the presence of O₂ (although I don't know if this loss is statistically supported). This implies to me that even in an artificial medium, that a subset of the population is ready to consume N₂O and does so when aerobic respiration is occurring.

We appreciate the comment, which is understandable because the N₂O concentration did decline with time (!). However, the decline in N₂O during the first 15 hours is accounted for by sampling loss, and the calculated N₂O respiration rate during this period is close to zero. To make this clear to the reader, we have added to following statement to the figure legend: *The decline of N₂O concentrations before ~18h is due to sampling loss.*

Readers with special interest in the issue can check this by inspecting the data uploaded in Figsare for Fig.1, where we have included the estimated electron flow rate to N₂O (which is indeed negligible until [O₂] approach depletion).

The issue of sampling loss is explained in Methods, with references to several of our previous papers.

Of note, the meticulous inspection of the onset of N₂O reduction as a function of O₂ concentration in Extended Data figure 2 provides ample evidence that N₂O respiration is effectively suppressed by [O₂] > ~3 μM.

2. The data shown fit a $F_{NosZ}=0.03$, which I interpret to be the fraction of cells with active NosZ when O₂ is depleted. It is important to note, however, in my opinion that this fraction is not a static number as depicted in this figure. The figure shows that the rate of N₂O reduction increases reflecting growth coupled to N₂O reduction. This would increase the fraction of N₂O reducers. Since the authors establish that the $F_{NosZ}=1$ in the digestate this point doesn't matter, but it does show that these organisms are not only active but they are capable of growth with N₂O.

This is correct: F_{NosZ} is the estimated fraction of cells that have synthesized NosZ at the time when oxygen is depleted. And yes: this is not a static number since the cells with *nosZ* will grow by respiring N₂O, while the others (without NosZ at oxygen depletion) are entrapped in anoxia without energy to synthesize NosZ. As a result, fraction of N₂O-respiring cells increases with time. In fact, this is quite clearly explained in the figure legend: "...only a fraction (F_{NosZ}) of the cells express NosZ and start growing by N₂O-respiration after O₂-depletion". The growth of the *nosZ* active cells is also shown in Extended data Fig 4b, and panel 4d shows the increasing fraction of NosZ-active cells.

I found that the author's responses to the four reviews and subsequent revisions to the manuscript are exceptional and as a result yield an even better paper than before.

Referee #4 (Remarks to the Author):

The authors took great care to answer each of my reviewer comments in great detail, which is appreciated. In particular, I appreciate the clarification of the types of physiological groupings of nitrogen metabolizing groups and when/where they become active in production or consumption of N₂O. The manuscript reads better than the original version. I have no further comments for the authors to consider.